# Building multi-satellite DEM time series for insight into mélange inside large rifts in Antarctica

**Menglian Xia**[1,2]**, Rongxing Li**[1,2]**, Marco Scaioni**[3]**, Lu An**[1,2]**, Zhenshi Li**[1,2]**, and Gang Qiao**[1,2]

[1]Center for Spatial Information Science and Sustainable Development, Tongji University,
1239 Siping Road, Shanghai, 200092, China
[2]College of Surveying and Geo-Informatics, Tongji University, 1239 Siping Road, Shanghai, 200092, China
[3]Department of Architecture, Built environment and Construction engineering (ABC), Politecnico di Milano,
via Ponzio 31, Milano 20133, Italy

**Correspondence:** Rongxing Li (ronli_282@hotmail.com)

**Abstract.** Front calving is a primary mechanism through which Antarctic ice shelves discharge ice mass into the Southern Ocean. It is an important process that influences ice shelf stability and thus, impacts the Antarctic Ice sheet's contribution to global sea level rise. Mélange dynamics inside rifts is recognized to potentially influence the rift propagation and subsequent iceberg calving. However, large-scale, high-resolution three-dimensional (3D) observations are scarce, which leads to their inability to capture small scale rift dynamics. Ultimately, the lack of knowledge in 3D rift structural changes and mélange dynamics hinders our understanding of the role of mélange in ice shelf retreat and mechanisms underlying the weakening of ice shelf stability. We propose an innovative multi-temporal digital elevation model (DEM) adjustment model (MDAM) to build a multi-satellite DEM time series from meter-level resolution small DEMs. It removes biases across large Antarctic ice shelves, as large as $\sim 6$ m in elevation, caused by tides, ice flow dynamics, and observation errors. Using 30 Reference Elevation Model of Antarctica (REMA) and ZY-3 sub-DEMs, we establish a cross-shelf DEM time series from 2014 to 2021 for the Filchner-Ronne Ice Shelf, the second largest ice shelf in Antarctica. This unified and integrated DEM series, with an unprecedented submeter elevation accuracy, reveals quantitative 3D structural and mélange features of two $\sim 50$ km long rifts, including rift lips, flank surface, pre-mélange cavities, and mélange elevations. For the first time, we have observed that while the mélange elevation in the rifts decreased by $2.7 \pm 0.6$ m from 2014 to 2021, the mélange within the rifts experienced a rapid expansion of $(10.31 \pm 0.03) \times 10^9$ km$^3$, or 139 %. This expansion is attributed to newly calved shelf ice from rift walls, associated rift widening, and other factors related to rift-mélange interactions. The developed MDAM system and the 3D mélange dynamics analysis methods can be applied for quantifying ice shelf instability and the future contribution of the Antarctic Ice Sheet to global sea level rise.

## 1 Introduction

The Antarctic Ice Sheet (AIS) has shown a persistent pattern of mass loss and has contributed to global sea level rise since the beginning of the satellite earth observation era in the 1960s (Jacobs et al., 1992; Allison et al., 2009; Rignot et al., 2019; Smith et al., 2020; Li et al., 2022, 2023a; Otosaka et al., 2023). The lost ice mass enters the Southern Ocean from ice shelves mainly through two processes, namely shelf front calving and basal melting, each accounting for $\sim 50$ % (Depoorter et al., 2013; Liu et al., 2015; Smith et al., 2020; Greene et al., 2022). In contrast to basal melting that occurs on the bottom of ice shelves, calving events are directly observed by using satellite remote sensing in the past decades. Examples of large iceberg calving include a major calving event in the Filchner Ice Shelf in 1986 with an area loss of $\sim 11\,500$ km$^2$ (Ferrigno and Gould, 1987), release of an iceberg of $\sim 5800$ km$^2$ from Larsen C Ice Shelf in 2017 (Larour et al., 2021; Wang et al., 2022), and break out of a part of the "loose tooth" of $\sim 1680$ km$^2$ from the Amery Ice Shelf

in 2019 (Francis et al., 2021; Walker et al., 2021). Although the mechanism of ice shelf caving is not fully understood, calving processes are clearly related to the evolution of rifts as they advect to the shelf front and cut icebergs (Joughin and MacAyeal, 2005; Hulbe et al., 2010; Walker and Gardner, 2019; Greene et al., 2022). Mélanges inside rifts, which consist of shelf ice, snow, sea ice, and water, have been investigated in relation to ice shelf fracturing in Antarctica and glacier calving in Greenland (Rignot and MacAyeal, 1998; Larour et al., 2004; Cassotto et al., 2021). Specifically, reductions in mélange thickness have been observed during rift widening on the Amery, Ronne, and Larsen C ice shelves (Fricker et al., 2005; Walker et al., 2021; Larour et al., 2021). Furthermore, modelling results indicate that a mélange may have the capability of transmitting stresses across rift flanks and thus, influences rift propagation and shelf front calving (Larour et al., 2004, 2021). However, high-resolution observations covering large regions are scarce and small-scale mélange textures and 3D rift dynamics cannot be captured with a necessary accuracy. This, in turn, hinders our understanding of the role of mélange in ice shelf retreating and the mechanism of ice shelf stability weakening.

In recent decades, altimetric and stereo mapping satellites have been used to acquire 3D data of the Earth surface features, which are further used to derive digital elevation models (DEMs) where 3D rift structure and mélange information is inherently available. DEMs from satellite altimetry data at the Antarctic continental scale have relatively lower resolutions, for example, the 500 m ICESat (DiMarzio et al., 2007) and ICESat-2 (Shen et al., 2022) DEMs, and the 1000 m CryoSat-2 DEM (Slater et al., 2018). The DEMs from optical and SAR satellite stereo mapping data are of higher resolutions and have the potential for geometric modeling and analysis of rifts and mélange features, including the 90 m TanDEM-X DEM (Wessel et al., 2021), the 30 m ASTER GDEM (Tachikawa et al., 2011), and especially, the 2 m REMA DEM (Howat et al., 2019). However, these high-resolution DEMs are derived from relatively narrow swath satellite data and have smaller extents. Thus, the smaller DEMs of different timespans are mosaiced together to cover large rifts and large ice shelves, in which significant discrepancies between individual DEMs are induced by influences of tides, ice flow dynamics, and observation errors (Fig. 1a). Existing methods for co-registration of DEMs are mostly developed based on global fitting and used in a land environment, e.g., a seven-parameter DEM transformation for mountain glacier mass balance estimation (Li et al., 2023b) and a global optimization process for computing three parameters between DEMs using all grid points (Nuth and Kääb, 2011). Furthermore, REMA DEMs are registered to a control data set, such as airborne altimetric data or satellite altimetric data of ICESat and CryoSat-2 (Howat et al., 2019; Shean et al., 2019; Zinck et al., 2023). While existing methods can be used to register DEMs, a mathematical model can help formalize the registration process and provide a consistent framework for quantifying and propagating uncertainties. Such a model-based approach may facilitate systematic comparison between different regions or time periods, and can offer clearer insight into the relationship between registration parameters and mélange dynamics.

We propose an innovative multi-temporal DEM adjustment model (MDAM) with the purpose to remove biases between adjacent sub-DEMs and to establish a unified and integrated multi-satellite DEM time series with a submeter elevation accuracy. We introduce tie points (TPs, see detailed description in Sect. 2.3) to connect the sub-DEMs and to formulate a DEM time series across a large ice shelf, which is then constrained to grounded regions using ground control points (GCPs). Since the GCPs are selected where ICESat-2 altimetric data are available, the connected DEM time series are then geometrically controlled at a centimeter elevation accuracy (Markus et al., 2017). We develop a data-driven approach to remove large offsets between adjacent sub-DEMs, namely the tide-induced elevation displacements and ice flow-induced horizontal discrepancies, by using a tide model and velocity maps, respectively. The residuals are then minimized in the MDAM system in a least-squares process to obtain the 3D correction parameters of each sub-DEM. The validated MDAM is applied to the Filchner Ice Shelf to establish a cross-shelf DEM time series from 2014–2021. The Filchner Ice Shelf (Fig. 1a) is one of the dual ice shelves of Filchner-Ronne Shelf that is the second largest ice shelf in Antarctica. Berkner Island and Coats Land are located at the western and eastern margins of the ice shelf, respectively. The largest calving event recorded for this ice shelf, with an area loss of $11\,500\,\mathrm{km}^2$, occurred in 1986 due to a rapid propagation of a prominent rift known as "Grand Chasm" (Ferrigno and Gould, 1987), producing three giant icebergs that greatly impacted the circulation and hydrography in the nearby ice shelf-ocean system (Grosfeld et al., 2001) (Fig. A1). Two large rifts, T1 and T2 (Fig. 3), were detected in a Landsat satellite image in 1996 and an ARGON satellite image in 1963, respectively (Li et al., 2017a; Walker and Gardner, 2019; Lv et al., 2022). These rifts have a combined length of $\sim 100\,\mathrm{km}$ and exhibit a similar propagation pattern as Grand Chasm. We demonstrate the capability of the MDAM by quantitatively characterizing 3D structural and mélange features of these two $\sim 50\,\mathrm{km}$ long rifts, including rift lips, flank surface, pre-mélange cavities, and mélange elevations. Finally, we demonstrate the observed mélange changes and estimated volumetric changes in relation to rift widening as the rifts advect toward the ice shelf front during the study period.

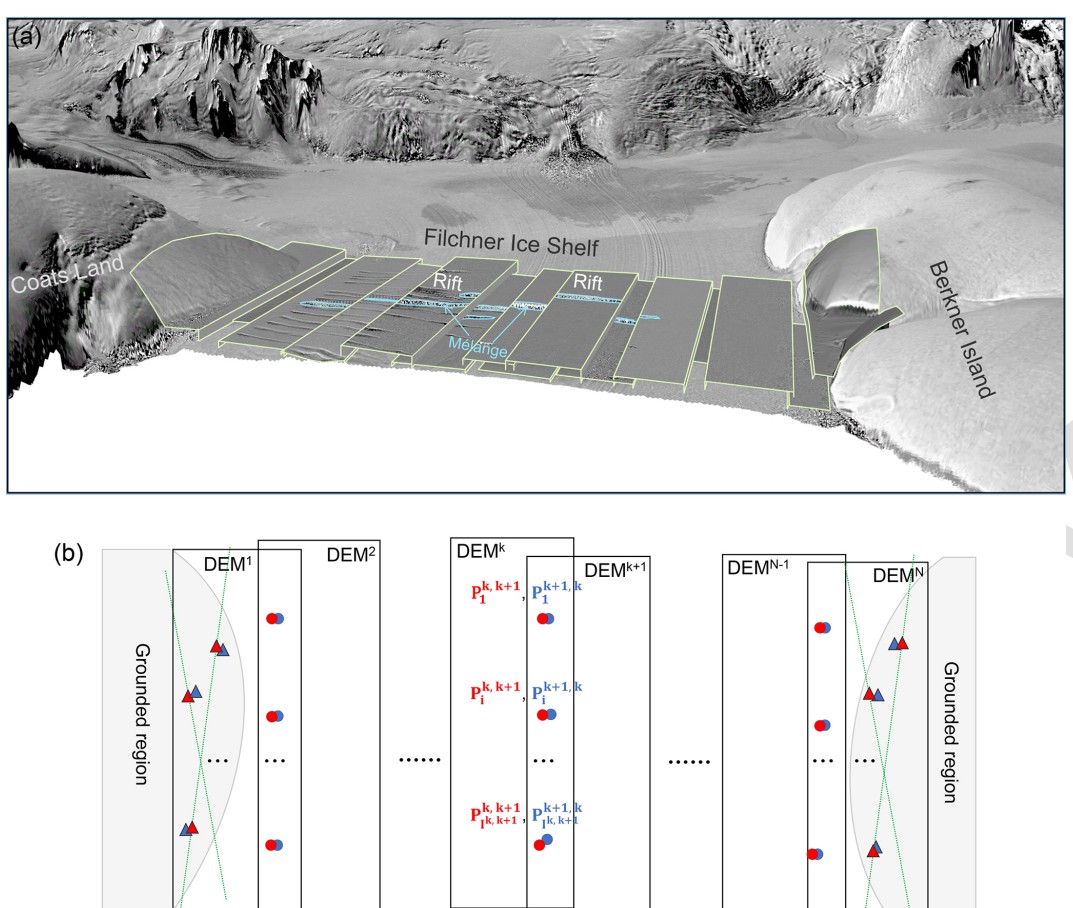

**Figure 1. (a)** High-resolution small-coverage DEMs (e.g., REMA strips of $\sim 16$–$18\,\text{km} \times \sim 110$–$120\,\text{km}$) are unified and integrated for accurate 3D rift structural and mélange dynamic monitoring in a large Antarctic ice shelf environment (e.g., $\sim 165\,\text{km}$ wide Filchner Ice Shelf). The LIMA mosaic (Bindschadler et al., 2008) is draped on the RAMP DEM (Liu et al., 2001) with a vertical exaggeration factor of 40 times. **(b)** Concept of the MDAM system to connect sub-DEMs across the ice shelf through "tie points" (TP, represented by circles) to create a DEM time series. The "ground control points" (GCP, represented by triangles) are used to fix the combined DEM model to grounded regions at ice shelf margins. The closely located red and blue circles (triangles) are from adjacent sub-DEMs and form corresponding TP (GCP) pairs.

## 2 Data and Method

### 2.1 Data

The sub-DEMs used in this study are REMA strips and a Ziyuan-3 (ZY-3) DEM which are generated from stereo satellite images of WorldView (Anderson and Marchisio, 2012) and ZY-3 (Wang et al., 2014), respectively (Table A1). Both are formed by the along-track stereo mechanism (Li, 1998). REMA DEMs are available for the entire Antarctica and are produced from WorldView images using an open-source automatic photogrammetric processing software system, Surface Extraction from TIN-based Searchspace Minimization (SETSM) (Noh and Howat, 2015). The WorldView data set includes WorldView-1, -2 and, -3, and GeoEye-1 images with a small swath of 13.1–17.6 km and a high resolution of 0.3–0.5 m (Howat et al., 2019). Thus, each derived REMA

DEM provides a high level of ice surface details, but with a relatively small extent, resulting in difficulties in covering large rifts or the entire ice shelf. More specifically, we use REMA DEM strips of 2016–2017 (Fig. 3a) and 2020–2021 (Fig. 3b) with a grid spacing of 2 m. On the other hand, ZY-3, launched in 2012, is the first Chinese high-resolution natural resource mapping satellite (Wang et al., 2014). It is equipped with a three-line mapping system that forms quasi-real-time along-track stereo image pairs using fore (3.5 m), nadir (2.1 m), and aft (3.5 m) views. The fore and aft combination has a large intersection angle of 47°, facilitating a high elevation accuracy for DEM generation. Its wide swath of $\sim 52\,\text{km}$ ensures a complete coverage of large rifts, such as T1 and T2 (Fig. 3a). The ZY-3 DEM used in this study has a time tag of 28 February 2014 and a grid spacing of 5 m. It has a relatively narrow strip because of cloud cover-

age, but it covers both T1 and T2. The DEM is registered to the ICESat-2 altimetry data and has an elevation accuracy of 0.3 m. The ZY-3 DEM of 2014 has a full coverage of T1 and T2 and can provide a reference for comparison with the ad-
5 justed sub-DEMs of REMA. The combined ZY-3 and REMA sub-DEMs form a seven-year long time series from 2014 to 2021 with a time interval of 3–4 years, which can be used to analyze mélange elevation and volume changes.

We use elevation points of the ICESat-2 ATL06 data from
10 2019 to 2021 (Markus et al., 2017) to register the ZY-3 DEM. The GCPs are also selected on or close to ICESat-2 ATL06 elevation points to establish a unified reference system in MDAM. To ensure that the GCPs are "stable" during the timespan between the ZY-3 DEM (REMA sub-DEMs) and
15 ICESat-2, these GCPs are further required to be on grounded features with a low velocity ($< 10$–$20\,\mathrm{m\,yr^{-1}}$). In a special case we also select GCPs on the ice shelf where both the DEM images and ICESat-2 data were acquired within one day. The ICESat-2 data are also used for validation of the
20 bias-corrected DEM time series. Along each of the six beam tracks, ICESat-2 ATL06 data provide aggregated elevation points with a 20 m spacing and a revisit cycle of 91 d. Field validation studies have demonstrated that the ICESat-2 data reached an elevation accuracy within 2–4 cm in Antarctica
(Brunt et al., 2019; Magruder et al., 2020; Li et al., 2021).

## 2.2 Multi-temporal DEM adjustment model (MDAM)

As illustrated in Fig. 1b, a set of sub-DEMs ($\mathrm{DEM^1}$, $\mathrm{DEM^2}$, ..., $\mathrm{DEM^k}$, ..., $\mathrm{DEM^{N-1}}$, $\mathrm{DEM^N}$) are connected through TPs in the overlapping areas between adjacent sub-DEMs.
The two sub-DEMs at the two shelf margins ($\mathrm{DEM^1}$ and $\mathrm{DEM^N}$) cover grounded regions, while those in between (from $\mathrm{DEM^2}$ to $\mathrm{DEM^{N-1}}$) cover the floating ice shelf. GCPs are used to fix the entire DEM time series to the grounded regions. They are input into the multi-temporal DEM ad-
justment model (MDAM) for removing biases between adjacent sub-DEMs (Fig. 2). Initially, relatively large vertical offsets, up to $\sim 1.2\,\mathrm{m}$, caused by tides are corrected for all sub-DEMs by using a tide model, CATS2008, with an accuracy of $\sim 8\,\mathrm{cm}$ (Padman et al., 2008; King et al., 2011). Then
we manually select TPs in the overlapping areas between sub-DEMs on the floating ice to cover the entire ice shelf (see Sect. 2.3 for details). The relatively large horizontal displacements between corresponding TP pairs (Fig. 1b), up to hundreds of meters, induced by ice flow dynamics are com-
pensated by using ITS_LIVE velocity maps (Gardner et al., 2019). At the same time, we tie the GCP locations to satellite altimetry data of ICESat-2 (see Sect. 2.3 for details). In the model inconsistencies at TP and GCP pairs are first reduced by the tide and velocity corrections, leaving the uncertainty
originated from the sub-DEM production as residuals, including photogrammetric measurement errors and ephemeris data errors. Thereafter, we use the least-squares adjustment method (McGlone et al., 2004) to estimate the unknown bias

corrections of all sub-DEMs by minimizing the residuals (Li, 1998; McGlone et al., 2004; Shean et al., 2019). Finally, the 55 corrected sub-DEMs are applied for 3D analysis and modeling of large rifts and mélange dynamics.

For each pair of TPs $P_i^{k,k+1}$ and $P_i^{k+1,k}$ ($i = 1, \ldots, i^{k,k+1}$) (Fig. 1b), we measure 3D coordinates ($X_{P_i}^k$, $Y_{P_i}^k$, $Z_{P_i}^k$) from $\mathrm{DEM}^k$ and ($X_{P_i}^{k+1}$, $Y_{P_i}^{k+1}$, $Z_{P_i}^{k+1}$) from $\mathrm{DEM}^{k+1}$. We define 60 3D translational parameters of ($\mathrm{d}X^k$, $\mathrm{d}Y^k$, $\mathrm{d}Z^k$) as unknowns for bias removal. Thus, we establish the 3D positional relationships at tie point $P_i^{k,k+1}$ between $\mathrm{DEM}^k$ and $\mathrm{DEM}^{k+1}$ as observation equations for each pair of TPs:

$$\begin{cases} X_{P_i}^k + \mathrm{d}X^k + \mathrm{Vel}X_i^k \times \left(t^{k+1} - t^k\right) = X_{P_i}^{k+1} + \mathrm{d}X^{k+1} + \varepsilon_{x_i}^k \\ Y_{P_i}^k + \mathrm{d}Y^k + \mathrm{Vel}Y_i^k \times \left(t^{k+1} - t^k\right) = Y_{P_i}^{k+1} + \mathrm{d}Y^{k+1} + \varepsilon_{y_i}^k \\ Z_{P_i}^k + \mathrm{d}Z^k = Z_{P_i}^{k+1} + \mathrm{d}Z^{k+1} + \varepsilon_{z_i}^k \end{cases} \quad (1)$$
65

where $k = 1, \ldots, N-1$; and $i = 1, \ldots, i^{k,k+1}$. $t^k$ and $t^{k+1}$ are the known data acquisition times for $\mathrm{DEM}^k$ and $\mathrm{DEM}^{k+1}$. $\mathrm{Vel}X_i^k$ and $\mathrm{Vel}Y_i^k$ are the ice flow speed values in $X$- and $Y$-directions at the TPs given in a velocity map. Here, ITS_LIVE image-pair velocity maps are selected to spatially 70 match the extent of each sub-DEM and to temporally align as closely as possible with the acquisition times of two adjacent sub-DEMs. Similarly, ($\mathrm{d}X^{k+1}$, $\mathrm{d}Y^{k+1}$, $\mathrm{d}Z^{k+1}$) are unknown bias correction parameters for $\mathrm{DEM}^{k+1}$. ($\varepsilon_{x_i}^k$, $\varepsilon_{y_i}^k$, $\varepsilon_{z_i}^k$) represent residuals at TPs in the observation equations, including 75 photogrammetric measurement errors and ephemeris data errors.

In a similar way, observation equations for ground control point $P_{\mathrm{GCP}_i}^k$ are given as:

$$\begin{cases} X_{\mathrm{GCP}_i}^k + \mathrm{d}X^k = X_{P_i}^{\mathrm{GCP},k} + \varepsilon_{x_i}^{\mathrm{GCP},k} \\ Y_{\mathrm{GCP}_i}^k + \mathrm{d}Y^k = Y_{P_i}^{\mathrm{GCP},k} + \varepsilon_{y_i}^{\mathrm{GCP},k} \\ Z_{\mathrm{GCP}_i}^k + \mathrm{d}Z^k = Z_{P_i}^{\mathrm{GCP},k} + \varepsilon_{z_i}^{\mathrm{GCP},k} \end{cases} \quad (2)$$
80

where $k = 1$ and $N$; and $i = 1, \ldots, i^{\mathrm{GPC},k}$. ($X_{\mathrm{GCP}_i}^k$, $Y_{\mathrm{GCP}_i}^k$, $Z_{\mathrm{GCP}_i}^k$) are 3D coordinates measured at the GCPs from $\mathrm{DEM}^k$. ($X_{P_i}^{\mathrm{GCP},k}$, $Y_{P_i}^{\mathrm{GCP},k}$, $Z_{P_i}^{\mathrm{GCP},k}$) are 3D coordinates at the same point from the control data set (e.g., ICESat-2). ($\varepsilon_{x_i}^{\mathrm{GCP},k}$, $\varepsilon_{y_i}^{\mathrm{GCP},k}$, $\varepsilon_{z_i}^{\mathrm{GCP},k}$) represent residuals at GCPs in the observa- 85 tion equations, describing inconsistencies between the integrated DEM and the outside control data (e.g., ICESat-2). Furthermore, it is noticed that in the study area, elevation disagreements between adjacent REMA DEMs present a linear trend in the $Y$-direction (mainly ice flow direction). Thus, we 90 use a linear term of $a_o^k + a_1^k Y_{\mathrm{OL}_i}^k$ to replace $\mathrm{d}Z^k$ in Eqs. (1) and (2). Here, $a_o^k$ and $a_1^k$ are coefficients as unknowns, and $Y_{\mathrm{OL}_i}^k$ is the $Y$ coordinate defined in the local central coordinate system in the overlapping region, with the origin defined at the center of the overlapping area. 95

Equations (1) and (2) are combined and populated with all TPs and GCPs of the sub-DEMs. They are further formulated as an observation equation system in a matrix form. Finally,

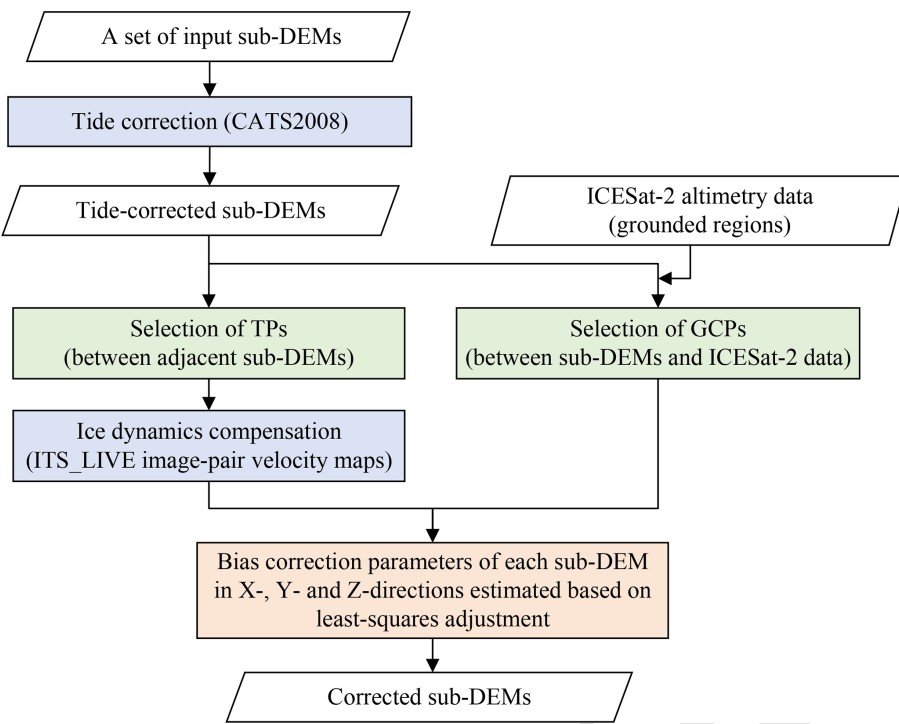

**Figure 2.** Data processing principle of the multi-temporal DEM adjustment model (MDAM).

the unknowns, namely bias correction parameters ($dX^k, dY^k$, $a_o^k$, and $a_1^k$) ($k = 1, \ldots, N$), are solved by an optimization procedure using the least-squares principle (McGlone et al., 2004). Uncertainties of the estimated bias correction parameters ($\sigma_{x_i}^k$, $\sigma_{y_i}^k$, $\sigma_{z_i}^k$, $\sigma_{x_i}^{\text{GCP},k}$, $\sigma_{y_i}^{\text{GCP},k}$, $\sigma_{z_i}^{\text{GCP},k}$) are computed through an error propagation within the optimization procedure.

## 2.3 Implementation considerations

### 2.3.1 Ground Control Point Selection

Selection of high quality GCPs is of great importance to the accuracy of the entire DEM time series. We choose GCPs on the grounded regions at both margins of the ice shelf (Figs. 1b and 3) with considerations of stability, low ice velocity, and local peaks (Feng et al., 2022). Meanwhile, the GCPs are also chosen to be on or close to the ICESat-2 ATL06 points from which their 3D coordinates can be measured. Despite the absence of outcrops and blue ice in the grounded regions of this study, we are able to identify relatively stable features for GCP selection, aided by multisource data, e.g., REMA DEMs, ICESat-2 data, ice flow velocity maps (Rignot et al., 2011), and bed topography (Morlighem et al., 2020).

Figure 4 illustrates an example of GCP selection on the ice surface of Coats Land. Within a local area we drape a shaded relief map and a velocity map (Rignot et al., 2011) on the REMA DEM to make sure that a GCP candidate is selected not on a steep slope ($< 5°$) and with a low ice flow speed ($< 10\,\text{m}\,\text{yr}^{-1}$) (Fig. 4a and b). The candidate is approximately localized at a local peak (red triangle). In a close-up view (Fig. 4c and d), an ICESat-2 elevation profile (blue line) is further used to finalize the GCP position by adjusting the candidate position to the local maximum of the terrain (red cross). Finally, 3D coordinates of the GCP are measured on the ICESat-2 profile as knowns ($X_{P_i}^{\text{GCP},N}$, $Y_{P_i}^{\text{GCP},N}$, $Z_{P_i}^{\text{GCP},N}$) and from the sub-DEM as observations ($X_{\text{GCP}_i}^N$, $Y_{\text{GCP}_i}^N$, $Z_{\text{GCP}_i}^N$) that are used in Eq. (2). The above GCP selection method is implemented as a manual procedure.

### 2.3.2 Tie Point Selection

The key consideration for TP selection is to choose features that are identifiable on both adjacent sub-DEMs and unchanged during the time interval between the acquisitions of the sub-DEMs (Figs. 1b and 3). Further specific rules are followed: (1) if applicable, TPs be selected at distinct features, such as crevasses, rifts, and other relatively "stable" ice surface features; (2) elevated locations are preferred to avoid influences of shadows casted by different lighting conditions and obstacles in 3D visualization; and (3) TPs be approximately evenly distributed in the overlapping area. For example, to select a TP candidate on a rift lip, two shaded relief maps are produced from the adjacent sub-DEMs and used for 3D visualization (Fig. 5a and b) to check whether

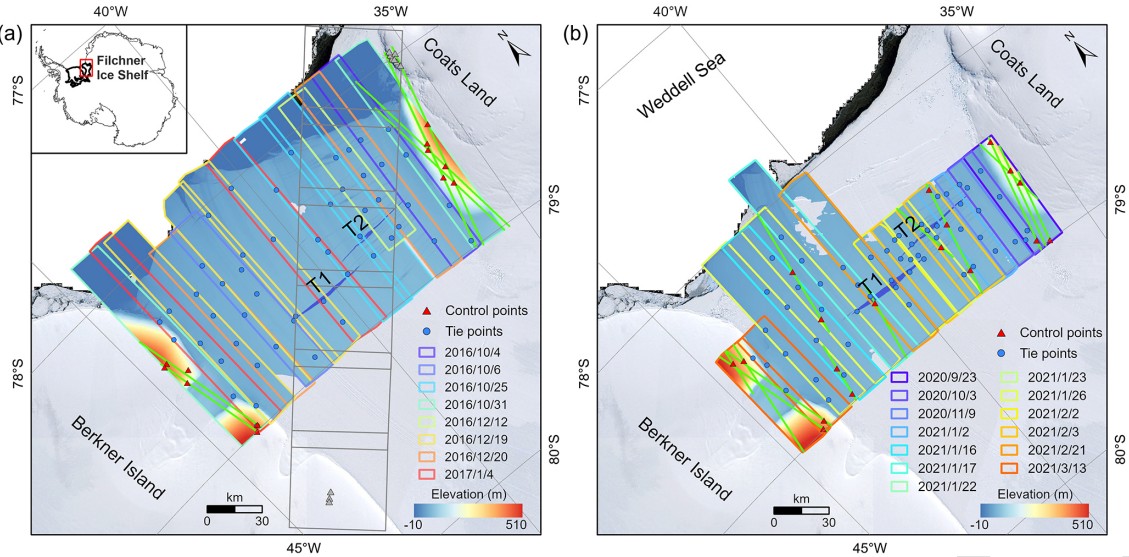

**Figure 3.** The Filchner Ice Shelf used as a validation and application site for the MDAM system. Colored boxes are REMA DEM coverages. Green lines at Berkner Island, Coats Land and on the ice shelf are ICESat-2 tracks. Blue circles are TPs and red triangles are GCPs. **(a)** REMA DEMs of 2016–2017, ZY-3 DEM of 2014 (light gray lines), and rifts T1 and T2. **(b)** REMA DEMs of 2020–2021. GCPs are also selected on the ice shelf where both the DEM images and ICESat-2 data were acquired within one day. Background elevation information is from the REMA DEM (Howat et al., 2019).

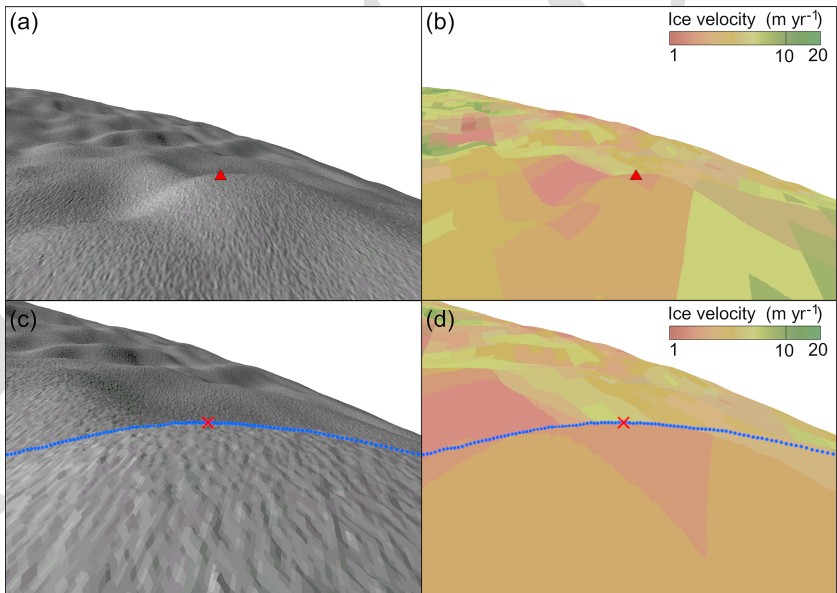

**Figure 4.** Local 3D view for GCP candidate (red triangle) selection: shaded relief map **(a)** and velocity map **(b)** draped on the REMA DEM. 3D close-up view for GCP finalization (red cross): local maximum selected with the help of the ICESat-2 elevation profile (blue line) **(c)** and velocity map **(d)**. TS1

the feature (orange and blue dots in Fig. 5) is changed during the time interval between the two sub-DEMs. The TP is finalized (red cross) by a verification using a 3D Landsat image view with a different illumination geometry (Fig. 5c). Figure 5d–f show another example of TP selection on a snow bridge that is formed by an ice surface process when snow is

blown into a crevasse and a rift or accumulated about tens of meters above the shelf surface (Williams et al., 2014; Thompson et al., 2020). Similarly, Fig. 5d and e show the 3D views from shaded relief maps with lower elevation angles produced from the adjacent sub-DEMs. After a verification using a Landsat image (Fig. 5f), the peak of the snow bridge

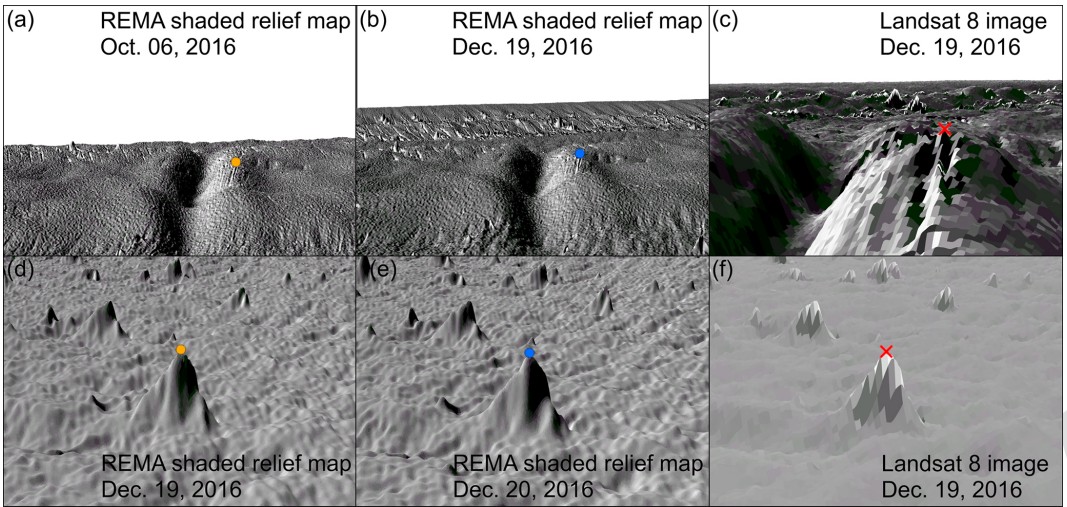

**Figure 5.** TP selection on rift lip: 3D views from two shaded relief maps of adjacent REMA DEMs **(a)** and **(b)**, and verification using a 3D Landsat image view **(c)**. TP selection on snow bridge: 3D views from two shaded relief maps of adjacent REMA DEMs **(d)** and **(e)**; verification using a 3D Landsat image view **(f)**.

is defined as the TP. In both cases, 3D coordinates $(X_{P_i}^k, Y_{P_i}^k, Z_{P_i}^k)$ and $(X_{P_i}^{k+1}, Y_{P_i}^{k+1}, Z_{P_i}^{k+1})$ are measured and used in Eq. (1). The TP selection method is also implemented as a manual procedure.

### 2.3.3 Mélange Volume Estimation

To compute the mélange volumetric change, we apply the hydrostatic equilibrium principle to obtain the unknown thickness $H$ of the mélange in each transect (Fricker et al., 2005):

$$H = \frac{\rho_{\text{water}}}{\rho_{\text{water}} - \rho_{\text{mélange}}} (h - h_{\text{sea}}) \qquad (3)$$

where $h$ is the average mélange elevation along the transect and $h_{\text{sea}}$ is the average sea level ($-15.2$ m) estimated from the ICESat-2 altimetry data within a range of $\sim 10$ km from the ice shelf front from March 2019 to December 2021 during which the ICESat-2 data become stable after the initial stage of the mission (Magruder et al., 2021). $\rho_{\text{water}}$ and $\rho_{\text{mélange}}$ are densities of seawater and mélange (1028 and 865 kg m$^{-3}$), respectively (King, 1994; Fricker et al., 2005; Adusumilli et al., 2020). The estimated change in mélange thickness $H$ is relevant to ice sheet modeling. Consequently, the mélange volume of the transect is the product of the thickness $H$, rift width $W$, and transect interval, which is integrated over all transects to compute the mélange volume of the entire rift: $\Sigma (H \times W \times \text{transect interval})$.

### 2.4 Bias correction

The proposed MDAM system estimates the bias corrections $(\mathrm{d}X^k, \mathrm{d}Y^k, \mathrm{d}Z^k)$ for all sub-DEM $\text{DEM}^k$ ($k = 1, 2, \ldots, N$) through the least-squares procedure where their uncertainties are provided in the covariance matrix (McGlone et al., 2004).

The REMA DEMs of 2016–2017 consists of 14 sub-DEMs that cover the entire front part of Filchner Ice Shelf (Fig. 3a). We use 12 GCPs that are measured by using ICESat-2 data to tie the connected DEMs to the grounded regions of Berkner Island and Coats Land where the average ice flow speed at the GCPs is $\sim 8$ m yr$^{-1}$. In total 51 TPs are used to connect the sub-DEMs on the floating part of the ice shelf, with 3–4 evenly distributed TPs in each overlapping area. Horizontal displacements at the TPs caused by the ice flow are on average $\sim 90$ m and are corrected by using the velocity map (Gardner et al., 2019). We establish the observation equations in Eqs. (1) and (2) using different weights. The weights for TPs are computed as inverse distances from the TPs to the nearest GCPs on grounded regions. Hence, the weights for GCPs are set to 1. Those TPs that are farther away from the grounded regions have smaller weights.

We process the second set of REMA DEMs of 2020–2021 (Fig. 3b) in the same way in the least-squares process. In addition, the ZY-3 DEM of 2014 is reconstructed as a cross-ice-shelf DEM and does not need to go through this bias-correction process. The DEM is co-registered to the ICESat-2 ATL06 data of 2019 through a bundle adjustment procedure (McGlone et al., 2004; Li, 1998; Li et al., 2017b) using GCPs that are selected from "stable" features in the same way for those used in REMA sub-DEM co-registration.

## 3 Results

### 3.1 Adjusted DEM time series

The bias corrections $(\mathrm{d}X^k, \mathrm{d}Y^k, \mathrm{d}Z^k)$ for all sub-DEM $\text{DEM}^k$ ($k = 1, 2, \ldots, 14$) in the REMA DEMs of 2016–2017 are listed in Table A2. As illustrated in Fig. 6, uncertainties

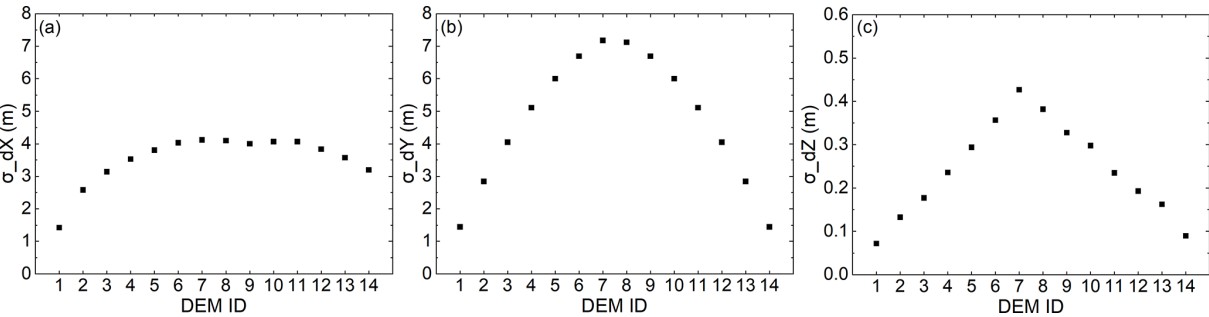

**Figure 6.** Illustration of uncertainties of the estimated bias corrections ($dX^k$, $dY^k$, $dZ^k$) of the REMA DEMs of 2016–2017 in Table A2.

of the corrections in both horizontal and elevation directions are effectively controlled for sub-DEMs close to grounded regions in Berkner Island and Coats Land through GCPs, leaving those in the middle section of the ice shelf relatively higher. The average uncertainty of the elevation corrections is 0.2 m, with the maximum of 0.4 m. The bias corrections ($dX^k$, $dY^k$, $dZ^k$) for the REMA DEMs of 2020–2021 are also listed in Table A2. The bias-corrected DEMs have an average uncertainty of 0.2 m for the elevation bias corrections. After a co-registration of the ZY-3 DEM of 2014 to the ICESat-2 ATL06 data of 2019, an elevation accuracy of 0.3 m is achieved.

These corrections are then applied to each sub-DEM to produce the bias-corrected REMA DEMs of 2016–2017. The effectiveness of the MDAM system is clearly demonstrated by a dramatic reduction of the elevation offset between adjacent sub-DEMs at the 51 TPs and 12 GCPs (Fig. 7), in average, from 12.7 m before bias-correction (black arrows) to 0.2 m thereafter (red arrows, also called residuals). Residuals vary from 0.1 to 0.5 m.

To validate the bias-correction results using an external dataset, we directly compare the bias-corrected REMA DEMs of 2020–2021 with two tracks of ICESat-2 ATL06 data acquired from January to February 2021 (green lines in Fig. 8a). The ICESat-2 data set is chosen because of its high elevation accuracy and along-track point density. Furthermore, we select two tracks of the ICESat-2 data that were acquired on the same times (within one day) as the REMA sub-DEMs to avoid displacements caused by the shelf ice advection and internal mélange changes. The timespans of other two sets of DEMs, REMA DEMs 0f 2016–2017 and ZY-3 DEM, fall outside the ICESat-2 temporal coverage. Along Orbit 0375 Ground track GT3L, REMA DEM points prior to bias correction (gray) are relatively close to the ICESat-2 ATL06 elevation points (blue points), with a smaller average difference of 1.1 m (Fig. 8b). However, the average difference of 11.2 m along Orbit 0634 Ground track GT3L is significantly larger (Fig. 8c). After bias-corrections, the differences are effectively controlled within 0.1 m on these two tracks and 0.2 m on mélange top.

## 3.2   3D mélange dynamics from 2014–2021

Based on the registered ZY-3 DEM of 2014 and bias-corrected sub-DEMs of REMA DEMs of 2016–2017 and 2020–2021, we establish a multi-satellite DEM time series of T1 and T2, from 2014 to 2021 (Fig. 9, Table A1). The 2D maps of the DEM time series are presented in Fig. A2. Each of these periodical unified and integrated DEMs presents the unprecedented 3D structural information of these $\sim 50$ km long rifts. The unique DEM time series allows us to exploit 3D structure characteristics, including ice surface flank, rift lips, and mélange top and thickness (Fig. 10c and g). For the first time, we can accurately compute mélange elevation and volume changes within the entire rifts as they propagate and advect towards the shelf front during the period of 2014–2021 (Fig. 10d and h). To quantify 3D dynamic mélange changes within the rifts, we define a transect system for each rift based on the ZY-3 DEM of 2014 with transects spaced at an interval of 1 km along the rift centerline from the west tip to the east tip, which is also applied to the bias-corrected REMA DEMs of 2016–2017 and 2020–2021.

The lengths of Rifts T1 and T2 are 47.1 and 48.3 km (maximum widths of 1.5 and 1.5 km), respectively, measured on the ZY-3 DEM of 2014. Based on the bias-corrected DEM time series from 2014 to 2021 (Fig. 9), we found that T1 and T2 propagated consistently during the period, resulting in an accelerated widening by 646 m (227 m yr$^{-1}$) and 412 m (145 m yr$^{-1}$), respectively, during 2014–2017; and 1109 m (268 m yr$^{-1}$) and 834 m (203 m yr$^{-1}$), respectively, during 2016–2021. Correspondingly, T1 and T2 have lengthened by 2393 m (840 m yr$^{-1}$) and 1701 m (597 m yr$^{-1}$), respectively, during 2014–2017; and 1082 m (262 m yr$^{-1}$) and 1667 m (405 m yr$^{-1}$), respectively, during 2016–2021. Based on the REMA DEMs of 2016–2017, the rift lips on the seaward sides are mostly higher than that on the landward side in the middle sections of the rifts (Table A3), with an average difference of 0.15 m. This phenomenon of the greater heights of seaward rift lips based on the precise measurements from the bias-corrected sub-DEMs is consistent with the results for some rifts on other Antarctic ice shelves presented in Walker and Gardner (2019 TS2).

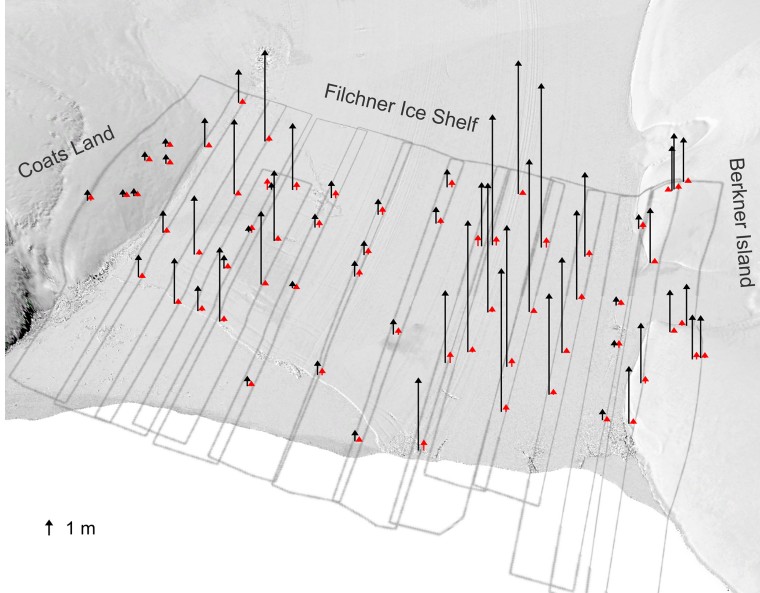

**Figure 7.** Elevation offsets between adjacent sub-DEMs at TPs and GCPs in the REMA DEMs of 2016–2017 before bias-correction (black arrows, tide and ice velocity compensated) and thereafter (red arrows).

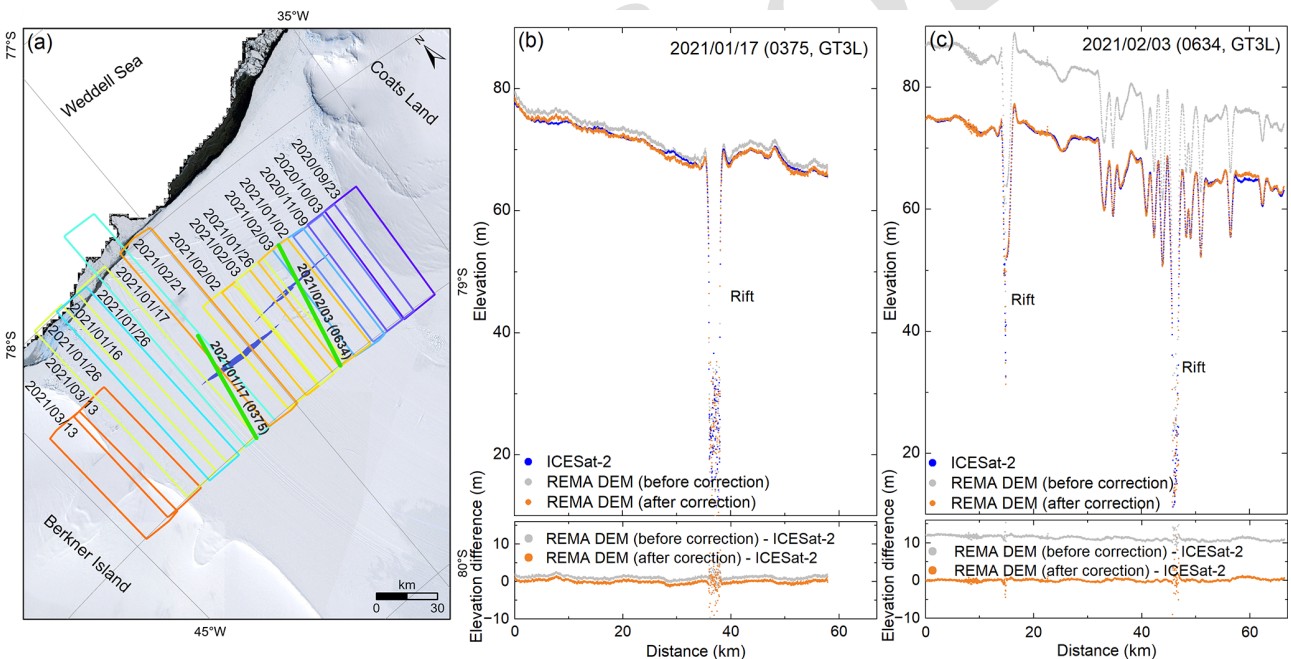

**Figure 8.** Verification of bias-corrected REMA DEMs of 2020–2021: **(a)** data acquisition times and footprints of REMA DEMs in different colors, and times and two ground tracks of ICESat-2 data in green color; **(b)** and **(c)** elevation profiles of REMA DEMs of 2020–2021 prior to (gray) and post (orange) bias-correction, and ICESat-2 ATL06 data of 2021 (blue), with their corresponding differences at bottom.

Figure 11a and c illustrate the average mélange top elevations inside T1 and T2 from the ZY-3 DEM of 2014, REMA DEMs of 2016–2017 and REMA DEMs of 2020–2021. Mélange top elevation profiles in the central sections of the rifts are relatively flat where distinct linear features, called Seracs in Neuburg et al. (1959) and Alley et al. (2023),

appear floating with a dominant orientation parallel to the rift centerline in both shaded relief map of DEM (Fig. 10a) and WorldView satellite image (Fig. 10b). We believe that the central sections of the rifts cut through the ice shelf and call the mélange "mature mélange". In contrast, the mélange in the sections close to both rift tips has relatively

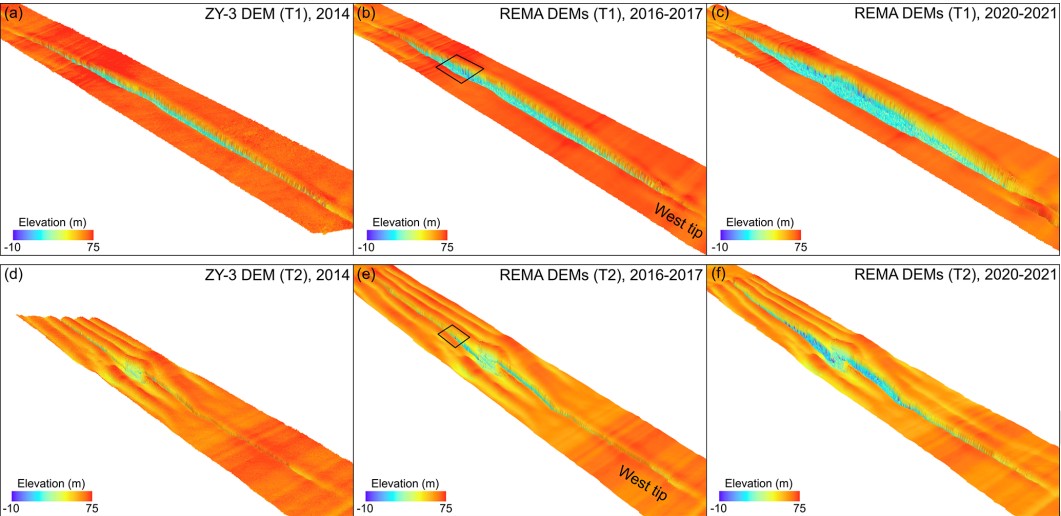

**Figure 9.** Multi-satellite DEM time series of two large rifts (T1 and T2) on Filchner Ice Shelf from 2014 to 2021 in 3D: **(a)** and **(d)** are reconstructed ZY-3 DEM of 2014, **(b)** and **(e)** are bias-corrected REMA DEMs of 2016–2017, and **(c)** and **(f)** are bias-corrected REMA DEMs of 2020–2021. The boxes in **(b)** and **(e)** indicate two rift sections where details of 3D structure and mélange are presented in Fig. 10.

rugged surface. More specifically, we found cavities that are freshly opened during the fracturing process in the unmatured mélange section close to the west tip of T1 (Fig. 11a). These pre-mélange cavities are verified in ZY-3 stereo images and reconstructed ZY-3 DEM (Fig. A3). They are initially as deep as 24.3 m in the ZY-3 DEM of 2014 (Fig. 11a). As the rift propagates in both length and width, new shelf ice in this section breaks apart from rift walls and fills in the cavities, resulting in a 5.7 m reduction in the cavity depth in the bias-corrected REMA DEM of 2016–2017 (Fig. 11a), and finally, disappearing during 2020–2021. During this process, we believe that the rift in this section cuts the ice shelf deeper and deeper, and the premature mélange becomes gradually mature.

The established DEM time series unveils an overall decrease of $1.8 \pm 0.6$ m, at a rate of $-0.4 \pm 0.1$ m yr$^{-1}$, in mélange elevation inside the two rifts from 2014 to 2021 ($3.1 \pm 0.4$ and $0.4 \pm 0.4$ m inside T1 (excluding cavity) and T2, respectively) (between blue and red lines in Fig. 11a and c). Correspondingly, the mélange thickness decreased by $10.0 \pm 3.3$ m ($1.6 \pm 0.4$ m yr$^{-1}$) during the period ($19.7 \pm 2.5$ and $2.8 \pm 2.2$ m for T1 and T2, respectively). This mélange thickness reduction observed during rift widening on Filchner Ice Shelf coincides with the earlier findings on Amery Ice shelf, Ronne Ice Shelf, and Larsen C Ice Shelf (Fricker et al., 2005; Walker et al., 2021; Larour et al., 2021). It may also indicate that the stress propagation between flanks may be compromised (Larour et al., 2004). More importantly, our results indicate that despite the significant decrease in mélange elevation from 2014 to 2021 (Fig. 11a and c), the inferred periodical mélange volumes reveal a rapid expansion of the mélange by $(10.15 \pm 0.03) \times 10^9$ km$^3$ (151 %) during the period, $(7.93 \pm 0.03) \times 10^9$ km$^3$ for T1

and $(2.22 \pm 0.01) \times 10^9$ km$^3$ for T2, respectively (Fig. 11), mainly attributed to newly vacated space due to partial collapse of rift flanks, associated rift widening, and other rift-mélange interaction factors.

## 4  Discussion

The developed multi-temporal DEM adjustment model MDAM has shown its effectiveness in removing biases between adjacent multi-satellite sub-DEMs and establishing a unified and integrated DEM time series. We demonstrate the full 3D mapping capability for characterizing rift structure (rift lips, pre-mélange cavities, mélange seracs, etc.) and estimating dynamic mélange volume changes from the ICESat-2 controlled DEM time series, extending from rift topography and mélange thickness estimated along ground tracks from ICESat and ICESat-2 measurements in previous studies (Fricker et al., 2005; Walker et al., 2021). Such high-resolution mélange dynamic observations allow us to understand the mélange movement inside a closed environment of large Antarctic rifts, and to further study its role in rapid rift propagation and iceberg calving. To make this MDAM model working in a more dynamic open ocean environment, such as Pine Island and Thwaites ice shelves where ice velocity is significantly higher, modifications need to be carried out to address the rapid calving process with incoherent mélange changes in ice shelf front. For example, tie points between adjacent sub DEMs may be selected with decreased time intervals to minimize the uncertainty caused by the compensation of horizontal displacement due to ice dynamics using velocity maps.

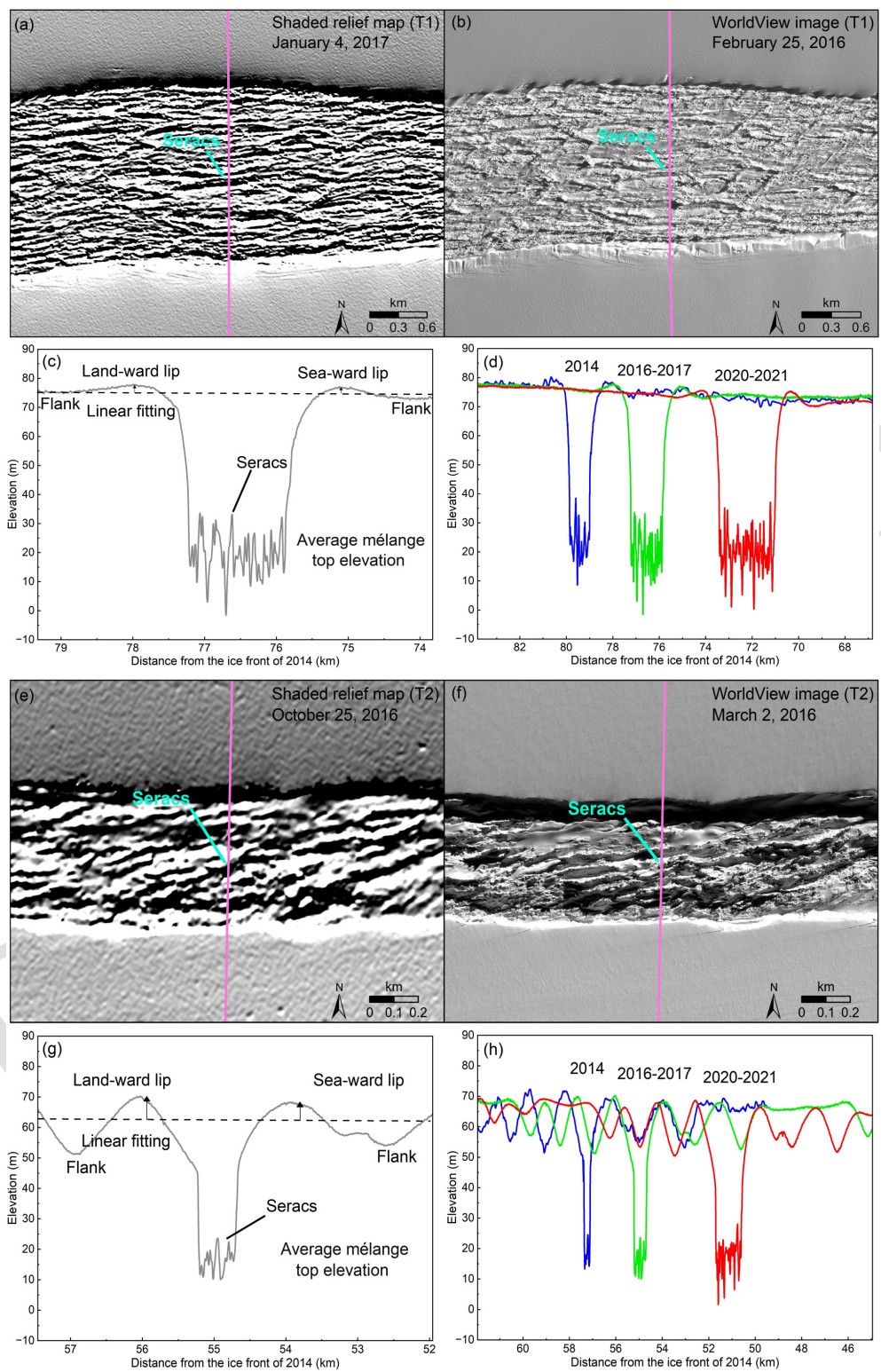

**Figure 10.** 3D sectional structure and mélange: **(a)** and **(e)** are sectional shaded relief maps of REMA DEMs of 2016–2017 (4 January 2017 for T1 and 25 October 2016 for T2) indicated by the boxes in Fig. 9; **(b)** and **(f)** are WorldView images (0.5 m resolution) of 25 February 2016 for T1 and 2 March 2016 for T2; **(c)** and **(g)** are elevation profiles along the pink lines in **(a)** and **(e)**, and 3D rift and mélange structure parameters; and **(d)** and (h) are rift and mélange changes along the profiles from 2014 to 2021 (2014 in blue, 2016–2017 in green, and 2020–2021 in red). Elevation displayed in **(a)** and **(e)** are exaggerated by 10 times.

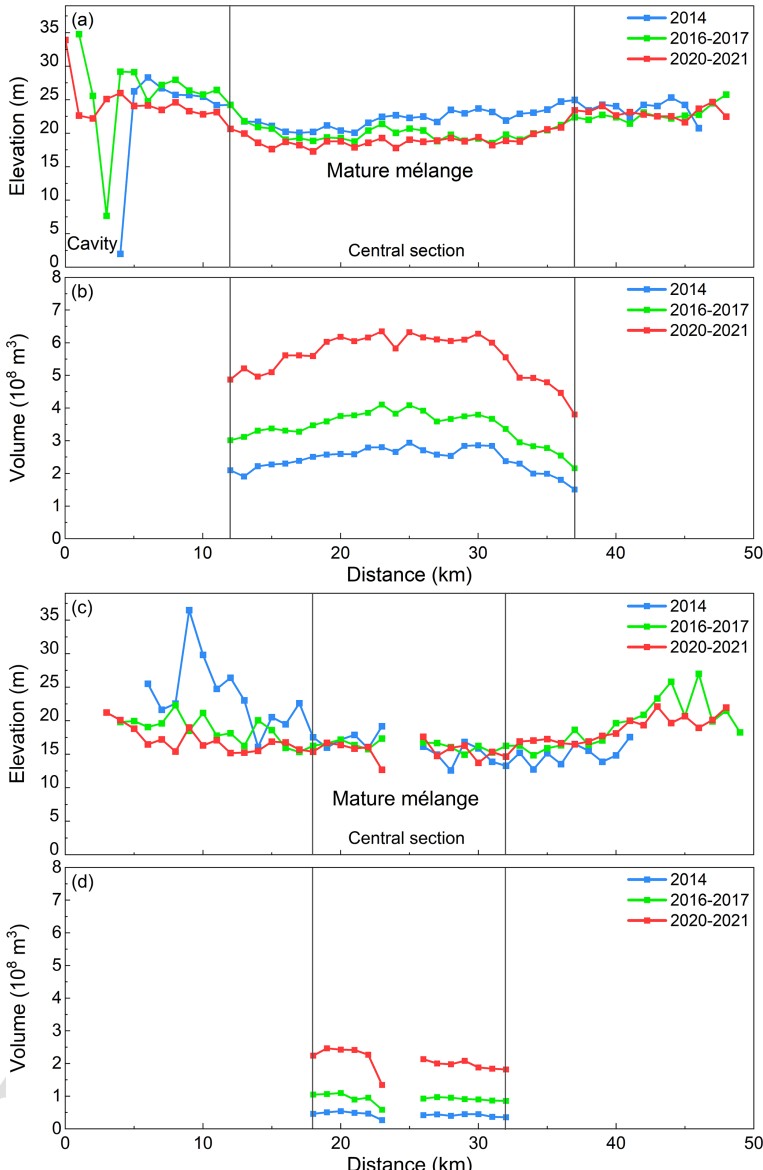

**Figure 11.** Elevation (**a** for T1 and c for T2) and volume (**b** for T1 and d for T2) of ice mélange inside the rifts from the multi-satellite DEM series from 2014 to 2021. Average elevation and volume of each transect that are separated every 1 km along the rift centerline are illustrated from the west tip to the east tip.

A thick mélange layer can effectively "freeze" a rift, enabling mechanical stress transmission between its flanks and ultimately suppressing rift propagation (Rignot and MacAyeal, 1998; Larour et al., 2021). In this study, we show 5 that the mélange thickness in T1 and T2 decreased as the rifts widened. Thus, the mélange layer may have thinned for a relatively long period, passing a possible stage of rift freeze. We further propose that the mélange production itself is related to the rift widening process. The seracs inside the rifts are 10 formed from partial collapsing of rift flanks. We suggest that the bottom part of the rift wall is first excavated through interactions by tides (Padman et al., 2002, 2008), melting caused

by intrusion of warmer sea water into the rift (Poinelli et al., 2023a), and other factors. This, in turn, causes the collapse of the upper part of the rift flank due to the removal of the bot- 15 tom support. This process, like coastal bluff erosion, repeats itself and becomes one of the mechanisms that widen the rift and increase the mélange volume. We find that the increased mélange volume in a rift promotes its widening rate.

Although the icebergs calved from the shelf front in 1986 20 are mostly located inside the passive shelf ice area (Fig. A1), meaning no significant buttress reduction from the ice shelf (Doake et al., 1998; Fürst et al., 2016), the combined rifts of T1 and T2 propagated rapidly recently and have already

covered ∼ 58 % of the ice shelf laterally from Coats Land to Berkner Island. They have the potential to cause a calving before reaching the passive shelf ice boundary (green line in Fig. A1). Furthermore, warm water observed near Berkner Island (Davis et al., 2022) may accelerate the propagation processes of T1 and T2 and destabilize the ice shelf through a mechanism like that proposed for the Larsen C Ice Shelf (Poinelli et al., 2023a, b).

## 5 Conclusions

We have developed an innovative multi-temporal DEM adjustment model (MDAM). This system successfully removed biases, as large as ∼ 6 m in elevation, between 50 sub-DEMs across the Filchner Ice Shelf in East Antarctica. The bias-corrected multi-satellite DEM time series achieved an accuracy of 0.1 m for the 2020–2021 REMA DEM assessed by ICESat-2 validation, and 0.2 m for the 2016–2017 REMA DEM and 0.3 m for the 2014 ZY-3 DEM estimated through the error propagation. Using this unified and integrated high-resolution DEM time series from 2014 to 2021, for the first time, we unveiled the 3D structural and mélange parameters of the ∼ 50 km long rifts of T1 and T2, including the rift lip height, pre-mélange cavity, and mélange elevation and thickness. Our findings indicate that while the mélange elevation decreased by $2.7 \pm 0.6$ m from 2014 to 2021, the mélange within the rifts experienced a rapid expansion by $(10.31 \pm 0.03) \times 10^9$ km³, or 139 %, during the period. This expansion is attributed to newly calved shelf ice from rift walls, associated rift widening, and other rift-mélange interaction factors. These new insights into 3D mélange dynamics are important for studying the mechanisms and the role of mélange in rift propagation.

Appendix A

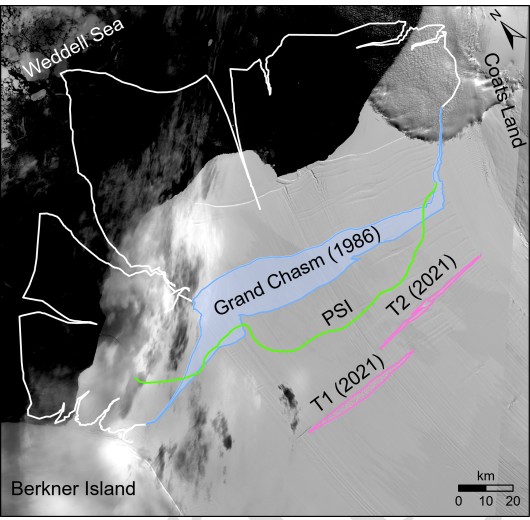

**Figure A1.** Rifts of T1 and T2 (pink) in 2021, Grand Chasm (blue) in 1986; passive shelf ice (PSI) boundary (green) (Fürst et al., 2016); ice bergs calved in 1986; background is Landsat image of 11 February 2021.

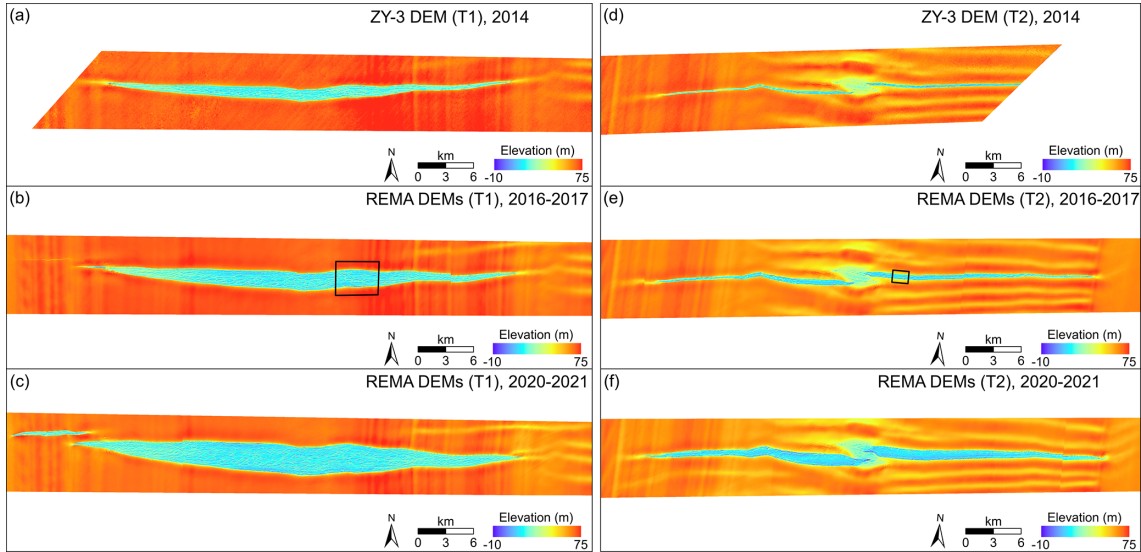

**Figure A2.** Multi-satellite DEM time series of the two large rifts (T1 and T2) on Filchner Ice Shelf from 2014 to 2021 in 2D: **(a)** and **(d)** are reconstructed ZY-3 DEM of 2014, **(b)** and **(e)** are bias-corrected REMA DEMs of 2016–2017, and **(c)** and **(f)** are bias-corrected REMA DEMs of 2020–2021. The boxes in **(b)** and **(e)** indicates two rift sections of the rift where details of 3D structure and mélange are presented in Fig. 10.

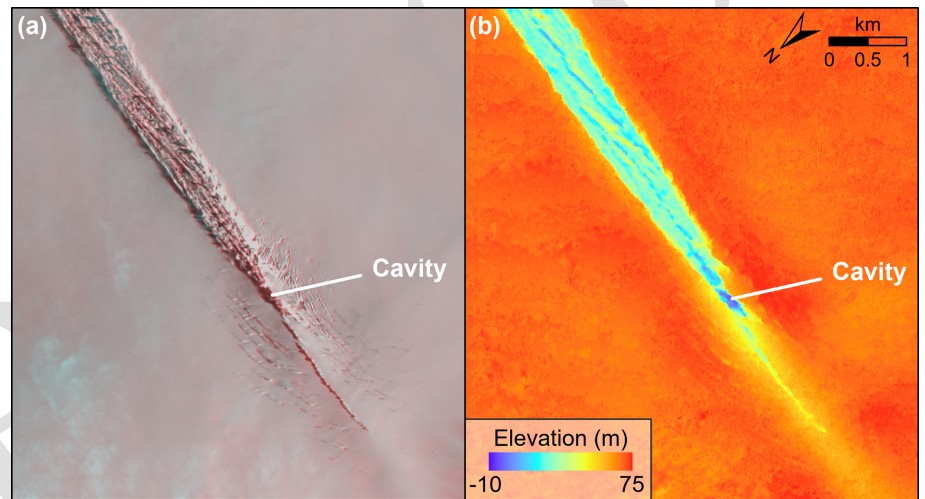

**Figure A3.** Anaglyph stereo pair of 2014 ZY-3 satellite images **(a)** and reconstructed ZY-3 DEM **(b)** showing cavities close to the west tip of rift T1.

**Table A1.** Information of images used for ZY-3 DEM reconstruction and REMA sub-DEMs (Howat et al., 2022).

| DEM type | DEM$^k$ | Time of image 1 | Time of image 2 | Time of image 3 |
|---|---|---|---|---|
| ZY-3 DEM | – | 2014/02/28 18:34:23 | 2014/02/28 18:34:54 | 2014/02/28 18:35:25 |
| **DEM type** | **DEM$^k$** | **ID of sub-DEM** | **Time of image 1** | **Time of image 2** |
| REMA DEMs (2016–2017) | 1 | SETSM_s2s041_WV01_ 20161031_10200100560C2900_ 1020010057847400_2m_lsf_seg1 | 2016/10/31 06:55:50 | 2016/10/31 06:54:45 |
| | 2 | SETSM_s2s041_WV03_ 20170104_10400100262E6300_ 10400100277D2F00_2m_lsf_seg1 | 2017/01/04 03:31:27 | 2017/01/04 03:30:17 |
| | 3 | SETSM_s2s041_WV01_ 20161219_1020010058C54700_ 1020010057729D00_2m_lsf_seg1 | 2016/12/19 13:55:22 | 2016/12/19 13:56:21 |
| | 4 | SETSM_s2s041_WV01_ 20161220_102001005AC45800_ 1020010058D62800_2m_lsf_seg1 | 2016/12/20 07:19:42 | 2016/12/20 07:20:36 |
| | 5 | SETSM_s2s041_WV02_ 20161006_103001005E9F9E00_ 103001005D8A5100_2m_lsf_seg1 | 2016/10/06 11:11:37 | 2016/10/06 11:10:09 |
| | 6 | SETSM_s2s041_WV02_ 20161219_10300100615A4D00_ 1030010062336300_2m_lsf_seg1 | 2016/12/19 08:59:46 | 2016/12/19 09:01:26 |
| | 7 | SETSM_s2s041_WV01_ 20161219_1020010057514C00_ 102001005A46ED00_2m_lsf_seg1 | 2016/12/19 13:56:00 | 2016/12/19 13:54:55 |
| | 8 | SETSM_s2s041_WV01_ 20170104_10200100594C1300_ 102001005ACFFC00_2m_lsf_seg1 | 2017/01/04 14:21:20 | 2017/01/04 14:20:11 |
| | 9 | SETSM_s2s041_WV01_ 20161025_1020010056B75500_ 102001005814FB00_2m_lsf_seg1 | 2016/10/25 07:33:35 | 2016/10/25 07:32:32 |
| | 10 | SETSM_s2s041_WV01_ 20161025_1020010055CA2A00_ 102001005658D200_2m_lsf_seg1 | 2016/10/25 07:33:55 | 2016/10/25 07:32:52 |
| | 11 | SETSM_s2s041_WV02_ 20161212_1030010061560200_ 10300100614D0D00_2m_lsf_seg1 | 2016/12/12 03:23:10 | 2016/12/12 03:22:00 |
| | 12 | SETSM_s2s041_WV02_ 20161220_10300100633CF200_ 1030010062317900_2m_lsf_seg1 | 2016/12/20 05:07:25 | 2016/12/20 05:05:43 |
| | 13 | SETSM_s2s041_WV02_ 20161004_103001005E897400_ 103001005E7A4B00_2m_lsf_seg1 | 2016/10/4 10:43:18 | 2016/10/4 10:45:21 |
| | 14 | SETSM_s2s041_WV01_ 20161031_1020010057D5DD00_ 1020010057C58900_2m_lsf_seg1 | 2016/10/31 06:55:20 | 2016/10/31 06:54:15 |
| REMA DEMs (2020–2021) | 1 | SETSM_s2s041_WV01_ 20210313_10200100A22EED00_ 10200100A41A1C00_2m_lsf_seg1 | 2021/03/13 14:29:27 | 2021/03/13 14:28:34 |
| | 2 | SETSM_s2s041_WV01_ 20210313_10200100A4D4E000_ 10200100A4DAC200_2m_lsf_seg1 | 2021/03/13 14:28:17 | 2021/03/13 14:29:08 |
| | 3 | SETSM_s2s041_WV01_ 20210126_10200100A3E1F100_ 10200100A6209000_2m_lsf_seg1 | 2021/01/26 14:01:59 | 2021/01/26 14:02:49 |
| | 4 | SETSM_s2s041_WV01_ 20210116_10200100A3347F00_ 10200100A6DCA900_2m_lsf_seg1 | 2021/01/16 06:48:32 | 2021/01/16 06:49:26 |
| | 5 | SETSM_s2s041_WV01_ 20210126_10200100A0336A00_ 10200100A3CAE800_2m_lsf_seg1 | 2021/01/26 14:01:37 | 2021/01/26 14:02:26 |
| | 6 | SETSM_s2s041_WV02_ 20210117_10300100B354E500_ 10300100B4CB5400_2m_lsf_seg1 | 2021/01/17 11:16:55 | 2021/01/17 11:18:08 |
| | 7 | SETSM_s2s041_WV02_ 20210221_10300100B2BA9B00_ 10300100B3853900_2m_lsf_seg1 | 2021/02/21 08:08:54 | 2021/02/21 08:07:50 |
| | 8 | SETSM_s2s041_WV01_ 20210202_10200100A0839200_ 10200100A67B9000_2m_lsf_seg1 | 2021/02/02 13:00:47 | 2021/02/02 13:01:35 |
| | 9 | SETSM_s2s041_WV02_ 20210203_10300100B33FB700_ 10300100B3535000_2m_lsf_seg1 | 2021/02/03 05:54:34 | 2021/02/03 05:53:18 |
| | 10 | SETSM_s2s041_WV02_ 20210126_10300100B37B2C00_ 10300100B4999A00_2m_lsf_seg1 | 2021/01/26 09:05:41 | 2021/01/26 09:06:49 |
| | 11 | SETSM_s2s041_WV02_ 20210203_10300100B409A800_ 10300100B4AEC000_2m_lsf_seg1 | 2021/02/03 10:51:17 | 2021/02/03 10:49:46 |
| | 12 | SETSM_s2s041_WV01_ 20210122_10200100A2B85700_ 10200100A3D3D700_2m_lsf_seg1 | 2021/01/22 07:42:53 | 2021/01/22 07:43:45 |
| | 13 | SETSM_s2s041_WV01_ 20201109_102001009EC56300_ 10200100A1685C00_2m_lsf_seg1 | 2020/11/09 06:48:36 | 2020/11/09 06:49:26 |
| | 14 | SETSM_s2s041_WV02_ 20201003_10300100AE149700_ 10300100AE656F00_2m_lsf_seg1 | 2020/10/03 09:45:56 | 2020/10/03 09:46:59 |
| | 15 | SETSM_s2s041_WV01_ 20200923_102001009B59CF00_ 102001009C2DFD00_2m_lsf_seg1 | 2020/09/23 13:06:20 | 2020/09/23 13:05:24 |

**Table A2.** Bias corrections ($dX^k$, $dY^k$, $dZ^k$) estimated for sub-DEMs of the REMA DEMs of 2016–2017 and the REMA DEMs of 2020–2021, ordered from Berkner Island to Coats Land.

| REMA DEMs of 2016–2017 | | |
|---|---|---|
| $DEM^k$ $\quad$ $dX^k \pm \sigma_{dX}$ (m) | $dY^k \pm \sigma_{dY}$ (m) | $dZ^k \pm \sigma_{dZ}$ (m) |
| 1 $\quad$ $4.6 \pm 1.4$ | $-4.2 \pm 1.5$ | $3.2 \pm 0.1$ |
| 2 $\quad$ $8.7 \pm 2.6$ | $-2.4 \pm 2.9$ | $-1.2 \pm 0.1$ |
| 3 $\quad$ $6.2 \pm 3.1$ | $-7.6 \pm 4.0$ | $-1.1 \pm 0.2$ |
| 4 $\quad$ $7.8 \pm 3.5$ | $-5.3 \pm 5.1$ | $5.9 \pm 0.2$ |
| 5 $\quad$ $-5.3 \pm 3.8$ | $-9.8 \pm 6.0$ | $-5.8 \pm 0.3$ |
| 6 $\quad$ $1.6 \pm 4.0$ | $2.3 \pm 6.7$ | $4.4 \pm 0.4$ |
| 7 $\quad$ $3.1 \pm 4.1$ | $0.2 \pm 7.2$ | $-0.3 \pm 0.4$ |
| 8 $\quad$ $-0.2 \pm 4.1$ | $1.1 \pm 7.1$ | $-1.1 \pm 0.4$ |
| 9 $\quad$ $-1.4 \pm 4.0$ | $10.0 \pm 6.7$ | $-2.0 \pm 0.3$ |
| 10 $\quad$ $-2.0 \pm 4.1$ | $11.6 \pm 6.0$ | $-2.3 \pm 0.3$ |
| 11 $\quad$ $-4.0 \pm 4.1$ | $13.7 \pm 5.1$ | $3.1 \pm 0.2$ |
| 12 $\quad$ $-0.8 \pm 3.8$ | $7.9 \pm 4.0$ | $2.7 \pm 0.2$ |
| 13 $\quad$ $0.7 \pm 3.6$ | $4.5 \pm 2.9$ | $-2.4 \pm 0.2$ |
| 14 $\quad$ $-1.9 \pm 3.2$ | $1.7 \pm 1.5$ | $-0.5 \pm 0.1$ |
| REMA DEMs of 2020–2021 | | |
| 1 $\quad$ $2.0 \pm 1.7$ | $-1.9 \pm 2.4$ | $-3.9 \pm 0.1$ |
| 2 $\quad$ $-4.6 \pm 1.7$ | $2.3 \pm 2.3$ | $-1.0 \pm 0.1$ |
| 3 $\quad$ $-0.2 \pm 1.7$ | $-0.8 \pm 2.4$ | $-2.9 \pm 0.2$ |
| 4 $\quad$ $4.7 \pm 2.2$ | $2.7 \pm 3.1$ | $0.7 \pm 0.2$ |
| 5 $\quad$ $1.9 \pm 1.8$ | $-3.0 \pm 2.5$ | $-0.2 \pm 0.2$ |
| 6 $\quad$ $6.8 \pm 2.2$ | $-5.2 \pm 3.0$ | $-1.2 \pm 0.2$ |
| 7 $\quad$ $-3.8 \pm 3.1$ | $6.0 \pm 4.2$ | $0.1 \pm 0.3$ |
| 8 $\quad$ $-1.2 \pm 3.3$ | $2.5 \pm 4.6$ | $-1.3 \pm 0.3$ |
| 9 $\quad$ $-3.6 \pm 2.9$ | $5.9 \pm 4.0$ | $2.1 \pm 0.2$ |
| 10 $\quad$ $5.3 \pm 2.2$ | $-6.2 \pm 3.0$ | $-0.7 \pm 0.2$ |
| 11 $\quad$ $0.6 \pm 1.7$ | $-0.3 \pm 2.3$ | $-11.0 \pm 0.1$ |
| 12 $\quad$ $-1.3 \pm 2.6$ | $3.2 \pm 3.6$ | $-1.7 \pm 0.2$ |
| 13 $\quad$ $-1.4 \pm 3.1$ | $5.3 \pm 4.2$ | $1.2 \pm 0.2$ |
| 14 $\quad$ $-0.6 \pm 2.5$ | $3.0 \pm 3.5$ | $2.2 \pm 0.2$ |
| 15 $\quad$ $0.6 \pm 1.5$ | $-1.6 \pm 2.1$ | $-1.8 \pm 0.1$ |

**Table A3.** Heights of rift lips measured on the land-ward (L) side and sea-ward (S) side, and their difference (S–L) at each transect along the centerlines of Rift T1 and T2 (REMA DEMs of 2016–2017).

| Rift T1: | | | |
| --- | --- | --- | --- |
| Transect ID from west tip to east tip (1 km separation) | Land-ward lip height (L) (m) | Sea-ward lip height (S) (m) | Difference (S–L) (m) |
| 0 | −1.7 | 0.5 | 2.2 |
| 1 | 3.6 | 4.5 | 1.0 |
| 2 | 2.7 | 4.7 | 2.0 |
| 3 | 0.8 | 1.7 | 0.9 |
| 4 | 4.4 | 1.5 | −2.9 |
| 5 | 0.7 | 1.7 | 1.0 |
| 6 | 1.8 | 2.0 | 0.2 |
| 7 | 2.3 | 2.6 | 0.3 |
| 8 | 1.3 | 1.4 | 0.2 |
| 9 | 0.1 | 1.7 | 1.7 |
| 10 | −1.6 | 1.8 | 3.4 |
| 11 | −0.7 | 1.1 | 1.7 |
| 12 | 1.6 | 3.4 | 1.8 |
| 13 | 2.8 | 3.0 | 0.2 |
| 14 | 1.6 | 2.6 | 1.0 |
| 15 | 1.6 | 2.3 | 0.8 |
| 16 | 1.8 | 2.1 | 0.3 |
| 17 | 1.5 | 1.9 | 0.5 |
| 18 | 2.5 | 2.2 | −0.3 |
| 19 | 1.6 | 1.9 | 0.2 |
| 20 | 1.9 | 1.8 | −0.1 |
| 21 | 2.3 | 1.6 | −0.7 |
| 22 | 2.0 | 1.4 | −0.6 |
| 23 | 1.7 | 0.9 | −0.7 |
| 24 | 1.7 | 1.0 | −0.7 |
| 25 | 0.9 | 1.0 | 0.1 |
| 26 | 1.9 | 0.3 | −1.6 |
| 27 | 1.0 | 0.1 | −0.9 |
| 28 | 0.7 | 1.5 | 0.8 |
| 29 | 2.4 | 3.4 | 1.0 |
| 30 | 1.4 | 2.1 | 0.7 |
| 31 | 3.3 | 3.7 | 0.5 |
| 32 | 3.3 | 3.9 | 0.6 |
| 33 | 3.7 | 2.1 | −1.6 |
| 34 | 3.7 | 4.8 | 1.1 |
| 35 | 2.0 | 1.9 | 0.0 |
| 36 | 0.4 | 1.2 | 0.8 |
| 37 | −0.3 | −2.3 | −2.0 |
| 38 | 2.8 | 3.5 | 0.7 |
| 39 | 2.5 | 1.3 | −1.2 |
| 40 | 2.4 | 1.7 | −0.6 |
| 41 | 2.0 | 1.1 | −0.9 |
| 42 | 0.8 | 1.5 | 0.7 |
| 43 | 2.6 | 2.6 | 0.0 |
| 44 | 4.6 | 4.2 | −0.4 |
| 45 | 3.1 | 2.7 | −0.5 |
| 46 | −0.8 | 0.1 | 1.0 |
| 47 | 1.8 | 1.9 | 0.1 |
| 48 | −1.4 | −1.8 | −0.4 |

Average lip height difference (s–l) within the section of mature mélange of T1: 0.1 m (max 1.8 m at 12 km from west tip).

Rift T2:

| Transect ID from west tip to east tip (1 km separation) | Land-ward lip height (L) (m) | Sea-ward lip height (S) (m) | Difference (S–L) (m) |
| --- | --- | --- | --- |
| 0 | 0.4 | 3.0 | 2.5 |
| 1 | 0.6 | 0.7 | 0.1 |
| 2 | −0.1 | 1.6 | 1.7 |
| 3 | 2.1 | 3.0 | 0.9 |
| 4 | 4.5 | 4.9 | 0.4 |
| 5 | 2.3 | 2.9 | 0.7 |
| 6 | 2.0 | 0.8 | −1.2 |
| 7 | 2.0 | 2.7 | 0.7 |
| 8 | −2.2 | −1.8 | 0.4 |
| 9 | −2.6 | −0.9 | 1.7 |
| 10 | 0.1 | 0.5 | 0.5 |
| 11 | −0.7 | 0.1 | 0.8 |
| 12 | 0.9 | 1.2 | 0.3 |
| 13 | 0.3 | −0.9 | −1.3 |
| 14 | −0.3 | −3.2 | −2.9 |
| 15 | −0.3 | −2.4 | −2.2 |
| 16 | 0.3 | −0.9 | −1.3 |
| 17 | 0.2 | 1.2 | 0.9 |
| 18 | 0.1 | 3.2 | 3.0 |
| 19 | −1.6 | 4.6 | 6.1 |
| 20 | −11.5 | 1.4 | – |
| 21 | −7.8 | −11.5 | −3.7 |
| 22 | −2.8 | 3.0 | 5.9 |
| 23 | 0.6 | 2.7 | 2.2 |
| 24 | 1.2 | 3.6 | 2.5 |
| 25 | 0.7 | −16.7 | – |
| 26 | −18.6 | −5.24 | – |
| 27 | −0.3 | −1.9 | −1.6 |
| 28 | 2.5 | 0.0 | −2.5 |
| 29 | 1.8 | 1.6 | −0.1 |
| 30 | 1.1 | 1.1 | 0.1 |
| 31 | 1.2 | 1.1 | −0.1 |
| 32 | 0.6 | −0.1 | −0.7 |
| 33 | 1.9 | 1.5 | −0.4 |
| 34 | 3.3 | 2.1 | −1.2 |
| 35 | 2.9 | 2.8 | 0.0 |
| 36 | 3.6 | 3.5 | −0.1 |
| 37 | 4.7 | 4.4 | −0.3 |
| 38 | 3.9 | 3.3 | −0.6 |
| 39 | 4.3 | 3.5 | −0.8 |
| 40 | 3.2 | 2.7 | −0.6 |
| 41 | 3.0 | 2.0 | −1.0 |
| 42 | 3.4 | 1.4 | −2.0 |
| 43 | 3.6 | 1.7 | −1.9 |
| 44 | 3.5 | 2.0 | −1.5 |
| 45 | 3.5 | 1.7 | −1.9 |
| 46 | 4.3 | 1.6 | −2.7 |
| 47 | 5.3 | 1.7 | −3.6 |
| 48 | 6.1 | 4.7 | −1.4 |
| 49 | 5.0 | 6.2 | 1.2 |

Average lip height difference (s–l) within the section of mature mélange of T2: 0.2 m (max 5.9 m at 22 km from west tip).

*Data availability.* REMA DEMs used in this study are available at https://doi.org/10.7910/DVN/X7NDNY TS3 (Howat et al., 2019). ITS_LIVE ice flow velocity maps are at https://doi.org/10.5067/IMR9D3PEI28U TS4 (Gardner et al., 2019). TS5

*Author contributions.* RL led the study and designed the MDAM system. MX programmed and implemented the model. ZL performed accuracy verification. RL, MX, and MS were involved in data analysis and presentation. RL, MS, LA, and GQ edited the manuscript.

*Competing interests.* The contact author has declared that none of the authors has any competing interests.

ther geographical representation in this paper. While Copernicus Publications makes every effort to include appropriate place names, the final responsibility lies with the authors. Views expressed in the text are those of the authors and do not necessarily reflect the views of the publisher.

*Acknowledgements.* We thank the editor, Professor Bert Wouters, for his guidance and suggestions during the review process. We sincerely thank Dr. Ann-Sofie Priergaard Zinck and Dr. Mattia Poinelli for their constructive comments and suggestions, which greatly improved the manuscript. We thank the National Snow and Ice Data Center (NSIDC) for the REMA DEMs, the United States Geological Survey (USGS) for the Landsat images, and the Chinese Ministry of Natural Resources for the ZY-3 images. The WorldView images are purchased from DigitalGlobe.

*Financial support.* This TS6 research has been supported by the National Natural Science Foundation of China (grant no. 42394131) and the Ministry of Science and Technology of the People's Republic of China, National Key Research and Development Program of China (grant no. 2024YFF0808302).

*Review statement.* This paper was edited by Bert Wouters and reviewed by Ann-Sofie Priergaard Zinck and Mattia Poinelli.

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
