# Peer review of "Building multi-satellite DEM time series for insight into mélange inside large rifts in Antarctica"

_EGUsphere, 2025_

## Author Comment (AC1)

We appreciate the constructive comments from two reviewers (Dr. Ann-Sofie Priergaard Zinck, and Dr. Mattia Poinelli). Our manuscript will be much improved by their input. As informed in the letter of requesting responses, this file contains author comments (ACs). The revised manuscript should not be prepared and submitted at this stage. In the following responses, we use "**bold**" text for comments, "non-bold" text for our responses, and "italic" for changed text in the later improved manuscript.

**Referee #1**

**The authors present a new method for co-registering and temporally aligning REMA DEMs. This method is in itself quite interesting, and a good addition to existing methods which are more focused on ice shelf basal melting. The resulting DEMs are further applied to describe the 3D development of a rift and its mélange on the Filchner Ice Shelf. The 3D investigation reveals rapid expansion of the rift which is attributed to calved shelf ice from the walls within the rift. This study thus underscores the importance of high-resolution elevation models in understanding rift and mélange development.**

**As seen below I do have a few major comments related to this paper. First, I find the current structure of the Methods section confusing and difficult to follow and have made some suggestions on how to improve that. Secondly, the current manuscript gives false expectations of analysis of two rifts on the Filchner Ice Shelf, whereas only one is thoroughly studied. I, therefore, suggest performing the analysis to both rifts. Furthermore, the current manuscript is lacking a discussion of the method and results as well as a broader discussion of the impact of this study. Lastly, to make this new MDAM method valuable to the community I think that it is important that the code becomes publicly available.**

Response:

1) We are performing the suggested restructure of the manuscript, see the detailed description in the following responses to the Major Comments.

2) As suggested, we have added the results of T2 into the manuscript, see the detailed description in the following responses to the Major Comments.

3) Yes, we are adding a Discussion section to address limitations of the methods, mélange dynamics learned, and impact of the results, as suggested.

4) We will select a public hosting website to post our MDAM code, data of the Filchner project (DEMs, TPs, and GCPs).

**Major comments:**
**Structure of the Methods section:**

**The current structure of the Methods section makes it very difficult to follow the different steps as there are quite some jumps in the storyline. First, I miss a Data section in-between the Introduction and Methods sections. In the current manuscript, the data is briefly described in L135-155, which is after the presentation of the MDAM method. That location of the data description makes it very difficult to follow the MDAM method. It also breaks the flow of the methods as the reader must wait quite long from the first mentioning of the TPs and GCPs before their selection procedure is described. Secondly, there are Introduction elements present in the Methods (L120-129), which I suggest being moved to the Introduction. Finally, the Results section contains a lot of elements which I think belong in the Methods section (Sect. 3.1 and L289-298), as they contain information to the methods as opposed to presenting actual results.**

Response:

We agree with the suggestions. We are restructuring the manuscript accordingly.

First, as suggested, we move the data description to the location between Introduction and Methods. Some minor revisions are made to provide the context and make this move smoothly.

Secondly, the Introduction elements presented in the Methods (L120-129) are moved to the Instruction section, as suggested.

Finally, we moved those elements in the Results section (Sect. 3.1 and L289-298) that contain methods information to the Methods section.

**Rift T2:**

**You present both rifts T1 and T2 in the Methods and Fig. 3, but your analysis focuses solely on rift T1. I would, therefore, suggest that you include a similar analysis of T2 as the one you have made for T1. This will likely also strengthen your manuscript with regards to melange dynamics.**

Response:

The corrected DEMs and measurements for T2 are readily available. We will add figures and data of T2 (see some of them here) to make the analysis more complete, as suggested. We will complete the relevant analysis in the revised manuscript later.

[Figure]

**Figure 9. Multi-satellite DEM time series of the large rifts of T1 and T2 on Filchner Ice Shelf from 2014 to 2021: (a) and (d) reconstructed ZY-3 DEM of 2014, (b) and (e) bias-corrected REMA DEMs of 2016-2017, and (c) and (f) bias-corrected REMA DEMs of 2020-2021. The boxes in (b) and (e) indicate the corresponding sections of T1 and T2 where details of 3D structures and mélange are presented in Fig. 10.**

[Figure]

[Figure]

**Figure 10. 3D sectional structure and mélange in T1 (a – d) and T2 (e – h): (a) shaded relief map of a section of T1 REMA DEMs of 2016-2017 (January 4, 2017) indicated by the box in Fig. 9b; (b) WorldView image (0.5 m resolution) of February 25, 2016; (c) elevation profile along the pink line in (a) and (b), and 3D rift and mélange structure parameters; and (d) rift and mélange changes along the profile from 2014 to 2021 (February 28, 2014 in blue, January 4, 2017 in green, and February 2, 2021 in red). (e) shaded relief map of a section of T2 REMA DEMs of 2016-2017 (October 25, 2016) indicated by the box in Fig. 9e; (f) WorldView image (0.5 m resolution) of March 2, 2016; (g) elevation profile along the pink line in (e) and (f), and 3D rift and mélange structure parameters; and (h) rift and mélange changes along the profile from 2014 to 2021 (February 28, 2014 in blue, October 25, 2016 in green, and February 3, 2021 in red). Elevation displayed in (a) and (e) is exaggerated by 10 times.**

[Figure]

**Figure 11. Thickness (a and c) and volume (b and d) of ice mélange inside the rift T1 and T2, respectively, from the multi-satellite DEM series from 2014 to 2021. Average thickness and volume of each transect that are separated every 1 km along the rift centerlines are illustrated from west tip to east tip**

Table A4. Rift lip heights and differences measured for T1 and T2

T1

| Transect ID from west tip to east tip (1 km separation) | Land-ward lip height (L) (m) | Sea-ward lip height (S) (m) | Difference (S-L) (m) |
|---|---|---|---|
| 1 | -1.67 | 0.54 | 2.21 |
| 2 | 3.58 | 4.53 | 0.95 |
| 3 | 2.68 | 4.65 | 1.97 |
| 4 | 0.84 | 1.72 | 0.88 |
| 5 | 4.39 | 1.49 | -2.91 |
| 6 | 0.72 | 1.74 | 1.02 |
| 7 | 1.85 | 2.01 | 0.16 |
| 8 | 2.28 | 2.59 | 0.32 |
| 9 | 1.25 | 1.43 | 0.17 |
| 10 | 0.07 | 1.74 | 1.67 |
| 11 | -1.57 | 1.84 | 3.40 |
| 12 | -0.68 | 1.06 | 1.75 |
| 13 | 1.63 | 3.40 | 1.77 |
| 14 | 2.81 | 2.98 | 0.17 |
| 15 | 1.58 | 2.55 | 0.97 |
| 16 | 1.56 | 2.29 | 0.74 |
| 17 | 1.84 | 2.13 | 0.29 |
| 18 | 1.46 | 1.91 | 0.46 |
| 19 | 2.48 | 2.18 | -0.30 |
| 20 | 1.65 | 1.89 | 0.24 |
| 21 | 1.91 | 1.82 | -0.09 |
| 22 | 2.52 | 1.81 | -0.71 |
| 23 | 1.99 | 1.37 | -0.61 |
| 24 | 1.65 | 0.95 | -0.70 |
| 25 | 1.70 | 1.03 | -0.67 |
| 26 | 0.89 | 0.99 | 0.09 |
| 27 | 1.90 | 0.19 | -1.71 |
| 28 | 0.98 | 0.05 | -0.93 |
| 29 | 0.68 | 1.47 | 0.79 |
| 30 | 2.42 | 3.43 | 1.00 |
| 31 | 1.37 | 2.08 | 0.71 |
| 32 | 3.26 | 3.74 | 0.47 |
| 33 | 3.32 | 3.90 | 0.58 |
| 34 | 3.69 | 2.08 | -1.61 |
| 35 | 3.71 | 4.80 | 1.09 |
| 36 | 1.95 | 1.90 | -0.05 |
| 37 | 0.42 | 1.21 | 0.79 |
| 38 | -0.27 | -2.25 | -1.98 |
| 39 | 2.77 | 3.48 | 0.71 |

| | | | |
|---|---|---|---|
| 40 | 2.52 | 1.35 | -1.18 |
| 41 | 2.39 | 1.64 | -0.74 |
| 42 | 2.02 | 1.10 | -0.92 |
| 43 | 0.81 | 1.48 | 0.67 |
| 44 | 2.64 | 2.60 | -0.05 |
| 45 | 4.62 | 4.19 | -0.43 |
| 46 | 3.14 | 2.66 | -0.48 |
| 47 | -0.84 | 0.13 | 0.97 |
| 48 | 1.79 | 1.86 | 0.07 |
| 49 | -1.40 | -1.75 | -0.35 |

Average rift lip height difference (seaward - landward) within the section of the T1 mature mélange: Mean: 0.03 m; Max: 1.77 m (13 km from the west tip)

T2:

| Transect ID from west tip to east tip (1 km separation) | Land-ward lip height (L) (m) | Sea-ward lip height (S) (m) | Difference (S-L) (m) |
|---|---|---|---|
| 1 | 0.43 | 2.98 | 2.55 |
| 2 | 0.56 | 0.67 | 0.11 |
| 3 | -0.06 | 1.64 | 1.70 |
| 4 | 2.14 | 3.01 | 0.87 |
| 5 | 4.49 | 4.87 | 0.38 |
| 6 | 2.25 | 2.91 | 0.66 |
| 7 | 1.96 | 0.77 | -1.19 |
| 8 | 2.04 | 2.69 | 0.65 |
| 9 | -2.21 | -1.81 | 0.40 |
| 10 | -2.63 | -0.92 | 1.71 |
| 11 | 0.06 | 0.54 | 0.48 |
| 12 | -0.70 | 0.14 | 0.84 |
| 13 | 0.95 | 1.25 | 0.30 |
| 14 | 0.32 | -0.94 | -1.26 |
| 15 | -0.33 | -3.21 | -2.88 |
| 16 | -0.26 | -2.41 | -2.15 |
| 17 | 0.33 | -0.94 | -1.27 |
| 18 | 0.23 | 1.18 | 0.94 |
| 19 | 0.14 | 3.16 | 3.03 |
| 20 | -1.55 | 4.58 | 6.13 |
| 21 | -11.54 | 1.37 | 12.91 |
| 22 | -7.76 | -11.51 | -3.75 |
| 23 | -2.83 | 3.04 | 5.87 |
| 24 | 0.57 | 2.72 | 2.15 |
| 25 | 1.16 | 3.65 | 2.49 |
| 26 | 0.73 | -16.66 | -17.39 |

| 27 | -18.62 | -5.24 | 13.38 |
| 28 | -0.31 | -1.95 | -1.64 |
| 29 | 2.49 | -0.03 | -2.52 |
| 30 | 1.75 | 1.64 | -0.11 |
| 31 | 1.06 | 1.14 | 0.08 |
| 32 | 1.18 | 1.07 | -0.11 |
| 33 | 0.62 | -0.07 | -0.69 |
| 34 | 1.89 | 1.45 | -0.44 |
| 35 | 3.25 | 2.06 | -1.19 |
| 36 | 2.86 | 2.84 | -0.01 |
| 37 | 3.65 | 3.54 | -0.11 |
| 38 | 4.66 | 4.38 | -0.28 |
| 39 | 3.90 | 3.34 | -0.56 |
| 40 | 4.34 | 3.54 | -0.81 |
| 41 | 3.22 | 2.65 | -0.56 |
| 42 | 3.01 | 2.05 | -0.96 |
| 43 | 3.36 | 1.38 | -1.98 |
| 44 | 3.58 | 1.67 | -1.91 |
| 45 | 3.52 | 2.00 | -1.53 |
| 46 | 3.53 | 1.65 | -1.88 |
| 47 | 4.33 | 1.63 | -2.70 |
| 48 | 5.32 | 1.68 | -3.64 |
| 49 | 6.09 | 4.68 | -1.42 |

Average rift lip height difference (seaward - landward) within the section of the T2 mature mélange: Mean: 0.6 m; Max: 13.38 m (27 km from the west tip)

**Discussion:**
**The manuscript in its current state does not contain any discussion of the results. I, therefore, miss a separate Discussion section which should at least include a discussion on the following topics:**
**1.**
**A broader discussion of your MDAM method, including a discussion of the limitations of the method.**
**2.**
**A broader discussion on what you have learned about mélange dynamics because of this study.**
**3.**
**A discussion on the broader impact of your results. E.g., how do these results aid in assessing ice shelf instability, which you for instance mention in the Introduction.**
Response:

Yes, we added a Discussion section to address these three points, along with those from Reviewer 2:

*"4 Discussion*

*The developed multi-temporal DEM adjustment model MDAM has shown its effectiveness in removing biases between adjacent multi-satellite sub-DEMs and establishing a unified and integrated DEM time series with an average elevation uncertainty of better than 0.24 m. We demonstrate the full 3D mapping capability for characterizing rift structure (rift lips, pre-mélange cavities, mélange seracs, etc.) and estimating dynamic mélange volume changes from the ICESat-2 controlled DEM time series, extending from rift topography and mélange thickness estimated along ground tracks from ICESat and ICESat-2 measurements in previous studies (Fricker et al., 2005; Walker et al., 2021). Such high-resolution mélange dynamic observations allow us to understand the mélange movement inside a closed environment of large Antarctic rifts, and to further study its role in rapid rift propagation and iceberg calving. To make this model working in a more dynamic open ocean environment, such as Pine Island and Thwaites ice shelves, modifications need to be carried out to address the rapid calving process with incoherent mélange changes in ice shelf front.*

*A thick mélange layer can effectively "freeze" a rift, enabling mechanical stress transmission between its flanks and ultimately suppressing rift propagation (Rignot, 1998; Larour, 2021). In this study, we show that seracs appearing on the mélange surface are a part of the infill inside rifts. They are formed from partial collapsing of rift flanks. We propose that the surface part of the bottom rift wall is first excavated through interactions by tides (Padman et al., 2002 and 2008), melting caused by intrusion of warmer sea water into the rift (Poinelli 2023a), and other factors. This, in turn, causes the collapse of the upper part of the rift flank due to the removal of the bottom support. This process, like coastal bluff erosion, repeats itself and becomes one of the mechanisms that widen the rift and increase the mélange volume. We find that the increased mélange volume in a rift promotes its widening rate, and may further increase ice front calving and effect the ice shelf stability.*

*Future calving triggered by T1 and T2 and the resulting ice front retreat may expose the ice shelf to increased warm water intrusion, a process simulated for the Larsen C ice shelf and supported by modelling results (Poinelli, 2023a and 2023b). Although the iceberg calved from the shelf front in 1986 is located inside the Passive Shelf Ice (PSI) area, meaning no actual buttress reduction for support from the ice shelf to the ice sheet (Doake, 1998; Fürst et al., 2016), the combined rifts of T1 and T2 propagated rapidly recently to cover ~58% the ice shelf laterally. In the context of global warming, this is particularly relevant for the Filchner Ice Shelf, as warm water has recently been observed near Berkner Island (e.g., Davis, 2022). Therefore, T1 and T2 have the potential to trigger a larger calving beyond the PSI boundary and could ultimately lead to destabilization mechanisms like those proposed for the Larsen C Ice Shelf."*

**Code and data availability:**
**A major part of this manuscript is the development of the MDAM method alongside with the resulting DEMs. I, therefore, think that it is necessary that the MDAM code (incl. TPs**

and GCPs) becomes publicly available on e.g., GitHub to be a truly addition to the cryospheric community. Other similar methods (Shean et al. 2019 and Zinck et al. 2023) are likewise publicly available. Furthermore, it would be desirable to make the produced DEMs publicly available online.

Response:

The MDAM code (including TPs and GCPs) is now available at GitHub (https://github.com/menglianxia/MDAM).

The produced DEMs (including adjusted REMA DEMs of 2016-2017 and 2020-2021, and ZY-3 DEM of 2014) are available at https://doi.org/10.5281/zenodo.15260323 (Xia et al. 2025).

Reference:

Xia, M., Li, R., Scaioni, M., An, L., Li, Z., and Qiao, G.: Dataset belonging to the article: Building multi-satellite DEM time series for insight into mélange inside large rifts in Antarctica, Zenodo, https://doi.org/10.5281/zenodo.15260323, 2025.

**Specific comments:**

**L12-15: The first two sentences of the abstract discuss "front calving", whereas this sentence discusses rift structural changes and mélange dynamics. I would suggest adding an extra sentence before this sentence which elaborates on mélange dynamics and how it is hypothesized to be related to front calving.**

Response:

We now add a sentence: "*Mélange dynamics inside rifts is recognized to potentially influence the rift propagation and subsequent iceberg calving.*"

**L14-17: Long and complex sentence. Consider splitting into two for clarity.**

Response:

The sentence is split into two sentences: "*We propose an innovative multi-temporal digital elevation model (DEM) adjustment model (MDAM) to build a multi-satellite DEM time series from meter-level resolution small DEMs. It removes biases across large Antarctic ice shelves, as large as ~6 m in elevation, caused by tides, ice flow dynamics, and observation errors.*"

**L14-15: Remember to explain acronyms. "We propose an innovative multi-temporal DEM…" → "We propose an innovative multi-temporal digital elevation model (DEM)…"**

Response:

We changed it accordingly.

**L17: Remember to explain acronyms. "Using 30 REMA…" → "Using 30 Reference Elevation Model of Antarctica (REMA)…"**

Response:

We changed it accordingly.

**L18: Consider changing to ", the second largest ice shelf in Antarctica." or remove entirely.**
Response:
We changed it accordingly.

**L24: You only use the acronym GSLR in this line and in L29, so I would write it out instead.**
Response:
Thanks. The suggestion is well taken.

**L26-27: Should be "The Antarctic Ice Sheet (AIS) has shown a persistent pattern of mass loss and has contributed to global sea level rise (GSLR) since the beginning of the satellite earth observation era in the 1960s (…"**
Response:
We changed it accordingly.

**L28: I don't understand the use of "Although" in the beginning of this sentence. I would rephrase this sentence with a focus on the importance of ice shelf calving instead of a focus on the grounding line.**
Response:
We rephrased it to: "*The lost ice mass enters the Southern Ocean from ice shelves mainly through two processes, namely shelf front calving and basal melting, each accounting for ~50% (Depoorter et al., 2013; Liu et al., 2015; Smith et al., 2020; Greene et al., 2022)*".

**L37: "advect to shelf front" → "advect to the shelf front"**
Response:
It is so changed.

**L38-41: I miss a clear definition of what a melange is.**
Response:
The sentence is changed to: "*Furthermore, the importance and challenges on studying mélange (a mixture of shelf ice, snow, sea ice and water) changes in relation to fractures in both Greenland and Antarctica are fully recognized (Rignot and Macayeal, 1998; Larour et al., 2004 and 2021; Cassotto et al., 2021).*"

**L56-57: Both Shean et al. 2019 and Zinck et al. 2023 handle heterogenous offsets between individual DEMs by applying dynamic corrections such as tides and by displacing the**

**DEMs based on ice flow. So, I am not sure that I agree that there is a lack in methods. However, both Shean et al. 2019 and Zinck et al. 2023 do their analysis with a focus on basal melting, and not with a focus on deriving elevation maps alone. I would, therefore, rephrase this sentence and make the focus on rifts and mélange dynamics stronger.**

Response:

Agree. It is changed to: "*There is a lack of methods for handling heterogenous offsets between individual DEMs caused by rifts, mélange dynamics, and other factors in a dynamic Antarctic ice shelf environment.*"

**L73: "the second largest in Antarctica" → "the second largest ice shelf in Antarctica"**

Response:

Changed accordingly.

**L74: "time series of 2014-" → "time series from 2014-"**

Response:

Changed accordingly.

**L74: "MDAM for quantitatively" → "MDAM by quantitatively"**

Response:

Changed accordingly.

**L83-89: You mention that you correct the DEMs for tides, but do you also correct the DEMs for the inverse barometer effect? And if not, what impact do you believe that to have on your results?**

Response:

We did not apply inverse barometer effect (IBE) corrections to the DEMs explicitly. We performed the following analysis to show that this does not have an impact on our results.

We use the fifth-generation ECMWF reanalysis (ERA5), with an hourly interval and a spatial resolution of $0.25° \times 0.25°$, to calculate the IBE corrections. Based on the mean sea level pressure from 2014 to 2021, we computed the sea level pressure anomalies at the observation times for each DEM, and subsequently converted them into elevation changes using a rate of 1 cm/hPa (Padman et al., 2003; Chen et al., 2023).

The IBE corrections (influence on surface elevation) for all REMA DEMs of 2016–2017 on the ice shelf vary between –13.2 cm and 10.9 cm, those for all REMA DEMs of 2020–2021 on the ice shelf vary between –4.9 cm and 29.7 cm. We further did an experiment to find the IBE correction variation within smaller areas, namely within areas of individual DEMs. We found that the IBE correction changes little within a smaller DEM extent, with standard deviations up to 7 mm. That means that within the area of a DEM, the IBE correction can be treated as a constant.

On the other hand, in our MDAM model, elevation bias at TPs of each DEM is removed by a linear adjustment formula $a_o^k + a_1^k Y_{OL_i}^k$ where the IBE correction, now treated as a constant, is combined with the constant term $a_o^k$ and corrected implicitly by the least-squares solution. The following is a numerical proof. For each DEM, we calculate the bias correction dz with and without the IBE correction.

REMA DEMs of 2016-2017:

| DEM_ID | dz (no IBE correction) (m) | dz (with IBE correction) (m) | d_dz (m) |
|--------|----------------------------|------------------------------|----------|
| 1 | 3.1462 ±0.0715 | 3.1464 ±0.0714 | -0.0002 |
| 2 | -1.1511 ±0.1325 | -1.1499 ±0.1325 | -0.0012 |
| 3 | -1.0808 ±0.1775 | -1.0785 ±0.1775 | -0.0023 |
| 4 | 5.9135 ±0.2356 | 5.9168 ±0.2356 | -0.0033 |
| 5 | -5.7584 ±0.2938 | -5.7547 ±0.2938 | -0.0037 |
| 6 | 4.3821 ±0.3558 | 4.3853 ±0.3557 | -0.0032 |
| 7 | -0.2665 ±0.4265 | -0.2648 ±0.4265 | -0.0017 |
| 8 | -1.0733 ±0.3817 | -1.0731 ±0.3817 | -0.0002 |
| 9 | -1.9812 ±0.3268 | -1.9848 ±0.3268 | 0.0035 |
| 10 | -2.3278 ±0.2979 | -2.3346 ±0.2979 | 0.0068 |
| 11 | 3.1072 ±0.2351 | 3.0997 ±0.2351 | 0.0075 |
| 12 | 2.6538 ±0.1933 | 2.6482 ±0.1933 | 0.0056 |
| 13 | -2.3968 ±0.1619 | -2.3994 ±0.1619 | 0.0026 |
| 14 | -0.4772 ±0.0898 | -0.4775 ±0.0898 | 0.0003 |

REMA DEMs of 2020-2021:

| DEM_ID | dz (no IBE correction) (m) | dz (with IBE correction) (m) | d_dz (m) |
|--------|----------------------------|------------------------------|----------|
| 1 | -3.9234 ±0.1294 | -3.9241 ±0.1293 | 0.0007 |
| 2 | -1.0231 ±0.1379 | -1.0231 ±0.1378 | -0.0001 |
| 3 | -2.8520 ±0.1477 | -2.8506 ±0.1476 | -0.0015 |
| 4 | 0.6495 ±0.1943 | 0.6503 ±0.1943 | -0.0007 |
| 5 | -0.2422 ±0.1684 | -0.2419 ±0.1683 | -0.0003 |
| 6 | -1.2167 ±0.1784 | -1.2169 ±0.1784 | 0.0002 |
| 7 | 0.1385 ±0.2629 | 0.1380 ±0.2628 | 0.0005 |
| 8 | -1.3039 ±0.2795 | -1.3046 ±0.2794 | 0.0007 |
| 9 | 2.0558 ±0.2143 | 2.0545 ±0.2143 | 0.0012 |
| 10 | -0.7338 ±0.1628 | -0.7343 ±0.1628 | 0.0004 |
| 11 | -11.0123 ±0.1256 | -11.0128 ±0.1256 | 0.0005 |
| 12 | -1.7191 ±0.2013 | -1.7187 ±0.2012 | -0.0004 |
| 13 | 1.1548 ±0.2385 | 1.1537 ±0.2385 | 0.0011 |
| 14 | 2.1978 ±0.1969 | 2.1983 ±0.1968 | -0.0005 |
| 15 | -1.8438 ±0.1122 | -1.8434 ±0.1122 | -0.0004 |

The average difference between the resulting dz with and without the IBE corrections is $0.08 \pm 0.47$ cm (maximum 1.3 cm) for REMA DEMs of 2016–2017. The corresponding value for REMA DEMs of 2020–2021 is $-0.01 \pm 0.10$ cm (maximum 0.27 cm). Similarly, for the ZY-3 DEM of 2014, the average difference does not exceed 2 cm.

Therefore, both methodological analysis and numerical experiments show that in our special case of using MDAM model to correct elevation bias, the IBE corrections can be treated as a constant within a small extent of the DEMs and can be taken care of by the linear correction model within the MDAM model. No explicit IBE corrections are needed here and no impact on the accuracy is found.

Reference:

Chen, H., Rignot, E., Scheuchl, B., and Ehrenfeucht, S.: Grounding Zone of Amery Ice Shelf, Antarctica, From Differential Synthetic-Aperture Radar Interferometry, Geophys. Res. Lett., 50, e2022GL102430, 10.1029/2022GL102430, 2023.

Padman, L., King, M., Goring, D., Corr, H., and Coleman, R.: Ice-shelf elevation changes due to atmospheric pressure variations, J. Glaciol., 49, 521-526, 10.3189/172756503781830386, 2003.

Hersbach, H., Bell, B., Berrisford, P., Biavati, G., Horányi, A., Muñoz Sabater, J., Nicolas, J., Peubey, C., Radu, R., Rozum, I., Schepers, D., Simmons, A., Soci, C., Dee, D., Thépaut, J-N., ERA5 hourly data on single levels from 1940 to present. Copernicus Climate Change Service (C3S) Climate Data Store (CDS), 10.24381/cds.adbb2d47 (Accessed on April 22, 2025), 2023.

**L85: In case you are not aware of it, there is an updated version to the CATS2008 tide model (CATS2008_v2023). I do not think that it is necessary to change tide model for this paper, but it might be worth changing in potential future paper as the updated tide model also has an updated grounding line.**
Response:

Thanks for the information. We will use the new version in future publications.

**L87: When you mention the TPs here, I would make sure to refer to the later section where you elaborate on how these TPs are selected. I would also do the same for the GCPs.**
Response:

In this paragraph we added text "*(see Section 2.2.2 for details)*" in places mentioning TPs and GCPs.

**L89-92: I find this sentence quite complicated and difficult to follow. Consider splitting it into two, first one about the residuals (so the second part of the sentence). And secondly a sentence on the least-squares adjustment method.**
Response:

We split it into two sentences: "*In the model inconsistencies at TP and GCP pairs are first reduced by the tide and velocity corrections, leaving the uncertainty originated from the sub-DEM production as residuals, including photogrammetric measurement errors and ephemeris data errors. Thereafter, we use the least-squares adjustment method (McGlone et al., 2004) to estimate the unknown bias corrections of all sub-DEMs by minimizing the residuals, (Li, 1998; McGlone et al., 2004; Shean et al., 2019).*"

**L103: "given in a velocity map": What velocity map do you refer to here? In L88 you mention that you use the ITS_LIVE velocities (which could mean both the average product or image-pairs) and in Figure 1 you mention that you use the ITS_LIVE image-pair velocity maps. So, what ITS_LIVE velocity product do you use? And how does that relate to the acquisition time of the various sub-DEMs?**
Response:

We added a sentence to explain it: "*Here we use averaged annual velocity maps of ITS_LIVE to avoid lower quality image-pair velocity maps potentially caused by low quality images and very short timespans between the pairs (Li et al., 2022).*"

**L104: Could you give an example of how these epsilons could be interpreted? Do they represent the error?**
Response:

The sentence is changed to: "($\varepsilon_{x_i}^{k}$, $\varepsilon_{y_i}^{k}$, $\varepsilon_{z_i}^{k}$) *represent residuals at TPs in the observation equations, including photogrammetric measurement errors and ephemeris data errors.*"

**L105-108: I miss an explanation of what the GCP epsilons represent.**
Response:

We added a sentence: "($\varepsilon_{x_i}^{GCP,k}$, $\varepsilon_{y_i}^{GCP,k}$, $\varepsilon_{z_i}^{GCP,k}$) *represent residuals at GCPs in the observation equations, describing inconsistencies between the integrated DEM and the outside control data (e.g., ICESat-2).*"

**L108-112: You mention tilting in the y-direction and how you deal with that. But how do you deal with tilting in the x-direction?**
Response:

This "tilting" in the Y-direction (ice flow direction) is dealt with by using the linear fitting method explained in the text (L108-112). We believe it is caused by advection of TPs and uneven ice shelf thickness (thicker from grounding line and thinner to shelf front). Since TPs "move" along the Y-direction, there is little "tilting" in the X-direction (cross shelf).

**L116-117: If the "bias correction parameters" mentioned here correspond to the epsilons in eq. 1 and 2 then I would state that here. E.g. "Uncertainties of the estimated bias correction parameters ($\varepsilon_{x_i}^k$, $\varepsilon_{y_i}^k$, $\varepsilon_{z_i}^k$, $\varepsilon_{x_i}^{GCP,k}$, $\varepsilon_{y_i}^{GCP,k}$, $\varepsilon_{z_i}^{GCP,k}$) are computed…".**

Response:

The sentence is changed to: "*Uncertainties of the estimated bias correction parameters ($\sigma_{x_i}^k$, $\sigma_{y_i}^k$, $\sigma_{z_i}^k$, $\sigma_{x_i}^{GCP,k}$, $\sigma_{y_i}^{GCP,k}$, $\sigma_{z_i}^{GCP,k}$) are computed through an error propagation within the optimization procedure.*"

**L120-129: I think that this part belongs in the Introduction as it nicely frames the need of studying rifts at a high resolution. I would suggest to incorporate it into the last paragraph of the Introduction (L65-73) and move some of the technical parts of the last paragraph to the Methods section.**

Response:

The paragraph is moved and revised accordingly.

**L125: "Recently, two large rifts, T1 and T2,…" → "Recently, two large rifts, T1 and T2 (Fig. 3),…"**

Response:

It is changed as suggested.

**L127: "To study their…" → "Studying their…"**

Response:

It is so changed.

**L135-140: I would rephrase some of these sentences here to make it clear that you use the REMA strips. The first time that I read it I was in doubt as to whether you used the REMA strips or if you processed the WorldView images yourself.**

Response:

It is now changed to make this clear: "*The sub-DEMs used in this study are REMA strips and a Ziyuan-3 (ZY-3) DEM which are generated from stereo satellite images of WorldView (Anderson and Marchisio, 2012) and ZY-3 (Wang et al., 2014), respectively. Both are formed by the along-track stereo mechanism (Li, 1998).*"

**L135-149: What is the motivation behind the three different time periods (ZY-3: 2014, REMA: 2016/17 and 2020/21) that you use? Is part of the goal to also compare ZY-3 with REMA? And why do you only use ZY-2 for year 2014 now that it has a better coverage of**

**the rifts? I miss a clearer justification of the study period and of the use of different satellite products.**

Response:

We added the following at the end of the paragraph: "*The ZY-3 DEM of 2014 has a full coverage of T1 and T2 without performing sub-DEM integration and can provide a reference for comparison with the adjusted sub-DEMs of REMA. These ZY-3 and REMA sub-DEMs form a seven-year long time series from 2014 to 2021 with a time interval of 3-4 years, within which shelf ice at any section would fall at least once from walls in to the rift so that elevation and volume changes can be calculated.*"

**L147-148: Why do you present rift T2 here in the Methods section and in Fig. 3 when you do not present any results from the rift? I would suggest that you add a similar analysis of T2 as you have done for T1. That could potentially also strengthen your manuscript with regards to melange dynamics.**

Response:

Please see the response to the relevant Major Comment.

**L150: You mention that you use ICESat-2 data from 2019-2021 to co-register the 2014 ZY-3 DEM. How can you justify this 5-7 years gap in between acquisition times?**

Response:

We added a sentence: "*To ensure that the GCPs are "stable" during the timespan between REMA and ICESat-2, these GCPs are further required to be on grounded features with a low velocity (< 10-20 m/y).*"

**L135-155: I would suggest that you make a "Data" section between the Introduction and the Methods section, and that you move all of this to the new Data section.**

Response:

Yes, we did it accordingly.

**L196-198: How do you use ICESat-2 to determine the GCP elevations for the REMA 2016/17 DEMs? Do you just assume that the elevations at those points are constant in time? And if so, does that assumption hold? And what impact does it have on the results?**

Response:

Please see the response above to "L150".

**L199: "on the flowing ice of the ice shelf,…" → "on the floating part of the ice shelf,…"**

Response:

It is so changed.

**L226-227: How is that co-registration performed How is ICESat-2 used given the large time difference?**

Response:

The sentence is changed to: "*The DEM is co-registered to the ICESat-2 ATL06 data of 2019 through a bundle adjustment procedure (McGlone et al., 2004; Li et al., 1998 and 2017b) using GCPs that are selected from "stable" features in the same way for those used in REMA sub-DEM co-registration. An elevation accuracy of 0.30 m is achieved.*"

**L233-242: As I understand the text here, you only validate the 2020/21 REMA DEM and not the ZY-3 2014 DEM and the 2016/17 REMA DEM. So how about those two other DEMs? How are they validated?**

Response:

Here validation is a process where we use "ground truth" (ICESat-2) to verify the accuracy of a set of DEMs. This external assessment requires that the ground truth data and DEMs to be in the same place and cover the same period. In this way we prove that the 2020-2021 REMA DEM is of a high accuracy, 0.09 m on surface and 0.18 m on mélange.

On the other hand, the ground truth data of ICESat-2 do not cover the same periods of the 2016-2017 REMA DEM and the 2014 ZY-3 DEM. However, all three DEM sets use the "stable" grounded features on Berkner Island to Coats Land as GCPs, so that the accuracy of the grounded part of the DEMs are thus validated in a manner of internal assessment (Fig. 7, 0.14 – 0.30 m).

As explained in the manuscript, we use an internal assessment of inconsistencies or residuals at the TPs to estimate the accuracy of the floating part of these two DEMs. The residuals are in average 0.14 m for the 2016-2017 REMA DEM (Fig. 7) and 0.30 m for the ZY-3 DEM as assessed by using the bundle adjustment. They are at the same level of 0.18 m, the internal accuracy assessment for the 2020-2021 REMA DEM.

Therefore, we trust that the 2016-2017 REMA DEM and the 2014 ZY-3 DEM have the same quality as the 2020-2021 REMA DEM.

**Sect. 3.1: In my opinion this section belongs in the Methods section and not in the Results.**

Response:

This section (3.1) reports the results and performance of the proposed MDAM model. Then Section 3.2 reports the mélange dynamics application results. To maintain the logic flow, we hope that you would not mind that we keep this section in the Results.

**L254-257: "… of 2014 where transects are spaced at…" → "… of 2014 with transects spaced at…"**

Response:

Changed accordingly.

**L264: Typo: I doubt that the rift is 4713.17 km long → it is probably 47.1317 km long.**
Response:
Thanks. It is corrected.

**L268-270: Interesting that the sea-ward side is mostly higher! Can you elaborate/speculate a bit on what the cause behind that could be?**
Response:
We found that the landward wall produces more seracs than the seaward wall. This may cut the higher part of the undulation on the landward flank, so the remaining surface gets lower. We need more measurements and modelling efforts to prove this speculation.

**L270-272: What does "The phenomenon" refer to in this sentence? Does it refer to the higher sea-ward side? And if so, what do they hypothesize as a reasoning for that?**
Response:
We changed the sentence: "*This phenomenon of the greater heights of seaward rift lips based on the precise measurements from the bias-corrected sub-DEMs is consistent with the results for some rifts on other Antarctic ice shelves presented in Walker et al. (2019).*" We leave the reasoning for future paper(s) (see response above)

**L277: Typo: YZ-3 should be ZY-3**
Response:
Done.

**L281: "We believe that this section of the rift…" → "We believe that this central section of the rift…"**
Response:
Done.

**L289-298: "To compute the corresponding… …in the mature melange section." Should be in the Methods section instead of in the results section.**
Response:
This part is moved to the Methods section.

**L305: I am not sure that I agree that the accuracy of all DEMs in general is 0.09 m as you only validate this for the 2020/21 DEMs.**
Response:

Agree. The sentence is changed to: "*The bias-corrected multi-satellite DEM time series achieved an accuracy of 0.09 m for the 2020-2021 REMA DEM assessed by ICESat-2 validation, and 0.18 m for the 2016-2017 REMA DEM to 0.3 m for the 2014 ZY-3 DEM, respectively, estimated through the error propagation.*"

**Figure 1: What small-coverage DEMs is it that you show on Figure 1a? Are those the REMA strips? I also miss an indication of the size of the strips relative to the ice shelf, so some sort of scale.**

Response:

We added the suggested scale information: "*Figure 1. (a) High-resolution small-coverage DEMs (e.g., REMA strips of ~16~18 km ✕ ~110~120 km) are unified and integrated for accurate 3D rift structural and mélange dynamic monitoring in a large Antarctic ice shelf environment (e.g., ~165 km wide Filchner Ice Shelf) ......*"

**Figure 3: How come there are GCPs on the ice shelf itself in panel b)? In the manuscript and Fig. 1 you mention that those are only on the grounded parts. Furthermore, the gery ZY-3 outlines in panel a) are very difficult to see, so I would choose a different color for those. I think that there is a typo in the colorbar of both panels, they say that the elevation go from -10 to -510. Finally, I would mark T1 and T2 in both panels.**

Response:

In the new Data section, we added: "...... *In a special case we also select GCPs on the ice shelf where both the DEM images and ICESat-2 data were acquired within one day.*" Similar sentence is also added in the caption of Fig. 3.

In the following revised Fig. 3, please see that the ZY-3 outlines are changed to darker and thicker lines. "-500 m" is changed to 500 m. T1 and T2 are also marked in (b).

[Figure]

**Figure 9: I would prefer to see these DEMs in 2D rather than 3D, as this 3D view does not allow for seeing the full depth of the rift. Secondly, I would consider adjusting the colorbar to make the ice shelf surface more visible without losing information from within the rift. Lastly, I miss a scale on the figures for reference, which will for instance make it easier to see by how much the rift has opened during the study period.**

Response:

We provide now both 2D DEMs and 3D DEMs (probably 3D in Appendix?).

[Figure]

**Figure 10: In panel c) and d) I would consider changing the x-axis to "Distance from the ice front" as that is an important parameter in terms of the calving risk.**

Response:

It is changed to "Distance from the ice front of 2014". We also added Figs. 10e-10h for T2.

[Figure]

**Figure A1: It is very difficult to see what is what on this figure. People that are not used to seeing stereo pairs plotted in this way will most likely not be able to tell that it is a cavity as opposed to a melange. Could you maybe show a stereo pair of a melange for comparison or add a colormap and colorbar which makes the cavity more visible?**

Response:

We revised the figure by adding a 2D color elevation map where the cavity is represented in blue color. Hope this makes the cavity visible. We will also revise the manuscript to make it clear.

[Figure]

**Data availability: How about the ZY-3 data? And will the DEM time series generated here also become publicly available somewhere?**

Response:

Yes, we will make the ZY-3 DEM publicly available, along with the corrected REMA DEMs.

**Code availability:**

**Will the MDAM code become publicly available? This paper is heavily based on the development of a new method, and I therefore think that it is important that that new method is also publicly available to the community.**

Response:

The MDAM code (including TPs and GCPs) is now available at GitHub (https://github.com/menglianxia/MDAM).

The produced DEMs (including adjusted REMA DEMs of 2016-2017 and 2020-2021, and ZY-3 DEM of 2014) are available at https://doi.org/10.5281/zenodo.15260323 (Xia et al. 2025).

Reference:

Xia, M., Li, R., Scaioni, M., An, L., Li, Z., and Qiao, G.: Dataset belonging to the article: Building multi-satellite DEM time series for insight into mélange inside large rifts in Antarctica, Zenodo, https://doi.org/10.5281/zenodo.15260323, 2025.

**References:**

**Shean, D. E., Joughin, I. R., Dutrieux, P., Smith, B. E., and Berthier, E.: Ice shelf basal melt rates from a high-resolution digital elevation model (DEM) record for Pine Island Glacier, Antarctica, The Cryosphere, 13, 2633–2656, https://doi.org/10.5194/tc-13-2633-2019, 2019.**

**Zinck, A.-S. P., Wouters, B., Lambert, E., and Lhermitte, S.: Unveiling spatial variability within the Dotson Melt Channel through high-resolution basal melt rates from the Reference Elevation Model of Antarctica, The Cryosphere, 17, 3785–3801, https://doi.org/10.5194/tc-17-3785-2023, 2023.**

Response:

These references are cited in right places of the manuscript.

---

## Author Comment (AC2)

We appreciate the constructive comments from two reviewers (Dr. Ann-Sofie Priergaard Zinck, and Dr. Mattia Poinelli). Our manuscript will be much improved by their input. As informed in the letter of requesting responses, this file contains author comments (ACs). The revised manuscript should not be prepared and submitted at this stage. In the following responses, we use "**bold**" text for comments, "non-bold" text for our responses, and "italic" for changed text in the later improved manuscript.

**Referee #2**

**Dear Menglian Xia and co-authors,**

**The manuscript "Building multi-satellite DEM time series for insight into mélange inside large rifts in Antarctica" by Xia et al. presents a novel approach to monitor rifts' infill on the Filchner-Ronne ice shelf using observations from different satellites at high resolution Understanding variations in rift infill is crucial for assessing rift dynamics, which ultimately influence calving events. I commend the authors for their creative approach, thorough analysis, and well-designed, highly informative figures. Overall, I strongly believe this paper would be a valuable addition to the literature, and its scope and format make it well-suited for publication in The Cryosphere. Below, I provide a general comment along with more specific, line-by-line suggestions that I hope will further strengthen the manuscript.**

**GENERAL COMMENT:**
**I admit that DEM processing is not my direct area of expertise, but the logic of the technical part seems sound and appropriate for a reader, like me, who is not necessarily familiar with technicalities of DEM analysis. However, I believe this manuscript would benefit from a few improvements to enhance readability and impact. In some sections, the text becomes somewhat difficult to follow, partly due to the frequent use of acronyms and the listing of numbers with excessive significant figures. At times, this made it challenging to stay engaged with the flow of the manuscript. I recommend streamlining certain parts, as indicated in my line-by-line comments, to improve clarity and coherence.**
Response:
Thanks for the suggestions. The manuscript is revised accordingly (see following responses).

**More importantly, I find that this manuscript lacks a thorough discussion, a comparison with existing literature, and a clear background motivation. I was surprised to see that it does not include a dedicated discussion section, where I had expected to find these elements. To strengthen the manuscript's impact, I recommend contextualizing the observational results within the broader framework of rift monitoring and their**

**implications for ice shelf and glacier dynamics, while also drawing comparisons with existing studies.**

Response:

We addressed the issues in two parts.

1) We added substantial text in Introduction to give a clear background motivation, comparison with existing studies, and context to this study … (also see following responses)

2) We added a Discussion section to address the remaining points:

*"4 Discussion*

*The developed multi-temporal DEM adjustment model MDAM has shown its effectiveness in removing biases between adjacent multi-satellite sub-DEMs and establishing a unified and integrated DEM time series with an average elevation uncertainty of better than 0.24 m. We demonstrate the full 3D mapping capability for characterizing rift structure (rift lips, pre-mélange cavities, mélange seracs, etc.) and estimating dynamic mélange volume changes from the ICESat-2 controlled DEM time series, extending from rift topography and mélange thickness estimated along ground tracks from ICESat and ICESat-2 measurements in previous studies (Fricker et al., 2005; Walker et al., 2021). Such high-resolution mélange dynamic observations allow us to understand the mélange movement inside a closed environment of large Antarctic rifts, and to further study its role in rapid rift propagation and iceberg calving. To make this model working in a more dynamic open ocean environment, such as Pine Island and Thwaites ice shelves, modifications need to be carried out to address the rapid calving process with incoherent mélange changes in ice shelf front.*

*A thick mélange layer can effectively "freeze" a rift, enabling mechanical stress transmission between its flanks and ultimately suppressing rift propagation (Rignot, 1998; Larour, 2021). In this study, we show that seracs appearing on the mélange surface are a part of the infill inside rifts. They are formed from partial collapsing of rift flanks. We propose that the surface part of the bottom rift wall is first excavated through interactions by tides (Padman et al., 2002 and 2008), melting caused by intrusion of warmer sea water into the rift (Poinelli 2023a), and other factors. This, in turn, causes the collapse of the upper part of the rift flank due to the removal of the bottom support. This process, like coastal bluff erosion, repeats itself and becomes one of the mechanisms that widen the rift and increase the mélange volume. We find that the increased mélange volume in a rift promotes its widening rate, and may further increase ice front calving and effect the ice shelf stability.*

*Future calving triggered by T1 and T2 and the resulting ice front retreat may expose the ice shelf to increased warm water intrusion, a process simulated for the Larsen C ice shelf and supported by modelling results (Poinelli, 2023a and 2023b). Although the iceberg calved from the shelf front in 1986 is located inside the Passive Shelf Ice (PSI) area, meaning no actual buttress reduction for support from the ice shelf to the ice sheet (Doake, 1998; Fürst et al., 2016), the combined rifts of T1 and T2 propagated rapidly recently to cover ~58% the ice shelf laterally. In the context of global warming, this is particularly relevant for the Filchner Ice Shelf, as warm water has recently been observed near Berkner Island (e.g., Davis, 2022). Therefore, T1 and T2 have the potential to trigger a larger calving beyond the PSI boundary*

*and could ultimately lead to destabilization mechanisms like those proposed for the Larsen C Ice Shelf."*

**While rifts and calving are natural processes in the life cycle of ice shelves, the calving of their seaward-most extensions does not necessarily lead to significant upstream glacier acceleration. I have not performed the calculations myself, but rifts T1 and T2 on the Filchner-Ronne Ice Shelf (FRIS) appear to lie beyond the 'compressive arch' (Doake, 1998), suggesting they may not directly precondition the ice shelf for collapse—similar to the 'passive portion' described by Fürst et al. (2016). This raises a key question: if these rifts are not an immediate destabilizing factor for FRIS, why is their study important?**

Response:

We added the following text in the Discussion: "…… *Although the iceberg calved from the shelf front in 1986 is located inside the Passive Shelf Ice (PSI) area, meaning no actual buttress reduction for support from the ice shelf to the ice sheet (Doake, 1998; Fürst et al., 2016), the combined rifts of T1 and T2 propagated rapidly recently to cover ~58% the ice shelf laterally. In the context of global warming, this is particularly relevant for the Filchner Ice Shelf, as warm water has recently been observed near Berkner Island (e.g., Davis, 2022). Therefore, T1 and T2 have the potential to trigger a larger calving beyond the PSI boundary and could ultimately lead to destabilization mechanisms like those proposed for the Larsen C Ice Shelf.*"

Also see the following figure.

[Figure]

**This is, of course, a provocative question, and I fully agree with the authors that rifts play a critical role. Future calving triggered by T1 and T2 and the resulting ice front retreat may expose the ice shelf to increased warm water intrusion, a process simulated for**

**Larsen C and supported by theoretical work (Poinelli, 2023a,b). In the context of global warming, this is particularly relevant for FRIS, as warm water has recently been observed near Berkner Island (e.g., Davis, 2022). Such changes could ultimately lead to destabilization mechanisms like those proposed for Larsen C. I encourage the authors to incorporate these considerations to provide a stronger and more comprehensive discussion of their findings.**

Response:

Thanks for suggestion. They are added into Discussion (see the response above).

**In the introduction, there is a strong focus on technical studies related to altimetric and stereo mapping of rifts, but the broader scientific goal and the significance of this region of Antarctica seem underdeveloped. For a paper in The Cryosphere, I would expect a more robust scientific background that clearly establishes the relevance and utility of this work.**

Response:

We added the following text in Introduction to address it.

*"…… Furthermore, mélange (a mixture of shelf ice, snow, sea ice and water) changes related to ice shelf fracturing in Antarctica and glacier calving in Greenland are studied (Rignot and Macayeal, 1998; Larour et al., 2004; Cassotto et al., 2021). Specifically, mélange thickness reduction is observed during rift widening on Amery Ice shelf, Ronne Ice Shelf and Larsen C Ice Shelf (Fricker et al., 2005; Walker et al., 2021; Larour et al., 2021). Modelling results indicate that mélange may have the capability for transmitting stresses across rift flanks and thus, influences rift propagation and shelf front calving (Larour et al., 2004 and 2021). However, there has been a lack of large-scale high-resolution observations of 3D rift structural changes and mélange dynamics in a sustained long period to support such a conclusion. This, in turn, hinders our understanding of the role of mélange in ice shelf retreating and the mechanism of ice shelf stability weakening."*

*"…… The proposed MDAM model is validated and applied in the Filchner Ice Shelf, one of the dual ice shelves of Filchner-Ronne Ice Shelf that is the second largest in Antarctica (Fig. 3). Berkner Island and Coats Land are located at western and eastern margins of the ice shelf, respectively. The largest calving event recorded for this ice shelf, with an area loss of 11500 km2, occurred in 1986 due to a rapid propagation of a prominent rift known as "Grand Chasm" (Ferrigno and Gould, 1987), producing three giant icebergs that greatly impacted the circulation and hydrography in the nearby ice shelf-ocean system (Grosfeld et al., 2001). Recently, two large rifts, T1 and T2 (Fig. 3), are detected in a Landsat satellite image in 1996 and an ARGON satellite image in 1963, respectively (Li et al., 2017; Walker and Gardner, 2019; Lv et al., 2022). These rifts have a combined length of ~100 km and exhibit a similar propagation pattern as Grand Chasm……"*

**One crucial aspect that appears to be missing is the role of ice mélange variation in modulating rift dynamics. A thick mélange layer can effectively "freeze" a rift, enabling mechanical stress transmission between its flanks and ultimately suppressing rift propagation (Rignot, 1998; Larour, 2021). This process is fundamental to understanding rift evolution and should be better integrated into the manuscript's scientific framing. I may have overlooked this point, and I apologize if it was addressed, but I encourage the authors to clarify and emphasize its importance.**

Response:

We added this sentence in Introduction: "*Modelling results indicate that mélange may have the capability for transmitting stresses across rift flanks and thus, influences rift propagation and shelf front calving (Larour et al., 2004 and 2021).*"

We further added the following text specifically in Discussion, but also make a link to the "volume" change results from this study:

"*A thick mélange layer can effectively "freeze" a rift, enabling mechanical stress transmission between its flanks and ultimately suppressing rift propagation (Rignot, 1998; Larour, 2021). In this study, we show that seracs appearing on the mélange surface are a part of the infill inside rifts. They are formed from partial collapsing of rift flanks. We propose that the surface part of the bottom rift wall is first excavated through interactions by tides (Padman et al., 2002 and 2008), melting caused by intrusion of warmer sea water into the rift (Poinelli 2023a), and other factors. This, in turn, causes the collapse of the upper part of the rift flank due to the removal of the bottom support. This process, like coastal bluff erosion, repeats itself and becomes one of the mechanisms that widen the rift and increase the mélange volume. We find that the increased mélange volume in a rift promotes its widening rate, and may further increase ice front calving and effect the ice shelf stability.*"

**SPECIFIC COMMENTS**

**Line 13: The connection between calving and ice mélange is not clearly established in the manuscript. Why is the observed "significant gap" in mélange dynamics particularly relevant to calving processes? Ice mélange, when sufficiently thick, can bond rift flanks together, suppressing rift widening and propagation. Clarifying this link would strengthen the manuscript's argument and better highlight the importance of studying mélange dynamics in the context of ice shelf stability.**

Response:

We added the following text in Introduction to clarify the link between mélange and calving:

"*…… Furthermore, mélange (a mixture of shelf ice, snow, sea ice and water) changes related to ice shelf fracturing in Antarctica and glacier calving in Greenland are studied (Rignot and Macayeal, 1998; Larour et al., 2004; Cassotto et al., 2021). Specifically, mélange thickness reduction is observed during rift widening on Amery Ice shelf, Ronne Ice Shelf and Larsen C*

*Ice Shelf (Fricker et al., 2005; Walker et al., 2021; Larour et al., 2021). Modelling results indicate that mélange may have the capability for transmitting stresses across rift flanks and thus, influences rift propagation and shelf front calving (Larour et al., 2004 and 2021). However, there has been a lack of large-scale high-resolution observations of 3D rift structural changes and mélange dynamics in a sustained long period to support such a conclusion. This, in turn, hinders our understanding of the role of mélange in ice shelf retreating and the mechanism of ice shelf stability weakening.*"

**Line 20: Mélange volume expansion is directly correlated with rift widening, but its relevance in the context of ice sheet modeling is unclear. Can we speculate on causality— does mélange volume drive rift widening, or is it simply a byproduct? Since mélange volume necessarily increases as the rift widens due to greater exposure of open ocean to surface heat loss, a more informative parameter to monitor might be mélange thickness relative to ice shelf thickness. This ratio is crucial in determining how stress is transmitted between rift flanks (Larour, 2021).**

Response:

Mélange thickness change is an important indicator. It is now reviewed in Introduction (see above response). We believe that volume is another important indicator that is related to both rift widening and supply of shelf ice to adjust the infill into the rift. We added how this process might occur in Discussion: "*A thick mélange layer can effectively "freeze" a rift, enabling mechanical stress transmission between its flanks and ultimately suppressing rift propagation (Rignot, 1998; Larour, 2021). In this study, we show that seracs appearing on the mélange surface are a part of the infill inside rifts. They are formed from partial collapsing of rift flanks. We propose that the surface part of the bottom rift wall is first excavated through interactions by tides (Padman et al., 2002 and 2008), melting caused by intrusion of warmer sea water into the rift (Poinelli 2023a), and other factors. This, in turn, causes the collapse of the upper part of the rift flank due to the removal of the bottom support. This process, like coastal bluff erosion, repeats itself and becomes one of the mechanisms that widen the rift and increase the mélange volume. We find that the increased mélange volume in a rift promotes its widening rate, and may further increase ice front calving and effect the ice shelf stability.*"

**Line 29: This line is a bit confusing, mass discharge across the GL is the cause of sea level rise, but what does 'lost ice mass enters' mean?**

Response:

We deleted the first half of the sentence. It is changed to: "…… *The lost ice mass enters the Southern Ocean from ice shelves mainly through two processes, namely shelf front calving and basal melting, each accounting for ~50% (Depoorter et al., 2013; Liu et al., 2015; Smith et al., 2020; Greene et al., 2022).*"

**Line 39: Like my comment in the abstract, I don't fully see the link between calving and ice mélange? What does 'the importance … on studying mélange … are fully recognized' mean?**

Response:

We changed this part of the section to strengthen the link: "*…… Furthermore, mélange (a mixture of shelf ice, snow, sea ice and water) changes related to ice shelf fracturing in Antarctica and glacier calving in Greenland are studied (Rignot and Macayeal, 1998; Larour et al., 2004; Cassotto et al., 2021). Specifically, mélange thickness reduction is observed during rift widening on Amery Ice shelf, Ronne Ice Shelf and Larsen C Ice Shelf (Fricker et al., 2005; Walker et al., 2021; Larour et al., 2021). Modelling results indicate that mélange may have the capability for transmitting stresses across rift flanks and thus, influences rift propagation and shelf front calving (Larour et al., 2004 and 2021). However, there has been a lack of large-scale high-resolution observations of 3D rift structural changes and mélange dynamics in a sustained long period to support such a conclusion. This, in turn, hinders our understanding of the role of mélange in ice shelf retreating and the mechanism of ice shelf stability weakening.*"

**Figure 1: Really nice figure.**
**Line 67: What do you mean with 'tie point'?**

Response:

We added the text to refer it to the detailed TP description in Section 2.2.2: "*We introduce tie points (TPs, see detailed description in Section 2.2.2) ……*"

**Line 76: This sentence is a bit confusing.**

Response:

We revised the sentence to clarify the point: "*Finally, we demonstrate the observed mélange changes and estimated volumetric changes in relation to rift widening as the rifts advect toward the ice shelf front during the study period.*"

**Line 124: This is good example of the scientific motivation of this study. Perhaps this should be included in the introduction and not in methodology.**

Response:

Yes, this part is now moved to Introduction.

**Line 157,176: Acronyms in the title are hard to follow ad makes the reading experience confusing.**

Response:

They are now spelled out in the titles.

**Figure 8-9: Great figures! Congrats!**

**Line 264: I appreciate the precision in these values but is it necessary? This may be a question of personal preference, but I find it a bit confusing to follow these numbers. These details are also reported in Table A3 no? So why don't discuss the orders of magnitude of these changes?**

Response:

We now keep the digits to meter for all numbers throughout the manuscript. We also give the orders of magnitude of these changes: "…… *we found that the rift T1 propagated consistently during the period, resulting in an accelerated widening by 647 m during 2014-2017 (44 %, 227 m y$^{-1}$) and 1,109 m during 2016-2021 (53 %, 268 m y$^{-1}$). Correspondingly, the rift has lengthened by 2,393 m during 2014-2017 (5%, 840 m y$^{-1}$), but only 1,082 m during 2016-2021 (2%, 262 m y$^{-1}$).*"

Here are data for both T1 and T2 (will update the manuscript for T2):

|  | Time | Rift T1 | Rift T2 |
|---|---|---|---|
| Length (m) | 2014 | 47131.86 | 48324.04 |
|  | 2017 | 49525.03 | 50025.03 |
|  | 2021 | 50607.86 | 51692.19 |
| Lengthening (m & m/y) | 2014-2017 | 2393.17 m (5%, 839.68 m/y) | 1701.00 m (%4, 597.39 m/y) |
|  | 2017-2021 | 1082.83 m (2%, 262.10 m/y) | 1667.15 m (%3, 405.95 m/y) |
| Width (m) | 2014 | 1456.65 | 1532.33 |
|  | 2017 | 2103.48 | 1944.04 |
|  | 2021 | 3212.20 | 2777.66 |
| Widening (m & m/y) | 2014-2017 | 646.83 m (44%, 226.95 m/y) | 411.71 m (27%, 144.59 m/y) |
|  | 2017-2021 | 1108.71 m (53%, 268.36 m/y) | 833.62 m (43%, 202.99 m/y) |

**Line 282: What is a 'cavity' in this context? Please specify.**

Response:

We changed it to: "*…… More specifically, we found cavities that are freshly opened during the fracturing process in the unmatured mélange section close to the west tip (Fig. 11a). These pre-mélange cavities are verified in ZY-3 stereo images (Fig. A1) ……*"

**Line 290: Volumetric change is a good parameter to monitor, but it would also be interesting to see an estimation of mélange thinning, which is relevant to ice sheet modeling. The observed decreased in ice mélange elevation most likely means that this layer has thinned. This is extremely important as the stress propagation between flanks may be compromised (Larour 2004).**

Response:

Here we added: "*The estimated change in mélange thickness H is relevant to ice sheet modeling. The observed mélange thinning may indicate that the stress propagation between flanks may be compromised (Larour 2004).*"

In Fig. 11 we changed the elevation profile to thickness profile, also added figures for T2.

[Figure]

**Line 300: What do you mean with 'newly calved'? I may have missed this, but I thought the ice shelf has not calved during this period. Do you perhaps mean that the rift has widened due to partial collapse of its flanks? If so, can you speculate about what may have caused it? Mélange thinning may be the answer itself, and I agree that it is hard to point at a cause. Setting up a discussion on this would be very important.**

Response:

We changed it to "…… *newly vacated space due to partial collapse of rift flank*".

We added a paragraph in Discussion to speculate its cause: "*A thick mélange layer can effectively "freeze" a rift, enabling mechanical stress transmission between its flanks and ultimately suppressing rift propagation (Rignot, 1998; Larour, 2021). In this study, we show that seracs appearing on the mélange surface are a part of the infill inside rifts. They are formed from partial collapsing of rift flanks. We propose that the surface part of the bottom rift wall is first excavated through interactions by tides (Padman et al., 2002 and 2008), melting caused by intrusion of warmer sea water into the rift (Poinelli 2023a), and other factors. This, in turn, causes the collapse of the upper part of the rift flank due to the removal of the bottom support. This process, like coastal bluff erosion, repeats itself and becomes one of the mechanisms that widen the rift and increase the mélange volume. We find that the increased mélange volume in a rift promotes its widening rate, and may further increase ice front calving and effect the ice shelf stability.*"

**Line 302: I was surprised to see that the manuscript does not include a discussion section. What are the strengths of this novel approach? How does this compare to previous studies that employed ICESat2? for example Walker 2021, Fricker 2005, …? Why is monitoring of these rifts important? Can you extend this novel processing techniques to other highly fractured areas of Antarctica (Larsen C, Totten, Brunt, Amery?)**

Response:

We added a Discussion section to address these points:

*"4 Discussion*

*The developed multi-temporal DEM adjustment model MDAM has shown its effectiveness in removing biases between adjacent multi-satellite sub-DEMs and establishing a unified and integrated DEM time series with an average elevation uncertainty of better than 0.24 m. We demonstrate the full 3D mapping capability for characterizing rift structure (rift lips, pre-mélange cavities, mélange seracs, etc.) and estimating dynamic mélange volume changes from the ICESat-2 controlled DEM time series, extending from rift topography and mélange thickness estimated along ground tracks from ICESat and ICESat-2 measurements in previous studies (Fricker et al., 2005; Walker et al., 2021). Such high-resolution mélange dynamic observations allow us to understand the mélange movement inside a closed environment of large Antarctic rifts, and to further study its role in rapid rift propagation and iceberg calving. To make this model working in a more dynamic open ocean environment, such as Pine Island and Thwaites ice shelves, modifications need to be carried out to address the rapid calving process with incoherent mélange changes in ice shelf front.*

*A thick mélange layer can effectively "freeze" a rift, enabling mechanical stress transmission between its flanks and ultimately suppressing rift propagation (Rignot, 1998; Larour, 2021). In this study, we show that seracs appearing on the mélange surface are a part of the infill inside rifts. They are formed from partial collapsing of rift flanks. We propose that the surface*

*part of the bottom rift wall is first excavated through interactions by tides (Padman et al., 2002 and 2008), melting caused by intrusion of warmer sea water into the rift (Poinelli 2023a), and other factors. This, in turn, causes the collapse of the upper part of the rift flank due to the removal of the bottom support. This process, like coastal bluff erosion, repeats itself and becomes one of the mechanisms that widen the rift and increase the mélange volume. We find that the increased mélange volume in a rift promotes its widening rate, and may further increase ice front calving and effect the ice shelf stability.*

*Future calving triggered by T1 and T2 and the resulting ice front retreat may expose the ice shelf to increased warm water intrusion, a process simulated for the Larsen C ice shelf and supported by modelling results (Poinelli, 2023a and 2023b). Although the iceberg calved from the shelf front in 1986 is located inside the Passive Shelf Ice (PSI) area, meaning no actual buttress reduction for support from the ice shelf to the ice sheet (Doake, 1998; Fürst et al., 2016), the combined rifts of T1 and T2 propagated rapidly recently to cover ~58% the ice shelf laterally. In the context of global warming, this is particularly relevant for the Filchner Ice Shelf, as warm water has recently been observed near Berkner Island (e.g., Davis, 2022). Therefore, T1 and T2 have the potential to trigger a larger calving beyond the PSI boundary and could ultimately lead to destabilization mechanisms like those proposed for the Larsen C Ice Shelf.”*

**Line 307: I may have missed it, but is there a similar analysis applicable to T2? The manuscript presents these rifts as a pair, yet the analysis appears to be restricted to T1, which seems inconsistent. To be clear, I am not suggesting additional analysis, but the rationale for focusing solely on T1 should be explicitly stated, even if it is due to data limitations.**

Response:

As suggested also by Reviewer 1, we added corresponding analysis for T2 throughout the manuscript. The following are the revised figures and tables. The details will be in the revised manuscript.

[Figure]

**Figure 9. Multi-satellite DEM time series of the large rifts of T1 and T2 on Filchner Ice Shelf from 2014 to 2021: (a) and (d) reconstructed ZY-3 DEM of 2014, (b) and (e) bias-corrected REMA DEMs of 2016-2017, and (c) and (f) bias-corrected REMA DEMs of 2020-2021. The boxes in (b) and (e) indicate the corresponding sections of T1 and T2 where details of 3D structures and mélange are presented in Fig. 10.**

[Figure]

**Figure 10. 3D sectional structure and mélange in T1 (a – d) and T2 (e – h): (a) shaded relief map of a section of T1 REMA DEMs of 2016-2017 (January 4, 2017) indicated by the box in Fig. 9b; (b) WorldView image (0.5 m resolution) of February 25, 2016; (c) elevation profile along the pink line in (a) and (b), and 3D rift and**

mélange structure parameters; and (d) rift and mélange changes along the profile from 2014 to 2021 (February 28, 2014 in blue, January 4, 2017 in green, and February 2, 2021 in red). (e) shaded relief map of a section of T2 REMA DEMs of 2016-2017 (October 25, 2016) indicated by the box in Fig. 9e; (f) WorldView image (0.5 m resolution) of March 2, 2016; (g) elevation profile along the pink line in (e) and (f), and 3D rift and mélange structure parameters; and (h) rift and mélange changes along the profile from 2014 to 2021 (February 28, 2014 in blue, October 25, 2016 in green, and February 3, 2021 in red). Elevation displayed in (a) and (e) is exaggerated by 10 times.

[Figure]

Figure 11. Thickness (a and c) and volume (b and d) of ice mélange inside the rift T1 and T2, respectively, from the multi-satellite DEM series from 2014 to 2021. Average thickness and volume of each transect that are separated every 1 km along the rift centerlines are illustrated from west tip to east tip

Table A4. Rift lip heights and differences measured for T1 and T2

T1

| Transect ID from west tip to east tip (1 km separation) | Land-ward lip height (L) (m) | Sea-ward lip height (S) (m) | Difference (S-L) (m) |
|---|---|---|---|
| 1 | -1.67 | 0.54 | 2.21 |
| 2 | 3.58 | 4.53 | 0.95 |
| 3 | 2.68 | 4.65 | 1.97 |
| 4 | 0.84 | 1.72 | 0.88 |
| 5 | 4.39 | 1.49 | -2.91 |
| 6 | 0.72 | 1.74 | 1.02 |
| 7 | 1.85 | 2.01 | 0.16 |
| 8 | 2.28 | 2.59 | 0.32 |
| 9 | 1.25 | 1.43 | 0.17 |
| 10 | 0.07 | 1.74 | 1.67 |
| 11 | -1.57 | 1.84 | 3.40 |
| 12 | -0.68 | 1.06 | 1.75 |
| 13 | 1.63 | 3.40 | 1.77 |
| 14 | 2.81 | 2.98 | 0.17 |
| 15 | 1.58 | 2.55 | 0.97 |
| 16 | 1.56 | 2.29 | 0.74 |
| 17 | 1.84 | 2.13 | 0.29 |
| 18 | 1.46 | 1.91 | 0.46 |
| 19 | 2.48 | 2.18 | -0.30 |
| 20 | 1.65 | 1.89 | 0.24 |
| 21 | 1.91 | 1.82 | -0.09 |
| 22 | 2.52 | 1.81 | -0.71 |
| 23 | 1.99 | 1.37 | -0.61 |
| 24 | 1.65 | 0.95 | -0.70 |
| 25 | 1.70 | 1.03 | -0.67 |
| 26 | 0.89 | 0.99 | 0.09 |
| 27 | 1.90 | 0.19 | -1.71 |
| 28 | 0.98 | 0.05 | -0.93 |
| 29 | 0.68 | 1.47 | 0.79 |
| 30 | 2.42 | 3.43 | 1.00 |
| 31 | 1.37 | 2.08 | 0.71 |
| 32 | 3.26 | 3.74 | 0.47 |
| 33 | 3.32 | 3.90 | 0.58 |
| 34 | 3.69 | 2.08 | -1.61 |
| 35 | 3.71 | 4.80 | 1.09 |
| 36 | 1.95 | 1.90 | -0.05 |
| 37 | 0.42 | 1.21 | 0.79 |

| 38 | -0.27 | -2.25 | -1.98 |
| 39 | 2.77 | 3.48 | 0.71 |
| 40 | 2.52 | 1.35 | -1.18 |
| 41 | 2.39 | 1.64 | -0.74 |
| 42 | 2.02 | 1.10 | -0.92 |
| 43 | 0.81 | 1.48 | 0.67 |
| 44 | 2.64 | 2.60 | -0.05 |
| 45 | 4.62 | 4.19 | -0.43 |
| 46 | 3.14 | 2.66 | -0.48 |
| 47 | -0.84 | 0.13 | 0.97 |
| 48 | 1.79 | 1.86 | 0.07 |
| 49 | -1.40 | -1.75 | -0.35 |

Average rift lip height difference (seaward - landward) within the section of the T1 mature mélange: Mean: 0.03 m; Max: 1.77 m (13 km from the west tip)

T2:

| Transect ID from west tip to east tip (1 km separation) | Land-ward lip height (L) (m) | Sea-ward lip height (S) (m) | Difference (S-L) (m) |
| --- | --- | --- | --- |
| 1 | 0.43 | 2.98 | 2.55 |
| 2 | 0.56 | 0.67 | 0.11 |
| 3 | -0.06 | 1.64 | 1.70 |
| 4 | 2.14 | 3.01 | 0.87 |
| 5 | 4.49 | 4.87 | 0.38 |
| 6 | 2.25 | 2.91 | 0.66 |
| 7 | 1.96 | 0.77 | -1.19 |
| 8 | 2.04 | 2.69 | 0.65 |
| 9 | -2.21 | -1.81 | 0.40 |
| 10 | -2.63 | -0.92 | 1.71 |
| 11 | 0.06 | 0.54 | 0.48 |
| 12 | -0.70 | 0.14 | 0.84 |
| 13 | 0.95 | 1.25 | 0.30 |
| 14 | 0.32 | -0.94 | -1.26 |
| 15 | -0.33 | -3.21 | -2.88 |
| 16 | -0.26 | -2.41 | -2.15 |
| 17 | 0.33 | -0.94 | -1.27 |
| 18 | 0.23 | 1.18 | 0.94 |
| 19 | 0.14 | 3.16 | 3.03 |
| 20 | -1.55 | 4.58 | 6.13 |
| 21 | -11.54 | 1.37 | 12.91 |
| 22 | -7.76 | -11.51 | -3.75 |
| 23 | -2.83 | 3.04 | 5.87 |
| 24 | 0.57 | 2.72 | 2.15 |

| | | | |
|---|---|---|---|
| 25 | 1.16 | 3.65 | 2.49 |
| 26 | 0.73 | -16.66 | -17.39 |
| 27 | -18.62 | -5.24 | 13.38 |
| 28 | -0.31 | -1.95 | -1.64 |
| 29 | 2.49 | -0.03 | -2.52 |
| 30 | 1.75 | 1.64 | -0.11 |
| 31 | 1.06 | 1.14 | 0.08 |
| 32 | 1.18 | 1.07 | -0.11 |
| 33 | 0.62 | -0.07 | -0.69 |
| 34 | 1.89 | 1.45 | -0.44 |
| 35 | 3.25 | 2.06 | -1.19 |
| 36 | 2.86 | 2.84 | -0.01 |
| 37 | 3.65 | 3.54 | -0.11 |
| 38 | 4.66 | 4.38 | -0.28 |
| 39 | 3.90 | 3.34 | -0.56 |
| 40 | 4.34 | 3.54 | -0.81 |
| 41 | 3.22 | 2.65 | -0.56 |
| 42 | 3.01 | 2.05 | -0.96 |
| 43 | 3.36 | 1.38 | -1.98 |
| 44 | 3.58 | 1.67 | -1.91 |
| 45 | 3.52 | 2.00 | -1.53 |
| 46 | 3.53 | 1.65 | -1.88 |
| 47 | 4.33 | 1.63 | -2.70 |
| 48 | 5.32 | 1.68 | -3.64 |
| 49 | 6.09 | 4.68 | -1.42 |

Average rift lip height difference (seaward - landward) within the section of the T2 mature mélange: Mean: 0.6 m; Max: 13.38 m (27 km from the west tip)

Yours sincerely,
Mattia Poinelli
Mattia.poinelli@uci.edu
University of California, Irvine
Jet Propulsion Laboratory, California Institute of Technology

REFERENCES:

Doake 1998 Breakup and conditions for stability of the northern Larsen Ice Shelf, Antarctica, Nature
Fuerst 2016, The safety band of Antarctic ice shelves, Nature Climate Change
Larour 2021, Physical processes controlling the rifting of Larsen C Ice Shelf, Antarctica, prior to the calving of iceberg A68, PNAS

**Davis 2022, Observations of Modified Warm Deep Water Beneath Ronne Ice Shelf, Antarctica, From an Autonomous Underwater Vehicle, JGR: Oceans**

**Poinelli 2023a: Can rifts alter ocean dynamics beneath ice shelves? The Cryosphere**

**Poinelli 2023b: Ice-Front Retreat Controls on Ocean Dynamics Under Larsen C Ice Shelf, Antarctica, GRL**

**Rignot 1998, Ice-shelf dynamics near the front of the Filchner-Ronne Ice Shelf, Antarctica, revealed by SAR interferometry, GRL**

**Larour 2004, Modelling of rift propagation on Ronne Ice Shelf, Antarctica, and sensitivity to climate change, GRL**

**Walker 2021, A High Resolution, Three-Dimensional View of the D-28 Calving Event From Amery Ice Shelf With ICESat-2 and Satellite Imagery, GRL**

**Fricker 2005, ICESat's new perspective on ice shelf rifts: The vertical dimension, GRL**

Response:

The above references are all cited in places, as suggested, in the manuscript.

---

## Author Response (AR1)

We appreciate the constructive comments from two reviewers (Dr. Ann-Sofie Priergaard Zinck, and Dr. Mattia Poinelli). Our manuscript will be much improved by their input. As informed in the letter of requesting responses, this file contains author comments (ACs). In the following responses, we use "**bold**" text for comments, "non-bold" text for our responses, and "italic" for changed text in the revised manuscript.

**Referee #1**

**The authors present a new method for co-registering and temporally aligning REMA DEMs. This method is in itself quite interesting, and a good addition to existing methods which are more focused on ice shelf basal melting. The resulting DEMs are further applied to describe the 3D development of a rift and its mélange on the Filchner Ice Shelf. The 3D investigation reveals rapid expansion of the rift which is attributed to calved shelf ice from the walls within the rift. This study thus underscores the importance of high-resolution elevation models in understanding rift and mélange development.**

**As seen below I do have a few major comments related to this paper. First, I find the current structure of the Methods section confusing and difficult to follow and have made some suggestions on how to improve that. Secondly, the current manuscript gives false expectations of analysis of two rifts on the Filchner Ice Shelf, whereas only one is thoroughly studied. I, therefore, suggest performing the analysis to both rifts. Furthermore, the current manuscript is lacking a discussion of the method and results as well as a broader discussion of the impact of this study. Lastly, to make this new MDAM method valuable to the community I think that it is important that the code becomes publicly available.**

Response:

1) We have performed the suggested restructure of the manuscript, see the detailed description in the following responses to the Major Comments.

2) As suggested, we have added the results of T2 into the manuscript, see the detailed description in the following responses to the Major Comments.

3) Yes, we are adding a Discussion section to address limitations of the methods, mélange dynamics learned, and impact of the results, as suggested.

4) We have posted our MDAM code and data of the Filchner project (DEMs, TPs, and GCPs) to the GitHub and the Zenodo, respectively. See the detailed description in the following responses to the specific comment.

**Major comments:**
**Structure of the Methods section:**

**The current structure of the Methods section makes it very difficult to follow the different steps as there are quite some jumps in the storyline. First, I miss a Data section in-between the Introduction and Methods sections. In the current manuscript, the data is briefly described in L135-155, which is after the presentation of the MDAM method. That location of the data description makes it very difficult to follow the MDAM method. It also breaks the flow of the methods as the reader must wait quite long from the first mentioning of the TPs and GCPs before their selection procedure is described. Secondly, there are Introduction elements present in the Methods (L120-129), which I suggest being moved to the Introduction. Finally, the Results section contains a lot of elements which I think belong in the Methods section (Sect. 3.1 and L289-298), as they contain information to the methods as opposed to presenting actual results.**

Response:

We agree with the suggestions. We are restructuring the manuscript accordingly.

First, as suggested, we move the data description to the location between Introduction and Method. Some minor revisions are made to provide the context and make this move smoothly.

Secondly, the Introduction elements presented in the Method (L120-129) are moved to the Introduction section, as suggested.

Finally, we moved those elements in the Results section (Sect. 3.1 and L289-298) that contain methods information to the Method section.

**Rift T2:**

**You present both rifts T1 and T2 in the Methods and Fig. 3, but your analysis focuses solely on rift T1. I would, therefore, suggest that you include a similar analysis of T2 as the one you have made for T1. This will likely also strengthen your manuscript with regards to melange dynamics.**

Response:

The corrected DEMs and measurements for T2 are readily available. We have added figures and data of T2 (see some of them here) to make the analysis more complete, as suggested. We have completed the relevant analysis in the revised manuscript.

[Figure]

**Figure 9. Multi-satellite DEM time series of the large rifts of T1 and T2 on Filchner Ice Shelf from 2014 to 2021: (a) and (d) reconstructed ZY-3 DEM of 2014, (b) and (e) bias-corrected REMA DEMs of 2016-2017, and (c) and (f) bias-corrected REMA DEMs of 2020-2021. The boxes in (b) and (e) indicate the corresponding sections of T1 and T2 where details of 3D structures and mélange are presented in Fig. 10.**

[Figure]

[Figure]

**Figure 10. 3D sectional structure and mélange: (a) and (e) are sectional shaded relief maps of REMA DEMs of 2016-2017 (January 4, 2017 and October 25, 2016) indicated by the boxes in Fig. 9; (b) and (f) are WorldView images (0.5 m resolution) of February 25, 2016 and March 2, 2016; (c) and (g) are elevation profiles along the pink lines in (a) and (e), and 3D rift and mélange structure parameters; (d) and (h) are rift and mélange changes along the profiles from 2014 to 2021 (2014 in blue, 2016-2017 in green, and 2020-2021 in red). Elevation displayed in (a) and (e) are exaggerated by 10 times.**

[Figure]

**Figure 11. Elevation (a and c) and volume (b and d) of ice mélange inside the rifts T1 and T2 from the multi-satellite DEM series from 2014 to 2021. Average elevation and volume of each transect that are separated every 1 km along the rift centerline are illustrated from west tip to east tip.**

**Table A3. Heights of rift lips measured on the land-ward (L) side and sea-ward (S) side, and their difference**

**(S-L) at each transect along the centerlines of Rift T1 and T2 (REMA DEMs of 2016-2017).**

Rift T1:

| Transect ID from west tip to east tip (1 km separation) | Land-ward lip height (L) (m) | Sea-ward lip height (S) (m) | Difference (S-L) (m) |
|---|---|---|---|
| 0 | -1.7 | 0.5 | 2.2 |
| 1 | 3.6 | 4.5 | 1.0 |
| 2 | 2.7 | 4.7 | 2.0 |
| 3 | 0.8 | 1.7 | 0.9 |
| 4 | 4.4 | 1.5 | -2.9 |
| 5 | 0.7 | 1.7 | 1.0 |
| 6 | 1.8 | 2.0 | 0.2 |
| 7 | 2.3 | 2.6 | 0.3 |
| 8 | 1.3 | 1.4 | 0.2 |
| 9 | 0.1 | 1.7 | 1.7 |
| 10 | -1.6 | 1.8 | 3.4 |
| 11 | -0.7 | 1.1 | 1.7 |
| 12 | 1.6 | 3.4 | 1.8 |
| 13 | 2.8 | 3.0 | 0.2 |
| 14 | 1.6 | 2.6 | 1.0 |
| 15 | 1.6 | 2.3 | 0.8 |
| 16 | 1.8 | 2.1 | 0.3 |
| 17 | 1.5 | 1.9 | 0.5 |
| 18 | 2.5 | 2.2 | -0.3 |
| 19 | 1.6 | 1.9 | 0.2 |
| 20 | 1.9 | 1.8 | -0.1 |
| 21 | 2.3 | 1.6 | -0.7 |
| 22 | 2.0 | 1.4 | -0.6 |
| 23 | 1.7 | 0.9 | -0.7 |
| 24 | 1.7 | 1.0 | -0.7 |
| 25 | 0.9 | 1.0 | 0.1 |
| 26 | 1.9 | 0.3 | -1.6 |
| 27 | 1.0 | 0.1 | -0.9 |
| 28 | 0.7 | 1.5 | 0.8 |
| 29 | 2.4 | 3.4 | 1.0 |
| 30 | 1.4 | 2.1 | 0.7 |
| 31 | 3.3 | 3.7 | 0.5 |
| 32 | 3.3 | 3.9 | 0.6 |
| 33 | 3.7 | 2.1 | -1.6 |
| 34 | 3.7 | 4.8 | 1.1 |
| 35 | 2.0 | 1.9 | 0.0 |
| 36 | 0.4 | 1.2 | 0.8 |
| 37 | -0.3 | -2.3 | -2.0 |
| 38 | 2.8 | 3.5 | 0.7 |

| | | | |
|---|---|---|---|
| 39 | 2.5 | 1.3 | -1.2 |
| 40 | 2.4 | 1.7 | -0.6 |
| 41 | 2.0 | 1.1 | -0.9 |
| 42 | 0.8 | 1.5 | 0.7 |
| 43 | 2.6 | 2.6 | 0.0 |
| 44 | 4.6 | 4.2 | -0.4 |
| 45 | 3.1 | 2.7 | -0.5 |
| 46 | -0.8 | 0.1 | 1.0 |
| 47 | 1.8 | 1.9 | 0.1 |
| 48 | -1.4 | -1.8 | -0.4 |

Average lip height difference (s – l) within the section of mature mélange of T1: 0.1 m (max 1.8 m at 12 km from west tip).

Rift T2:

| Transect ID from west tip to east tip (1 km separation) | Land-ward lip height (L) (m) | Sea-ward lip height (S) (m) | Difference (S-L) (m) |
|---|---|---|---|
| 0 | 0.4 | 3.0 | 2.5 |
| 1 | 0.6 | 0.7 | 0.1 |
| 2 | -0.1 | 1.6 | 1.7 |
| 3 | 2.1 | 3.0 | 0.9 |
| 4 | 4.5 | 4.9 | 0.4 |
| 5 | 2.3 | 2.9 | 0.7 |
| 6 | 2.0 | 0.8 | -1.2 |
| 7 | 2.0 | 2.7 | 0.7 |
| 8 | -2.2 | -1.8 | 0.4 |
| 9 | -2.6 | -0.9 | 1.7 |
| 10 | 0.1 | 0.5 | 0.5 |
| 11 | -0.7 | 0.1 | 0.8 |
| 12 | 0.9 | 1.2 | 0.3 |
| 13 | 0.3 | -0.9 | -1.3 |
| 14 | -0.3 | -3.2 | -2.9 |
| 15 | -0.3 | -2.4 | -2.2 |
| 16 | 0.3 | -0.9 | -1.3 |
| 17 | 0.2 | 1.2 | 0.9 |
| 18 | 0.1 | 3.2 | 3.0 |
| 19 | -1.6 | 4.6 | 6.1 |
| 20 | -11.5 | 1.4 | - |
| 21 | -7.8 | -11.5 | -3.7 |
| 22 | -2.8 | 3.0 | 5.9 |
| 23 | 0.6 | 2.7 | 2.2 |
| 24 | 1.2 | 3.6 | 2.5 |
| 25 | 0.7 | -16.7 | - |
| 26 | -18.6 | -5.2 | - |

| | | | |
|---|---|---|---|
| 27 | -0.3 | -1.9 | -1.6 |
| 28 | 2.5 | 0.0 | -2.5 |
| 29 | 1.8 | 1.6 | -0.1 |
| 30 | 1.1 | 1.1 | 0.1 |
| 31 | 1.2 | 1.1 | -0.1 |
| 32 | 0.6 | -0.1 | -0.7 |
| 33 | 1.9 | 1.5 | -0.4 |
| 34 | 3.3 | 2.1 | -1.2 |
| 35 | 2.9 | 2.8 | 0.0 |
| 36 | 3.6 | 3.5 | -0.1 |
| 37 | 4.7 | 4.4 | -0.3 |
| 38 | 3.9 | 3.3 | -0.6 |
| 39 | 4.3 | 3.5 | -0.8 |
| 40 | 3.2 | 2.7 | -0.6 |
| 41 | 3.0 | 2.0 | -1.0 |
| 42 | 3.4 | 1.4 | -2.0 |
| 43 | 3.6 | 1.7 | -1.9 |
| 44 | 3.5 | 2.0 | -1.5 |
| 45 | 3.5 | 1.7 | -1.9 |
| 46 | 4.3 | 1.6 | -2.7 |
| 47 | 5.3 | 1.7 | -3.6 |
| 48 | 6.1 | 4.7 | -1.4 |
| 49 | 5.0 | 6.2 | 1.2 |

Average lip height difference (s – l) within the section of mature mélange of T2: 0.2 m (max 5.9 m at 22 km from west tip).

**Discussion:**

**The manuscript in its current state does not contain any discussion of the results. I, therefore, miss a separate Discussion section which should at least include a discussion on the following topics:**

**1.**

**A broader discussion of your MDAM method, including a discussion of the limitations of the method.**

**2.**

**A broader discussion on what you have learned about mélange dynamics because of this study.**

**3.**

**A discussion on the broader impact of your results. E.g., how do these results aid in assessing ice shelf instability, which you for instance mention in the Introduction.**

Response:

Yes, we added a Discussion section to address these three points, along with those from Reviewer 2:

*"4 Discussion*

*The developed multi-temporal DEM adjustment model MDAM has shown its effectiveness in removing biases between adjacent multi-satellite sub-DEMs and establishing a unified and integrated DEM time series. We demonstrate the full 3D mapping capability of characterizing rift structure (rift lips, pre-mélange cavities, mélange seracs, etc.) and estimating dynamic mélange volume changes from the ICESat-2 controlled DEM time series, extending from rift topography and mélange thickness estimated along ground tracks from ICESat and ICESat-2 measurements in previous studies (Fricker et al., 2005; Walker et al., 2021). Such high-resolution mélange dynamic observations allow us to understand the mélange movement inside a closed environment of large Antarctic rifts, and to further study its role in rapid rift propagation and iceberg calving. To make this model working in a more dynamic open ocean environment, such as Pine Island and Thwaites ice shelves, modifications need to be carried out to address the rapid calving process with incoherent mélange changes in ice shelf front.*

*A thick mélange layer can effectively "freeze" a rift, enabling mechanical stress transmission between its flanks and ultimately suppressing rift propagation (Rignot and MacAyeal, 1998; Larour et al, 2021). In this study, we show that seracs appearing on the mélange surface are a part of the infill inside rifts. They are formed from partial collapsing of rift flanks. We propose that the surface part of the bottom rift wall is first excavated through interactions by tides (Padman et al., 2002 and 2008), melting caused by intrusion of warmer seawater into the rift (Poinelli et al., 2023a) and other factors. This, in turn, causes the collapse of the upper part of the rift flank due to the removal of the bottom support. This process, like coastal bluff erosion, repeats itself and becomes one of the mechanisms that widen the rift and increase the mélange volume. We find that the increased mélange volume in a rift promotes its widening rate, and may further increase ice front calving and effect the ice shelf stability.*

*Although the iceberg calved from the shelf front in 1986 is mostly located inside the Passive Shelf Ice (PSI) area (Fig. A1), meaning no significant buttress reduction from the ice shelf (Doake et al., 1998; Fürst et al., 2016), the combined rifts of T1 and T2 propagated rapidly recently and have already covered ~58% the ice shelf laterally from Coats Land to Berkner Island. As warm water has recently been observed near Berkner Island (Davis et al., 2022), T1 and T2 have the potential to trigger a larger calving beyond the PSI boundary, which may be caused by a destabilization mechanism like that proposed for the Larsen C Ice Shelf (Poinelli et al., 2023a and 2023b)."*

**Code and data availability:**
**A major part of this manuscript is the development of the MDAM method alongside with the resulting DEMs. I, therefore, think that it is necessary that the MDAM code (incl. TPs and GCPs) becomes publicly available on e.g., GitHub to be a truly addition to the cryospheric community. Other similar methods (Shean et al. 2019 and Zinck et al. 2023) are likewise publicly available. Furthermore, it would be desirable to make the produced DEMs publicly available online.**
Response:

The MDAM code (including TPs and GCPs) is now available at GitHub (https://github.com/menglianxia/MDAM).

The produced DEMs (including adjusted REMA DEMs of 2016-2017 and 2020-2021, and ZY-3 DEM of 2014) are available at https://doi.org/10.5281/zenodo.15260323 (Xia et al. 2025).

Reference:

Xia, M., Li, R., Scaioni, M., An, L., Li, Z., and Qiao, G.: Dataset belonging to the article: Building multi-satellite DEM time series for insight into mélange inside large rifts in Antarctica, Zenodo, https://doi.org/10.5281/zenodo.15260323, 2025.

**Specific comments:**

**L12-15: The first two sentences of the abstract discuss "front calving", whereas this sentence discusses rift structural changes and mélange dynamics. I would suggest adding an extra sentence before this sentence which elaborates on mélange dynamics and how it is hypothesized to be related to front calving.**

Response:

We now add a sentence: "*Mélange dynamics inside rifts is recognized to potentially influence the rift propagation and subsequent iceberg calving.*"

**L14-17: Long and complex sentence. Consider splitting into two for clarity.**

Response:

The sentence is split into two sentences: "*We propose an innovative multi-temporal digital elevation model (DEM) adjustment model (MDAM) to build a multi-satellite DEM time series from meter-level resolution small DEMs. It removes biases across large Antarctic ice shelves, as large as ~6 m in elevation, caused by tides, ice flow dynamics, and observation errors.*"

**L14-15: Remember to explain acronyms. "We propose an innovative multi-temporal DEM…" → "We propose an innovative multi-temporal digital elevation model (DEM)…"**

Response:

We changed it accordingly.

**L17: Remember to explain acronyms. "Using 30 REMA…" → "Using 30 Reference Elevation Model of Antarctica (REMA)…"**

Response:

We changed it accordingly.

**L18: Consider changing to ", the second largest ice shelf in Antarctica." or remove entirely.**

Response:

We changed it to ", *the second largest ice shelf in Antarctica.*"

**L24: You only use the acronym GSLR in this line and in L29, so I would write it out instead.**
Response:
Thanks. The suggestion is well taken.

**L26-27: Should be "The Antarctic Ice Sheet (AIS) has shown a persistent pattern of mass loss and has contributed to global sea level rise (GSLR) since the beginning of the satellite earth observation era in the 1960s (…"**
Response:
We changed it accordingly.

**L28: I don't understand the use of "Although" in the beginning of this sentence. I would rephrase this sentence with a focus on the importance of ice shelf calving instead of a focus on the grounding line.**
Response:
We rephrased it to: "*The lost ice mass enters the Southern Ocean from ice shelves mainly through two processes, namely shelf front calving and basal melting, each accounting for ~50% (Depoorter et al., 2013; Liu et al., 2015; Smith et al., 2020; Greene et al., 2022)*".

**L37: "advect to shelf front" → "advect to the shelf front"**
Response:
It is so changed.

**L38-41: I miss a clear definition of what a melange is.**
Response:
The sentence is changed to: "*Furthermore, mélange (a mixture of shelf ice, snow, sea ice and water) changes related to ice shelf fracturing in Antarctica and glacier calving in Greenland are studied (Rignot and MacAyeal, 1998; Larour et al., 2004; Cassotto et al., 2021).*"

**L56-57: Both Shean et al. 2019 and Zinck et al. 2023 handle heterogenous offsets between individual DEMs by applying dynamic corrections such as tides and by displacing the DEMs based on ice flow. So, I am not sure that I agree that there is a lack in methods. However, both Shean et al. 2019 and Zinck et al. 2023 do their analysis with a focus on basal melting, and not with a focus on deriving elevation maps alone. I would, therefore, rephrase this sentence and make the focus on rifts and mélange dynamics stronger.**
Response:

Agree. It is changed to: "*There is a lack of methods for handling heterogenous offsets between individual DEMs caused by rifts, mélange dynamics, and other factors in a dynamic Antarctic ice shelf environment.*"

**L73: "the second largest in Antarctica" → "the second largest ice shelf in Antarctica"**
Response:
Changed accordingly.

**L74: "time series of 2014-" → "time series from 2014-"**
Response:
Changed accordingly.

**L74: "MDAM for quantitatively" → "MDAM by quantitatively"**
Response:
Changed accordingly.

**L83-89: You mention that you correct the DEMs for tides, but do you also correct the DEMs for the inverse barometer effect? And if not, what impact do you believe that to have on your results?**
Response:
We did not apply inverse barometer effect (IBE) corrections to the DEMs explicitly. We performed the following analysis to show that this does not have an impact on our results.

We use the fifth-generation ECMWF reanalysis (ERA5), with an hourly interval and a spatial resolution of $0.25° \times 0.25°$, to calculate the IBE corrections. Based on the mean sea level pressure from 2014 to 2021, we computed the sea level pressure anomalies at the observation times for each DEM, and subsequently converted them into elevation changes using a rate of 1 cm/hPa (Padman et al., 2003; Chen et al., 2023).

The IBE corrections (influence on surface elevation) for all REMA DEMs of 2016–2017 on the ice shelf vary between –13.2 cm and 10.9 cm, those for all REMA DEMs of 2020–2021 on the ice shelf vary between –4.9 cm and 29.7 cm. We further did an experiment to find the IBE correction variation within smaller areas, namely within areas of individual DEMs. We found that the IBE correction changes little within a smaller DEM extent, with standard deviations up to 7 mm. That means that within the area of a DEM, the IBE correction can be treated as a constant.

On the other hand, in our MDAM model, elevation bias at TPs of each DEM is removed by a linear adjustment formula $a_o^k + a_1^k Y_{OL_i}^{\ k}$ where the IBE correction, now treated as a constant, is combined with the constant term $a_o^k$ and corrected implicitly by the least-squares solution. The following is a numerical proof. For each DEM, we calculate the bias correction dz with and without the IBE correction.

REMA DEMs of 2016-2017:

| DEM_ID | dz (no IBE correction) (m) | dz (with IBE correction) (m) | d_dz (m) |
|---|---|---|---|
| 1 | 3.1462 ±0.0715 | 3.1464 ±0.0714 | -0.0002 |
| 2 | -1.1511 ±0.1325 | -1.1499 ±0.1325 | -0.0012 |
| 3 | -1.0808 ±0.1775 | -1.0785 ±0.1775 | -0.0023 |
| 4 | 5.9135 ±0.2356 | 5.9168 ±0.2356 | -0.0033 |
| 5 | -5.7584 ±0.2938 | -5.7547 ±0.2938 | -0.0037 |
| 6 | 4.3821 ±0.3558 | 4.3853 ±0.3557 | -0.0032 |
| 7 | -0.2665 ±0.4265 | -0.2648 ±0.4265 | -0.0017 |
| 8 | -1.0733 ±0.3817 | -1.0731 ±0.3817 | -0.0002 |
| 9 | -1.9812 ±0.3268 | -1.9848 ±0.3268 | 0.0035 |
| 10 | -2.3278 ±0.2979 | -2.3346 ±0.2979 | 0.0068 |
| 11 | 3.1072 ±0.2351 | 3.0997 ±0.2351 | 0.0075 |
| 12 | 2.6538 ±0.1933 | 2.6482 ±0.1933 | 0.0056 |
| 13 | -2.3968 ±0.1619 | -2.3994 ±0.1619 | 0.0026 |
| 14 | -0.4772 ±0.0898 | -0.4775 ±0.0898 | 0.0003 |

REMA DEMs of 2020-2021:

| DEM_ID | dz (no IBE correction) (m) | dz (with IBE correction) (m) | d_dz (m) |
|---|---|---|---|
| 1 | -3.9234 ±0.1294 | -3.9241 ±0.1293 | 0.0007 |
| 2 | -1.0231 ±0.1379 | -1.0231 ±0.1378 | -0.0001 |
| 3 | -2.8520 ±0.1477 | -2.8506 ±0.1476 | -0.0015 |
| 4 | 0.6495 ±0.1943 | 0.6503 ±0.1943 | -0.0007 |
| 5 | -0.2422 ±0.1684 | -0.2419 ±0.1683 | -0.0003 |
| 6 | -1.2167 ±0.1784 | -1.2169 ±0.1784 | 0.0002 |
| 7 | 0.1385 ±0.2629 | 0.1380 ±0.2628 | 0.0005 |
| 8 | -1.3039 ±0.2795 | -1.3046 ±0.2794 | 0.0007 |
| 9 | 2.0558 ±0.2143 | 2.0545 ±0.2143 | 0.0012 |
| 10 | -0.7338 ±0.1628 | -0.7343 ±0.1628 | 0.0004 |
| 11 | -11.0123 ±0.1256 | -11.0128 ±0.1256 | 0.0005 |
| 12 | -1.7191 ±0.2013 | -1.7187 ±0.2012 | -0.0004 |
| 13 | 1.1548 ±0.2385 | 1.1537 ±0.2385 | 0.0011 |
| 14 | 2.1978 ±0.1969 | 2.1983 ±0.1968 | -0.0005 |
| 15 | -1.8438 ±0.1122 | -1.8434 ±0.1122 | -0.0004 |

The average difference between the resulting dz with and without the IBE corrections is 0.08 ± 0.47 cm (maximum 1.3 cm) for REMA DEMs of 2016–2017. The corresponding value for REMA DEMs of 2020–2021 is –0.01 ± 0.10 cm (maximum 0.27 cm). Similarly, for the ZY-3 DEM of 2014, the average difference does not exceed 2 cm.

Therefore, both methodological analysis and numerical experiments show that in our special case of using MDAM model to correct elevation bias, the IBE corrections can be treated as a

constant within a small extent of the DEMs and can be taken care of by the linear correction model within the MDAM model. No explicit IBE corrections are needed here and no impact on the accuracy is found.

Reference:

Chen, H., Rignot, E., Scheuchl, B., and Ehrenfeucht, S.: Grounding Zone of Amery Ice Shelf, Antarctica, From Differential Synthetic-Aperture Radar Interferometry, Geophys. Res. Lett., 50, e2022GL102430, 10.1029/2022GL102430, 2023.

Padman, L., King, M., Goring, D., Corr, H., and Coleman, R.: Ice-shelf elevation changes due to atmospheric pressure variations, J. Glaciol., 49, 521-526, 10.3189/172756503781830386, 2003.

Hersbach, H., Bell, B., Berrisford, P., Biavati, G., Horányi, A., Muñoz Sabater, J., Nicolas, J., Peubey, C., Radu, R., Rozum, I., Schepers, D., Simmons, A., Soci, C., Dee, D., Thépaut, J-N., ERA5 hourly data on single levels from 1940 to present. Copernicus Climate Change Service (C3S) Climate Data Store (CDS), 10.24381/cds.adbb2d47 (Accessed on April 22, 2025), 2023.

**L85: In case you are not aware of it, there is an updated version to the CATS2008 tide model (CATS2008_v2023). I do not think that it is necessary to change tide model for this paper, but it might be worth changing in potential future paper as the updated tide model also has an updated grounding line.**

Response:

Thanks for the information. We will use the new version in future publications.

**L87: When you mention the TPs here, I would make sure to refer to the later section where you elaborate on how these TPs are selected. I would also do the same for the GCPs.**

Response:

In this paragraph we added text "*(see detailed description in Section 2.3 )*" in places mentioning TPs and GCPs.

**L89-92: I find this sentence quite complicated and difficult to follow. Consider splitting it into two, first one about the residuals (so the second part of the sentence). And secondly a sentence on the least-squares adjustment method.**

Response:

We split it into two sentences: "*In the model inconsistencies at TP and GCP pairs are first reduced by the tide and velocity corrections, leaving the uncertainty originated from the sub-DEM production as residuals, including photogrammetric measurement errors and ephemeris data errors. Thereafter, we use the least-squares adjustment method (McGlone et al., 2004) to estimate the unknown bias corrections of all sub-DEMs by minimizing the residuals (Li, 1998; McGlone et al., 2004; Shean et al., 2019).*"

**L103: "given in a velocity map": What velocity map do you refer to here? In L88 you mention that you use the ITS_LIVE velocities (which could mean both the average product or image-pairs) and in Figure 1 you mention that you use the ITS_LIVE image-pair velocity maps. So, what ITS_LIVE velocity product do you use? And how does that relate to the acquisition time of the various sub-DEMs?**

Response:

We added a sentence to explain it: " *Here, ITS_LIVE image-pair velocity maps are selected to spatially match the extent of each sub-DEM and to temporally align as closely as possible with the acquisition times of two adjacent sub-DEMs.*" Most sub-DEMs and the corresponding velocity maps show good temporal overlap.

REMA DEMs of 2016-2017:

| ID of adjacent sub-DEMs | Time span of adjacent sub-DEMs | Time span of image-pair velocity map |
| --- | --- | --- |
| 1, 2 | 2016/10/31-2017/01/04 | 2016/11/01-2017/01/04 |
| 2, 3 | 2016/12/19-2017/01/04 | 2016/12/19-2017/01/04 |
| 3, 4 | 2016/12/19-2016/12/20 | / |
| 4, 5 | 2016/10/06-2016/12/20 | 2016/10/06-2016/11/07 |
| 5, 6 | 2016/10/06-2016/12/19 | 2016/10/06-2016/11/07 |
| 6, 7 | 2016/12/19-2016/12/19 | / |
| 7, 8 | 2016/12/19-2017/01/04 | 2016/12/19-2017/01/04
2016/10/31-2017/01/03 |
| 8, 9 | 2016/10/25-2017/01/04 | 2016/10/31-2017/01/03 |
| 9, 10 | 2016/10/25-2016/10/25 | / |
| 10, 11 | 2016/10/25-2016/12/12 | 2016/10/08-2016/11/25 |
| 11, 12 | 2016/12/12-2016/12/20 | 2016/10/31-2017/01/03 |
| 12, 13 | 2016/10/04-2016/12/20 | 2016/10/31-2017/01/03 |
| 13, 14 | 2016/10/04-2016/10/31 | 2016/09/29-2016/10/31
2016/10/06-2016/11/07 |

REMA DEMs of 2020-2021:

| ID of adjacent sub-DEMs | Time span of adjacent sub-DEMs | Time span of image-pair velocity map |
| --- | --- | --- |
| 1, 2 | 2021/03/13-2021/03/13 | / |
| 2, 3 | 2021/01/26-2021/03/13 | 2020/11/23-2021/02/21 |
| 3, 4 | 2021/01/16-2021/01/26 | 2020/11/23-2021/02/21 |
| 4, 5 | 2021/01/16-2021/01/26 | 2020/10/04-2021/02/09 |

| 5, 6 | 2021/01/17-2021/01/26 | 2020/10/04-2021/02/09 |
|---|---|---|
| 6, 7 | 2021/01/17-2021/02/21 | 2021/01/07-2021/02/08 |
| 7, 8 | 2021/02/02-2021/02/21 | 2021/01/07-2021/02/08 |
| 8, 9 | 2021/02/02-2021/02/03 | / |
| 9, 10 | 2021/01/26-2021/02/03 | 2021/01/07-2021/02/08 |
| 10, 11 | 2021/01/26-2021/02/03 | 2021/01/07-2021/02/08 |
| 11, 12 | 2021/01/22-2021/02/03 | 2021/01/07-2021/02/08 |
| 12, 13 | 2020/11/09-2021/01/22 | 2020/11/14-2021/01/23 |
| 13, 14 | 2020/10/03-2020/11/09 | 2020/10/29-2021/01/07 |
| 14, 15 | 2020/09/23-2020/10/03 | 2020/10/03-2020/12/22 |

**L104: Could you give an example of how these epsilons could be interpreted? Do they represent the error?**

Response:

The sentence is changed to: "($\varepsilon_{x_i}^{\ k}$, $\varepsilon_{y_i}^{\ k}$, $\varepsilon_{z_i}^{\ k}$) *represent residuals at TPs in the observation equations, including photogrammetric measurement errors and ephemeris data errors.*"

**L105-108: I miss an explanation of what the GCP epsilons represent.**

Response:

We added a sentence: "($\varepsilon_{x_i}^{\ GCP,k}$, $\varepsilon_{y_i}^{\ GCP,k}$, $\varepsilon_{z_i}^{\ GCP,k}$) *represent residuals at GCPs in the observation equations, describing inconsistencies between the integrated DEM and the outside control data (e.g., ICESat-2).*"

**L108-112: You mention tilting in the y-direction and how you deal with that. But how do you deal with tilting in the x-direction?**

Response:

According to Shean et al. (2016, 2019), a limited number of REMA strips longer than 110 km may exhibit "tilting" in the y-direction (along-track). Here it is approximately aligned with the ice flow direction on the Filchner Ice Shelf. This type of tilting typically ranges from ~1 to 3 m. Tilting in the x-direction (across-track) also occurs occasionally, but within a smaller magnitude (~1 m).

We only correct for the tilting in the primary (y) direction. Since the ground control points are distributed on both sides of the ice shelf, the influence of cross-track (x-direction) tilting can be effectively constrained. According to the error estimates and validation against ICESat-2 data, the DEM tilting has been effectively corrected.


**L116-117: If the "bias correction parameters" mentioned here correspond to the epsilons in eq. 1 and 2 then I would state that here. E.g. "Uncertainties of the estimated bias correction parameters ($\varepsilon_{x_i}^{k}$, $\varepsilon_{y_i}^{k}$, $\varepsilon_{z_i}^{k}$, $\varepsilon_{x_i}^{GCP,k}$, $\varepsilon_{y_i}^{GCP,k}$, $\varepsilon_{z_i}^{GCP,k}$) are computed…".**

Response:

The sentence is changed to: "*Uncertainties of the estimated bias correction parameters ($\sigma_{x_i}^{k}$, $\sigma_{y_i}^{k}$, $\sigma_{z_i}^{k}$, $\sigma_{x_i}^{GCP,k}$, $\sigma_{y_i}^{GCP,k}$, $\sigma_{z_i}^{GCP,k}$) are computed through an error propagation within the optimization procedure.*"

**L120-129: I think that this part belongs in the Introduction as it nicely frames the need of studying rifts at a high resolution. I would suggest to incorporate it into the last paragraph of the Introduction (L65-73) and move some of the technical parts of the last paragraph to the Methods section.**

Response:

The paragraph is moved and revised accordingly.

**L125: "Recently, two large rifts, T1 and T2,…" → "Recently, two large rifts, T1 and T2 (Fig. 3),…"**

Response:

It is changed as suggested.

**L127: "To study their…" → "Studying their…"**

Response:

It is so changed.

**L135-140: I would rephrase some of these sentences here to make it clear that you use the REMA strips. The first time that I read it I was in doubt as to whether you used the REMA strips or if you processed the WorldView images yourself.**

Response:

It is now changed to make this clear: "*The sub-DEMs used in this study are REMA strips and a Ziyuan-3 (ZY-3) DEM which are generated from stereo satellite images of WorldView (Anderson and Marchisio, 2012) and ZY-3 (Wang et al., 2014), respectively (Table A1). Both are formed by the along-track stereo mechanism (Li, 1998).*"

**L135-149: What is the motivation behind the three different time periods (ZY-3: 2014, REMA: 2016/17 and 2020/21) that you use? Is part of the goal to also compare ZY-3 with REMA? And why do you only use ZY-2 for year 2014 now that it has a better coverage of the rifts? I miss a clearer justification of the study period and of the use of different satellite products.**

Response:

We added the following at the end of the paragraph: "*The ZY-3 DEM of 2014 has a full coverage of T1 and T2 without performing sub-DEM integration and can provide a reference for comparison with the adjusted sub-DEMs of REMA. The combined ZY-3 and REMA sub-DEMs form a seven-year long time series from 2014 to 2021 with a time interval of 3-4 years, which can be used to analyze mélange elevation and volume changes.*"

**L147-148: Why do you present rift T2 here in the Methods section and in Fig. 3 when you do not present any results from the rift? I would suggest that you add a similar analysis of T2 as you have done for T1. That could potentially also strengthen your manuscript with regards to melange dynamics.**

Response:

Please see the response to the relevant Major Comment.

**L150: You mention that you use ICESat-2 data from 2019-2021 to co-register the 2014 ZY-3 DEM. How can you justify this 5-7 years gap in between acquisition times?**

Response:

We added a sentence: "*To ensure that the GCPs are "stable" during the timespan between the ZY-3 DEM (REMA sub-DEMs) and ICESat-2, these GCPs are further required to be on grounded features with a low velocity (< 10-20 m y$^{-1}$). In a special case we also select GCPs on the ice shelf where both the DEM images and ICESat-2 data were acquired within one day.*"

**L135-155: I would suggest that you make a "Data" section between the Introduction and the Methods section, and that you move all of this to the new Data section.**

Response:

Yes, we did it accordingly (Section 2.1).

**L196-198: How do you use ICESat-2 to determine the GCP elevations for the REMA 2016/17 DEMs? Do you just assume that the elevations at those points are constant in time? And if so, does that assumption hold? And what impact does it have on the results?**
Response:
Please see the response to "L150" above.

**L199: "on the flowing ice of the ice shelf,…" → "on the floating part of the ice shelf,…"**
Response:
It is so changed.

**L226-227: How is that co-registration performed How is ICESat-2 used given the large time difference?**
Response:
The sentence is changed to: "*The DEM is co-registered to the ICESat-2 ATL06 data of 2019 through a bundle adjustment procedure (McGlone et al., 2004; Li et al., 1998 and 2017b) using GCPs that are selected from "stable" features in the same way for those used in REMA sub-DEM co-registration. An elevation accuracy of 0.30 m is achieved.*"

**L233-242: As I understand the text here, you only validate the 2020/21 REMA DEM and not the ZY-3 2014 DEM and the 2016/17 REMA DEM. So how about those two other DEMs? How are they validated?**
Response:
Here validation is a process where we use "ground truth" (ICESat-2) to verify the accuracy of a set of DEMs. This external assessment requires that the ground truth data and DEMs to be in the same place and cover the same period. In this way we prove that the 2020-2021 REMA DEM is of a high accuracy, 0.09 m on surface and 0.18 m on mélange.

On the other hand, the ground truth data of ICESat-2 do not cover the same periods of the 2016-2017 REMA DEM and the 2014 ZY-3 DEM. However, all three DEM sets use the "stable" grounded features on Berkner Island to Coats Land as GCPs, so that the accuracy of the grounded part of the DEMs are thus validated in a manner of internal assessment (Fig. 7, 0.14 – 0.30 m).

As explained in the manuscript, we use an internal assessment of inconsistencies or residuals at the TPs to estimate the accuracy of the floating part of these two DEMs. The residuals are in average 0.14 m for the 2016-2017 REMA DEM (Fig. 7) and 0.30 m for the ZY-3 DEM as assessed by using the bundle adjustment. They are at the same level of 0.18 m, the internal accuracy assessment for the 2020-2021 REMA DEM.

Therefore, we trust that the 2016-2017 REMA DEM and the 2014 ZY-3 DEM have the same quality as the 2020-2021 REMA DEM.

**Sect. 3.1: In my opinion this section belongs in the Methods section and not in the Results.**
Response:
This section (3.1) reports the results and performance of the proposed MDAM model. Then Section 3.2 reports the mélange dynamics application results. To maintain the logic flow, we hope that you would not mind that we keep this section in the Results.

**L254-257: "… of 2014 where transects are spaced at…" → "… of 2014 with transects spaced at…"**
Response:
Changed accordingly.

**L264: Typo: I doubt that the rift is 4713.17 km long → it is probably 47.1317 km long.**
Response:
Thanks. It is corrected.

**L268-270: Interesting that the sea-ward side is mostly higher! Can you elaborate/speculate a bit on what the cause behind that could be?**
Response:
The average differences are not very large. We found that the landward wall produces more seracs than the seaward wall. This may cut the higher part of the undulation on the landward flank, so the remaining surface gets lower. We need more measurements and modelling efforts to prove this speculation.

**L270-272: What does "The phenomenon" refer to in this sentence? Does it refer to the higher sea-ward side? And if so, what do they hypothesize as a reasoning for that?**
Response:
We changed the sentence: "*This phenomenon of the greater heights of seaward rift lips based on the precise measurements from the bias-corrected sub-DEMs is consistent with the results for some rifts on other Antarctic ice shelves presented in Walker et al. (2019).*" We leave the reasoning for future paper(s) (see response above).

**L277: Typo: YZ-3 should be ZY-3**
Response:
Done.

**L281: "We believe that this section of the rift…" → "We believe that this central section of the rift…"**
Response:
Done.

**L289-298: "To compute the corresponding… …in the mature melange section." Should be in the Methods section instead of in the results section.**

Response:

This part is moved to the Method section.

**L305: I am not sure that I agree that the accuracy of all DEMs in general is 0.09 m as you only validate this for the 2020/21 DEMs.**

Response:

Agree. The sentence is changed to: "*The bias-corrected multi-satellite DEM time series achieved an accuracy of 0.09 m for the 2020-2021 REMA DEM assessed by ICESat-2 validation, and 0.18 m for the 2016-2017 REMA DEM to 0.3 m for the 2014 ZY-3 DEM, respectively, estimated through the error propagation.*"

**Figure 1: What small-coverage DEMs is it that you show on Figure 1a? Are those the REMA strips? I also miss an indication of the size of the strips relative to the ice shelf, so some sort of scale.**

Response:

We added the suggested scale information: "*Figure 1. (a) High-resolution small-coverage DEMs (e.g., REMA strips of ~16~18 km×~110~120 km) are unified and integrated for accurate 3D rift structural and mélange dynamic monitoring in a large Antarctic ice shelf environment (e.g., ~165 km wide Filchner Ice Shelf) ……*"

**Figure 3: How come there are GCPs on the ice shelf itself in panel b)? In the manuscript and Fig. 1 you mention that those are only on the grounded parts. Furthermore, the gery ZY-3 outlines in panel a) are very difficult to see, so I would choose a different color for those. I think that there is a typo in the colorbar of both panels, they say that the elevation go from -10 to -510. Finally, I would mark T1 and T2 in both panels.**

Response:

In the new Data section, we added: "*…… In a special case we also select GCPs on the ice shelf where both the DEM images and ICESat-2 data were acquired within one day.*" Similar sentence is also added in the caption of Fig. 3.

In the following revised Fig. 3, please see that the ZY-3 outlines are changed to darker and thicker lines. "-500 m" is changed to 500 m. T1 and T2 are also marked in (b).

[Figure]

**Figure 9: I would prefer to see these DEMs in 2D rather than 3D, as this 3D view does not allow for seeing the full depth of the rift. Secondly, I would consider adjusting the colorbar to make the ice shelf surface more visible without losing information from within the rift. Lastly, I miss a scale on the figures for reference, which will for instance make it easier to see by how much the rift has opened during the study period.**

Response:

Now we provide both 2D DEMs and 3D DEMs.

[Figure]

**Figure 9. Multi-satellite DEM time series of two large rifts (T1 and T2) on Filchner Ice Shelf from 2014 to 2021 in 3D: (a) and (d) are reconstructed ZY-3 DEM of 2014, (b) and (e) are bias-corrected REMA DEMs of 2016-2017, and (c) and (f) are bias-corrected REMA DEMs of 2020-2021. The boxes in (b) and (e) indicate two rift sections where details of 3D structure and mélange are presented in Fig. 10.**

[Figure]

**Figure A2.** Multi-satellite DEM time series of the two large rifts (T1 and T2) on Filchner Ice Shelf from 2014 to 2021 in 2D: (a) and (d) are reconstructed ZY-3 DEM of 2014, (b) and (e) are bias-corrected REMA DEMs of 2016-2017, and (c) and (f) are bias-corrected REMA DEMs of 2020-2021. The boxes in (b) and (e) indicates two rift sections of the rift where details of 3D structure and mélange are presented in Fig. 10.

**Figure 10: In panel c) and d) I would consider changing the x-axis to "Distance from the ice front" as that is an important parameter in terms of the calving risk.**

Response:

It is changed to "*Distance from the ice front of 2014*". We also added Figs. 10e-10h for T2.

[Figure]

[Figure]

**Figure 10. 3D sectional structure and mélange: (a) and (e) are sectional shaded relief maps of REMA DEMs of 2016-2017 (January 4, 2017 and October 25, 2016) indicated by the boxes in Fig. 9; (b) and (f) are WorldView images (0.5 m resolution) of February 25, 2016 and March 2, 2016; (c) and (g) are elevation profiles along the pink lines in (a) and (e), and 3D rift and mélange structure parameters; (d) and (h) are rift and mélange changes along the profiles from 2014 to 2021 (2014 in blue, 2016-2017 in green, and 2020-2021 in red). Elevation displayed in (a) and (e) are exaggerated by 10 times.**

**Figure A1: It is very difficult to see what is what on this figure. People that are not used to seeing stereo pairs plotted in this way will most likely not be able to tell that it is a cavity as opposed to a melange. Could you maybe show a stereo pair of a melange for comparison or add a colormap and colorbar which makes the cavity more visible?**

Response:

We revised the figure by adding a 2D color elevation map where the cavity is represented in blue color. Hope this makes the cavity visible. We will also revise the manuscript to make it clear.

[Figure]

**Figure A3. Anaglyph stereo pair of 2014 ZY-3 satellite images (a) and reconstructed ZY-3 DEM (b) showing cavities close to the west tip of rift T1.**

**Data availability: How about the ZY-3 data? And will the DEM time series generated here also become publicly available somewhere?**

Response:

Yes, we will make the ZY-3 DEM publicly available, along with the corrected REMA DEMs.

**Code availability:**

**Will the MDAM code become publicly available? This paper is heavily based on the development of a new method, and I therefore think that it is important that that new method is also publicly available to the community.**

Response:

The MDAM code (including TPs and GCPs) is now available at GitHub (https://github.com/menglianxia/MDAM).

The produced DEMs (including adjusted REMA DEMs of 2016-2017 and 2020-2021, and ZY-3 DEM of 2014) are available at https://doi.org/10.5281/zenodo.15260323 (Xia et al. 2025).

Response:

These references are cited in right places of the manuscript.

**Referee #2**

**Dear Menglian Xia and co-authors,**

**The manuscript "Building multi-satellite DEM time series for insight into mélange inside large rifts in Antarctica" by Xia et al. presents a novel approach to monitor rifts' infill on the Filchner-Ronne ice shelf using observations from different satellites at high resolution Understanding variations in rift infill is crucial for assessing rift dynamics, which ultimately influence calving events. I commend the authors for their creative approach, thorough analysis, and well-designed, highly informative figures. Overall, I strongly believe this paper would be a valuable addition to the literature, and its scope and format make it well-suited for publication in The Cryosphere. Below, I provide a general comment along with more specific, line-by-line suggestions that I hope will further strengthen the manuscript.**

**GENERAL COMMENT:**

**I admit that DEM processing is not my direct area of expertise, but the logic of the technical part seems sound and appropriate for a reader, like me, who is not necessarily familiar with technicalities of DEM analysis. However, I believe this manuscript would benefit from a few improvements to enhance readability and impact. In some sections, the text becomes somewhat difficult to follow, partly due to the frequent use of acronyms and the listing of numbers with excessive significant figures. At times, this made it challenging to stay engaged with the flow of the manuscript. I recommend streamlining certain parts, as indicated in my line-by-line comments, to improve clarity and coherence.**

Response:

Thanks for the suggestions. The manuscript is revised accordingly (see following responses).

**More importantly, I find that this manuscript lacks a thorough discussion, a comparison with existing literature, and a clear background motivation. I was surprised to see that it does not include a dedicated discussion section, where I had expected to find these elements. To strengthen the manuscript's impact, I recommend contextualizing the observational results within the broader framework of rift monitoring and their implications for ice shelf and glacier dynamics, while also drawing comparisons with existing studies.**

Response:

We addressed the issues in two parts.

1) We added substantial text in Introduction to give a clear background motivation, comparison with existing studies, and context to this study … (also see following responses)

2) We added a Discussion section to address the remaining points:

*"4 Discussion*

*The developed multi-temporal DEM adjustment model MDAM has shown its effectiveness in removing biases between adjacent multi-satellite sub-DEMs and establishing a unified and integrated DEM time series. We demonstrate the full 3D mapping capability of characterizing rift structure (rift lips, pre-mélange cavities, mélange seracs, etc.) and estimating dynamic mélange volume changes from the ICESat-2 controlled DEM time series, extending from rift topography and mélange thickness estimated along ground tracks from ICESat and ICESat-2 measurements in previous studies (Fricker et al., 2005; Walker et al., 2021). Such high-resolution mélange dynamic observations allow us to understand the mélange movement inside a closed environment of large Antarctic rifts, and to further study its role in rapid rift propagation and iceberg calving. To make this model working in a more dynamic open ocean environment, such as Pine Island and Thwaites ice shelves, modifications need to be carried out to address the rapid calving process with incoherent mélange changes in ice shelf front.*

*A thick mélange layer can effectively "freeze" a rift, enabling mechanical stress transmission between its flanks and ultimately suppressing rift propagation (Rignot and MacAyeal, 1998; Larour et al, 2021). In this study, we show that seracs appearing on the mélange surface are a part of the infill inside rifts. They are formed from partial collapsing of rift flanks. We propose that the surface part of the bottom rift wall is first excavated through interactions by tides (Padman et al., 2002 and 2008), melting caused by intrusion of warmer seawater into the rift (Poinelli et al., 2023a) and other factors. This, in turn, causes the collapse of the upper part of the rift flank due to the removal of the bottom support. This process, like coastal bluff erosion, repeats itself and becomes one of the mechanisms that widen the rift and increase the mélange volume. We find that the increased mélange volume in a rift promotes its widening rate, and may further increase ice front calving and effect the ice shelf stability.*

*Although the iceberg calved from the shelf front in 1986 is mostly located inside the Passive Shelf Ice (PSI) area (Fig. A1), meaning no significant buttress reduction from the ice shelf (Doake et al., 1998; Fürst et al., 2016), the combined rifts of T1 and T2 propagated rapidly recently and have already covered ~58% the ice shelf laterally from Coats Land to Berkner Island. As warm water has recently been observed near Berkner Island (Davis et al., 2022), T1*

*and T2 have the potential to trigger a larger calving beyond the PSI boundary, which may be caused by a destabilization mechanism like that proposed for the Larsen C Ice Shelf (Poinelli et al., 2023a and 2023b)."*

**While rifts and calving are natural processes in the life cycle of ice shelves, the calving of their seaward-most extensions does not necessarily lead to significant upstream glacier acceleration. I have not performed the calculations myself, but rifts T1 and T2 on the Filchner-Ronne Ice Shelf (FRIS) appear to lie beyond the 'compressive arch' (Doake, 1998), suggesting they may not directly precondition the ice shelf for collapse—similar to the 'passive portion' described by Fürst et al. (2016). This raises a key question: if these rifts are not an immediate destabilizing factor for FRIS, why is their study important?**

Response:

We added the following text in the Discussion: "……*Although the iceberg calved from the shelf front in 1986 is mostly located inside the Passive Shelf Ice (PSI) area (Fig. A1), meaning no significant buttress reduction from the ice shelf (Doake et al., 1998; Fürst et al., 2016), the combined rifts of T1 and T2 propagated rapidly recently and have already covered ~58% the ice shelf laterally from Coats Land to Berkner Island. As warm water has recently been observed near Berkner Island (Davis et al., 2022), T1 and T2 have the potential to trigger a larger calving beyond the PSI boundary, which may be caused by a destabilization mechanism like that proposed for the Larsen C Ice Shelf (Poinelli et al., 2023a and 2023b).*" Also see the following figure.

[Figure]

**Figure A1. Rifts of T1 and T2 (pink) in 2021, Grand Chasm (blue) in 1986; PSI boundary (green) (Fürst et al., 2016); ice bergs calved in 1986 (white); background is a Landsat image of February 11, 2021.**

**This is, of course, a provocative question, and I fully agree with the authors that rifts play a critical role. Future calving triggered by T1 and T2 and the resulting ice front retreat may expose the ice shelf to increased warm water intrusion, a process simulated for Larsen C and supported by theoretical work (Poinelli, 2023a,b). In the context of global warming, this is particularly relevant for FRIS, as warm water has recently been observed near Berkner Island (e.g., Davis, 2022). Such changes could ultimately lead to destabilization mechanisms like those proposed for Larsen C. I encourage the authors to incorporate these considerations to provide a stronger and more comprehensive discussion of their findings.**

Response:

Thanks for suggestion. They are added into Discussion (see the response above).

**In the introduction, there is a strong focus on technical studies related to altimetric and stereo mapping of rifts, but the broader scientific goal and the significance of this region of Antarctica seem underdeveloped. For a paper in The Cryosphere, I would expect a more robust scientific background that clearly establishes the relevance and utility of this work.**

Response:

We added the following text in Introduction to address it.

*"…… Furthermore, mélange (a mixture of shelf ice, snow, sea ice and water) changes related to ice shelf fracturing in Antarctica and glacier calving in Greenland are studied (Rignot and MacAyeal, 1998; Larour et al., 2004; Cassotto et al., 2021). Specifically, mélange thickness reduction is observed during rift widening on Amery Ice shelf, Ronne Ice Shelf and Larsen C Ice Shelf (Fricker et al., 2005; Walker et al., 2021; Larour et al., 2021). Modelling results indicate that mélange may have the capability of transmitting stresses across rift flanks and thus, influences rift propagation and shelf front calving (Larour et al., 2004 and 2021). However, there has been a lack of large-scale high-resolution observations of 3D rift structural changes and mélange dynamics in a sustained long period to support such a conclusion. This, in turn, hinders our understanding of the role of mélange in ice shelf retreating and the mechanism of ice shelf stability weakening."*

*"…… The validated MDAM is applied in the Filchner Ice Shelf to establish a cross-shelf DEM time series from 2014-2021. The Filchner Ice Shelf (Fig. 1a) is one of the dual ice shelves of Filchner-Ronne Ice Shelf that is the second largest in Antarctica. Berkner Island and Coats Land are located at western and eastern margins of the ice shelf, respectively. The largest calving event recorded for this ice shelf, with an area loss of 11500 km², occurred in 1986 due to a rapid propagation of a prominent rift known as "Grand Chasm" (Ferrigno and Gould, 1987), producing three giant icebergs that greatly impacted the circulation and hydrography in the nearby ice shelf-ocean system (Grosfeld et al., 2001) (Fig. A1). Recently, two large rifts, T1 and T2 (Fig. 3), are detected in a Landsat satellite image in 1996 and an ARGON satellite*

*image in 1963, respectively (Li et al., 2017a; Walker and Gardner, 2019; Lv et al., 2022). These rifts have a combined length of ~100 km and exhibit a similar propagation pattern as Grand Chasm……"*

**One crucial aspect that appears to be missing is the role of ice mélange variation in modulating rift dynamics. A thick mélange layer can effectively "freeze" a rift, enabling mechanical stress transmission between its flanks and ultimately suppressing rift propagation (Rignot, 1998; Larour, 2021). This process is fundamental to understanding rift evolution and should be better integrated into the manuscript's scientific framing. I may have overlooked this point, and I apologize if it was addressed, but I encourage the authors to clarify and emphasize its importance.**

Response:

We added this sentence in Introduction: "*Modelling results indicate that mélange may have the capability of transmitting stresses across rift flanks and thus, influences rift propagation and shelf front calving (Larour et al., 2004 and 2021).*"

We further added the following text specifically in Discussion, but also make a link to the "volume" change results from this study:

"

*A thick mélange layer can effectively "freeze" a rift, enabling mechanical stress transmission between its flanks and ultimately suppressing rift propagation (Rignot and MacAyeal, 1998; Larour et al., 2021). In this study, we show that seracs appearing on the mélange surface are a part of the infill inside rifts. They are formed from partial collapsing of rift flanks. We propose that the surface part of the bottom rift wall is first excavated through interactions by tides (Padman et al., 2002 and 2008), melting caused by intrusion of warmer sea water into the rift (Poinelli et al., 2023a) and other factors. This, in turn, causes the collapse of the upper part of the rift flank due to the removal of the bottom support. This process, like coastal bluff erosion, repeats itself and becomes one of the mechanisms that widen the rift and increase the mélange volume. We find that the increased mélange volume in a rift promotes its widening rate, and may further increase ice front calving and effect the ice shelf stability.*"

**SPECIFIC COMMENTS**

**Line 13: The connection between calving and ice mélange is not clearly established in the manuscript. Why is the observed "significant gap" in mélange dynamics particularly relevant to calving processes? Ice mélange, when sufficiently thick, can bond rift flanks together, suppressing rift widening and propagation. Clarifying this link would strengthen the manuscript's argument and better highlight the importance of studying mélange dynamics in the context of ice shelf stability.**

Response:

We added the following text in Introduction to clarify the link between mélange and calving: "......*Furthermore, mélange (a mixture of shelf ice, snow, sea ice and water) changes related to ice shelf fracturing in Antarctica and glacier calving in Greenland are studied (Rignot and MacAyeal, 1998; Larour et al., 2004; Cassotto et al., 2021). Specifically, mélange thickness reduction is observed during rift widening on Amery Ice shelf, Ronne Ice Shelf and Larsen C Ice Shelf (Fricker et al., 2005; Walker et al., 2021; Larour et al., 2021). Modelling results indicate that mélange may have the capability of transmitting stresses across rift flanks and thus, influences rift propagation and shelf front calving (Larour et al., 2004 and 2021). However, there has been a lack of large-scale high-resolution observations of 3D rift structural changes and mélange dynamics in a sustained long period to support such a conclusion. This, in turn, hinders our understanding of the role of mélange in ice shelf retreating and the mechanism of ice shelf stability weakening.*"

**Line 20: Mélange volume expansion is directly correlated with rift widening, but its relevance in the context of ice sheet modeling is unclear. Can we speculate on causality— does mélange volume drive rift widening, or is it simply a byproduct? Since mélange volume necessarily increases as the rift widens due to greater exposure of open ocean to surface heat loss, a more informative parameter to monitor might be mélange thickness relative to ice shelf thickness. This ratio is crucial in determining how stress is transmitted between rift flanks (Larour, 2021).**

Response:

Mélange thickness change is an important indicator. It is now reviewed in Introduction (see above response). We believe that volume is another important indicator that is related to both rift widening and supply of shelf ice to adjust the infill into the rift. We added how this process might occur in Discussion: "*A thick mélange layer can effectively "freeze" a rift, enabling mechanical stress transmission between its flanks and ultimately suppressing rift propagation (Rignot and MacAyeal, 1998; Larour et al., 2021). In this study, we show that seracs appearing on the mélange surface are a part of the infill inside rifts. They are formed from partial collapsing of rift flanks. We propose that the surface part of the bottom rift wall is first excavated through interactions by tides (Padman et al., 2002 and 2008), melting caused by intrusion of warmer sea water into the rift (Poinelli 2023a) and other factors. This, in turn, causes the collapse of the upper part of the rift flank due to the removal of the bottom support. This process, like coastal bluff erosion, repeats itself and becomes one of the mechanisms that widen the rift and increase the mélange volume. We find that the increased mélange volume in a rift promotes its widening rate, and may further increase ice front calving and effect the ice shelf stability.*"

**Line 29: This line is a bit confusing, mass discharge across the GL is the cause of sea level rise, but what does 'lost ice mass enters' mean?**

Response:

We deleted the first half of the sentence. It is changed to: "…… *The lost ice mass enters the Southern Ocean from ice shelves mainly through two processes, namely shelf front calving and basal melting, each accounting for ~50% (Depoorter et al., 2013; Liu et al., 2015; Smith et al., 2020; Greene et al., 2022).*"

**Line 39: Like my comment in the abstract, I don't fully see the link between calving and ice mélange? What does 'the importance … on studying mélange … are fully recognized' mean?**

Response:

We changed this part of the section to strengthen the link: "*…… Furthermore, mélange (a mixture of shelf ice, snow, sea ice and water) changes related to ice shelf fracturing in Antarctica and glacier calving in Greenland are studied (Rignot and MacAyeal, 1998; Larour et al., 2004; Cassotto et al., 2021). Specifically, mélange thickness reduction is observed during rift widening on Amery Ice shelf, Ronne Ice Shelf and Larsen C Ice Shelf (Fricker et al., 2005; Walker et al., 2021; Larour et al., 2021). Modelling results indicate that mélange may have the capability of transmitting stresses across rift flanks and thus, influences rift propagation and shelf front calving (Larour et al., 2004 and 2021). However, there has been a lack of large-scale high-resolution observations of 3D rift structural changes and mélange dynamics in a sustained long period to support such a conclusion. This, in turn, hinders our understanding of the role of mélange in ice shelf retreating and the mechanism of ice shelf stability weakening.*"

**Figure 1: Really nice figure.**
**Line 67: What do you mean with 'tie point'?**
Response:

We added the text to refer it to the detailed TP description in Section 2.3: "*We introduce tie points (TPs, see detailed description in Section 2.3) ……*"

**Line 76: This sentence is a bit confusing.**
Response:

We revised the sentence to clarify the point: "*Finally, we demonstrate the observed mélange changes and estimated volumetric changes in relation to rift widening as the rifts advect toward the ice shelf front during the study period.*"

**Line 124: This is good example of the scientific motivation of this study. Perhaps this should be included in the introduction and not in methodology.**
Response:

Yes, this part is now moved to Introduction.

**Line 157,176: Acronyms in the title are hard to follow ad makes the reading experience confusing.**

Response:

They are now spelled out in the titles.

**Figure 8-9: Great figures! Congrats!**

**Line 264: I appreciate the precision in these values but is it necessary? This may be a question of personal preference, but I find it a bit confusing to follow these numbers. These details are also reported in Table A3 no? So why don't discuss the orders of magnitude of these changes?**

Response:

We now keep the digits to meter for all numbers throughout the manuscript. We also give the orders of magnitude of these changes: "…… *we found that the rift T1 propagated consistently during the period, resulting in an accelerated widening by 647 m during 2014-2017 (44 %, 227 m y$^{-1}$) and 1,109 m during 2016-2021 (53 %, 268 m y$^{-1}$). Correspondingly, the rift has lengthened by 2,393 m during 2014-2017 (5%, 840 m y$^{-1}$), but only 1,082 m during 2016-2021 (2%, 262 m y$^{-1}$).*"

Here are data for both T1 and T2 and have been updated in the manuscript for T2:

| | Time | Rift T1 | Rift T2 |
|---|---|---|---|
| Length (m) | 2014 | 47132 | 48324 |
| | 2017 | 49525 | 50025 |
| | 2021 | 50608 | 51692 |
| Lengthening (m & m/y) | 2014-2017 | 2393 m (5%, 840 m/y) | 1701 m (%4, 597 m/y) |
| | 2016-2021 | 1083 m (2%, 262 m/y) | 166 m (%3, 406 m/y) |
| Width (m) | 2014 | 1457 | 1532 |
| | 2017 | 2103 | 1944 |
| | 2021 | 3212 | 2778 |
| Widening (m & m/y) | 2014-2017 | 647 m (44%, 226.95 m/y) | 412 m (27%, 145 m/y) |
| | 2016-2021 | 1109 m (53%, 268.36 m/y) | 834 m (43%, 203 m/y) |

**Line 282: What is a 'cavity' in this context? Please specify.**

Response:

We changed it to: "*……More specifically, we found cavities that are freshly opened during the fracturing process in the unmatured mélange section close to the west tip of T1 (Fig. 11a). These pre-mélange cavities are verified in ZY-3 stereo images and reconstructed ZY-3 DEM (Fig. A3) ……*"

**Line 290: Volumetric change is a good parameter to monitor, but it would also be interesting to see an estimation of mélange thinning, which is relevant to ice sheet**

**modeling. The observed decreased in ice mélange elevation most likely means that this layer has thinned. This is extremely important as the stress propagation between flanks may be compromised (Larour 2004).**

Response:

Here we added: "*The estimated change in mélange thickness H is relevant to ice sheet modeling. The observed mélange thinning may indicate that the stress propagation between flanks may be compromised (Larour et al., 2004).*"

**Line 300: What do you mean with 'newly calved'? I may have missed this, but I thought the ice shelf has not calved during this period. Do you perhaps mean that the rift has widened due to partial collapse of its flanks? If so, can you speculate about what may have caused it? Mélange thinning may be the answer itself, and I agree that it is hard to point at a cause. Setting up a discussion on this would be very important.**

Response:

We changed it to "*…… newly vacated space due to partial collapse of rift flank*".

We added a paragraph in Discussion to speculate its cause: "*A thick mélange layer can effectively "freeze" a rift, enabling mechanical stress transmission between its flanks and ultimately suppressing rift propagation (Rignot and MacAyeal, 1998; Larour et al., 2021). In this study, we show that seracs appearing on the mélange surface are a part of the infill inside rifts. They are formed from partial collapsing of rift flanks. We propose that the surface part of the bottom rift wall is first excavated through interactions by tides (Padman et al., 2002 and 2008), melting caused by intrusion of warmer sea water into the rift (Poinelli et al., 2023a) and other factors. This, in turn, causes the collapse of the upper part of the rift flank due to the removal of the bottom support. This process, like coastal bluff erosion, repeats itself and becomes one of the mechanisms that widen the rift and increase the mélange volume. We find that the increased mélange volume in a rift promotes its widening rate, and may further increase ice front calving and effect the ice shelf stability.*"

**Line 302: I was surprised to see that the manuscript does not include a discussion section. What are the strengths of this novel approach? How does this compare to previous studies that employed ICESat2? for example Walker 2021, Fricker 2005, …? Why is monitoring of these rifts important? Can you extend this novel processing techniques to other highly fractured areas of Antarctica (Larsen C, Totten, Brunt, Amery?)**

Response:

We added a Discussion section to address these points:

*"4 Discussion*

*The developed multi-temporal DEM adjustment model MDAM has shown its effectiveness in removing biases between adjacent multi-satellite sub-DEMs and establishing a unified and*

*integrated DEM time series. We demonstrate the full 3D mapping capability of characterizing rift structure (rift lips, pre-mélange cavities, mélange seracs, etc.) and estimating dynamic mélange volume changes from the ICESat-2 controlled DEM time series, extending from rift topography and mélange thickness estimated along ground tracks from ICESat and ICESat-2 measurements in previous studies (Fricker et al., 2005; Walker et al., 2021). Such high-resolution mélange dynamic observations allow us to understand the mélange movement inside a closed environment of large Antarctic rifts, and to further study its role in rapid rift propagation and iceberg calving. To make this model working in a more dynamic open ocean environment, such as Pine Island and Thwaites ice shelves, modifications need to be carried out to address the rapid calving process with incoherent mélange changes in ice shelf front.*

*A thick mélange layer can effectively "freeze" a rift, enabling mechanical stress transmission between its flanks and ultimately suppressing rift propagation (Rignot and MacAyeal, 1998; Larour et al., 2021). In this study, we show that seracs appearing on the mélange surface are a part of the infill inside rifts. They are formed from partial collapsing of rift flanks. We propose that the surface part of the bottom rift wall is first excavated through interactions by tides (Padman et al., 2002 and 2008), melting caused by intrusion of warmer seawater into the rift (Poinelli et al., 2023a) and other factors. This, in turn, causes the collapse of the upper part of the rift flank due to the removal of the bottom support. This process, like coastal bluff erosion, repeats itself and becomes one of the mechanisms that widen the rift and increase the mélange volume. We find that the increased mélange volume in a rift promotes its widening rate, and may further increase ice front calving and effect the ice shelf stability.*

*Although the iceberg calved from the shelf front in 1986 is mostly located inside the Passive Shelf Ice (PSI) area (Fig. A1), meaning no significant buttress reduction from the ice shelf (Doake et al., 1998; Fürst et al., 2016), the combined rifts of T1 and T2 propagated rapidly recently and have already covered ~58% the ice shelf laterally from Coats Land to Berkner Island. As warm water has recently been observed near Berkner Island (Davis et al., 2022), T1 and T2 have the potential to trigger a larger calving beyond the PSI boundary, which may be caused by a destabilization mechanism like that proposed for the Larsen C Ice Shelf (Poinelli et al., 2023a and 2023b)."*

**Line 307: I may have missed it, but is there a similar analysis applicable to T2? The manuscript presents these rifts as a pair, yet the analysis appears to be restricted to T1, which seems inconsistent. To be clear, I am not suggesting additional analysis, but the rationale for focusing solely on T1 should be explicitly stated, even if it is due to data limitations.**

Response:

As suggested also by Reviewer 1, we added corresponding analysis for T2 throughout the manuscript. The following are the revised figures and tables. The details have been revised in the manuscript.

[Figure]

**Figure 9. Multi-satellite DEM time series of the large rifts of T1 and T2 on Filchner Ice Shelf from 2014 to 2021: (a) and (d) reconstructed ZY-3 DEM of 2014, (b) and (e) bias-corrected REMA DEMs of 2016-2017, and (c) and (f) bias-corrected REMA DEMs of 2020-2021. The boxes in (b) and (e) indicate the corresponding sections of T1 and T2 where details of 3D structures and mélange are presented in Fig. 10.**

[Figure]

[Figure]

**Figure 10. 3D sectional structure and mélange: (a) and (e) are sectional shaded relief maps of REMA DEMs of 2016-2017 (January 4, 2017 and October 25, 2016) indicated by the boxes in Fig. 9; (b) and (f) are WorldView images (0.5 m resolution) of February 25, 2016 and March 2, 2016; (c) and (g) are elevation profiles along the pink lines in (a) and (e), and 3D rift and mélange structure parameters; (d) and (h) are rift and mélange changes along the profiles from 2014 to 2021 (2014 in blue, 2016-2017 in green, and 2020-2021 in red). Elevation displayed in (a) and (e) are exaggerated by 10 times.**

[Figure]

**Figure 11. Thickness (a and c) and volume (b and d) of ice mélange inside the rifts T1 and T2, respectively, from the multi-satellite DEM series from 2014 to 2021. Average thickness and volume of each transect that are separated every 1 km along the rift centerlines are illustrated from west tip to east tip**

[revised manuscript text omitted]

**Fricker 2005, ICESat's new perspective on ice shelf rifts: The vertical dimension, GRL**

Response:

The above references are all cited in places, as suggested, in the manuscript.

---

## Referee Report (RR1)

**Review of "Building multi-satellite DEM time series for insight into mélange inside large rifts in Antarctica"**

**Overview:**

I appreciate that the authors have done their best to address the comments from both me and the other reviewer. I do, however, still have a few concerns with regards to the novelty and impact of the paper. A DEM-registration method, as MDAM presented here, is in itself not novel, as similar approaches already exist. Therefore, I think that the paper requires a stronger analysis and discussion of the observed melange dynamics to be impactful enough to be published. Please see my comments below for a more elaborate suggestion on how to achieve that. All line numbers refer to the track-changes version of the manuscript.

**Minor comments:**

MDAM method:

The authors present their DEM-registration method MDAM as being the only existing method which can be used to study 3D melange dynamics, which I would disagree with. While existing methods (e.g., Shean at al. 2019 and Zinck et al., 2023) do not use their DEM-registration methods to study melange dynamics, it does not mean that they cannot be used for that. I do find the MDAM method to be a good addition to the existing methods, but it should be clearly acknowledged that other methods do exist. Furthermore, I miss a discussion of the limitations of the MDAM method. For instance, it seems that both TPs and GCPs are manually chosen, which makes the MDAM method difficult to upscale to larger regions as it would require a significant amount of manual labour. Secondly, the authors mention that the method requires adjustment to be applied to areas such as Pine Island and Thwaites. What adjustments would that be? And why?

Melange dynamics analysis and discussion:

You discuss a few causes behind the observed rift widenings. However, I am still left with an impression that I do not know what caused the widening of the rifts. Why did the melanges not freeze the rifts? How do you expect the rifts to continue their development? Is there anything to learn from the evolution of rift T2, which seems to be two rifts approaching each other and potentially merging? I think that the manuscript could benefit from a more detailed discussion on the impacts of the results. What have we learned from these results which we did not already know? Having a more detailed discussion on that would increase the impact of the paper, its findings, and also on the MDAM method itself.

Results/Methods structure:

The restructuring of the Data and Methods section has improved the readability of the paper greatly. However, there are still Methods elements present in the first section of the Results. In my opinion most of section "3.1 Bias correction and adjusted DEM time series" belongs in the Methods part of the paper and not in the Results, as it describes how you generate the final DEM timeseries. You could keep the figures in the Results, including a description of the improved uncertainty after the bias-correction. However, the sentences describing how you calculate different components (i.e. assigning different weights to TPs in L242-249, the validation method of the bias correction and the data used therefore in L283-291, etc.) should be moved to the Methods section in my opinion. You could also choose to move the entire section to the Methods.

**Specific comments:**

L40-45: I would suggest rephrasing it to something like this: "Mélanges inside rifts, which consist of shelf ice, snow, sea ice, and water, have been investigated in relation to ice shelf fracturing in Antarctica and glacier calving in Greenland (References). Specifically, reductions in mélange thickness have been observed during rift widening on the Amery, Ronne, and Larsen C ice shelves (References). Furthermore, modelling results indicate that a mélange may... "

L63-64: This is where I suggest you to be careful with how you present the MDAM method as this sentence still reads as if there is a general lack of methods handling heterogenous offsets between individual DEMs. I, therefore, suggest you to acknowledge that other methods already exist, but all with a different goal in mind, than the 3D melange dynamics presented in this paper.

L190-204: Are the GCPs manually chosen? Or are they "computationally" selected based on the mentioned criteria? Please mention what you did in a revised manuscript.

L210-224: Is it correctly understood that the TPs are manually selected? If so, please state that explicitly in the manuscript.

L236-237: The comment about the observed melange thinning seems misplaced here in the Methods section. I would suggest to move it to the Results or Discussion.

L241-292: This is the part which I suggest that you (partly) move to the Methods section of the paper.

Table 1: I do not see the need for this table as the numbers are illustrated much better in Figure 6 and 7.

Figure 10: Please specificy in both figure and caption which is T1 and which is T2.

L321-322: I would suggest to reduce the number of digits: 47.132 km → 47.1 km, 48.324 km → 48.3 km, 1.457 km → 1.5 km, 1.532 km → 1.5 km

Figure 11: Would it maybe make more sense to use two different centerlines for T2 as it seems from Fig. A2 that T2 is maybe two separate rifts approaching eachother rather than one long rift. This would also help to avoid confusion as to why the elevation and volumne is so much higher/bigger in the middle of the central section of T2.

L375-377: I don't fully understand this sentence. What does "this model" refer to? Is that MDAM? And if so, why wouldn't it work for Pine Island and Thwaites? What adjustments would be needed?

L378-386: You mention that a thick melange layer can freeze a rift, but that is clearly not what happens at T1 and T2. Why is that? This entire paragraph in general seems loosely connected. As in, how does the fact that a thick melange layer can freeze rifts relate to the next sentence seracs inside the melange?

L385-386: Calving does not necessarily impact ice shelf stability and as you mention in the following paragraph it seems that these rifts are in areas which do not provide much buttressing. I would, therefore, suggest that you remove the ice shelf stability part of this sentence.

L385: Change Passive Shelf Ice (PSI) to passive shelf ice. There is no need for an acronym which you only use two times.

L404-405: Could you clarify how the MDAM system can bu used to study ice shelf instability and sea level rise contribution?

**Typos and grammar:**

General grammar comment: The manuscript misses quite a few definitive articles ("the") throughout the text, which makes the text difficult to follow at times. I have only marked a few of them in my review, but I recommend the authors to make use of a grammar tool or similar to locate the remaing missing definitive articles.

L28: Antarctic Ice Sheet (AIS)… → **The** Antarctic Ice Sheet (AIS)…

L49: Be aware of using the definite article "the". Change "earth surface features" to "**the Earth's** surface features"

L52-55: Be aware of using the definite article "the". Change to: "… for example, **the** 500 m ICESat (Dimarzio et al., 2007) and ICESat-2 (Shen et al., 2022) DEMs, and **the** 1000 m CryoSat-2 DEM (Slater et al., 2018). Those from optical and SAR satellite stereo mapping data are of higher resolutions and have the potential for geometric modeling and analysis of rifts and mélange features, including **the** 90 m TanDEM-X DEM (Wessel et al., 2021),

the 30 m ASTER DEM (Tachikawa et al., 2011), and especially, **the** 2 m REMA DEM (Howat et al., 2019)."

L68-69 (Figure 1 caption): Be aware of using the definite article "the". Change to "**The** LIMA mosaic ... **the** RAMP DEM ..."

L81: Change to "The validated MDAM is applied **to** the Filchner..."

L83: Change to "... is the second largest **ice shelf** in Antarctica."

L125: Be aware of using the definite article "the". Change to: "at **the** two shelf margins"

L126: Change "cover the floating ice of the ice shelf." to "cover the floating ice shelf".

L155: Change to "In **a** similar way,"

L246: Change to: "**on** average"

References:

Shean, D. E., Joughin, I. R., Dutrieux, P., Smith, B. E., and Berthier, E.: Ice shelf basal melt rates from a high-resolution digital elevation model (DEM) record for Pine Island Glacier, Antarctica, The Cryosphere, 13, 2633–2656, https://doi.org/10.5194/tc-13-2633-2019, 2019.

Zinck, A.-S. P., Wouters, B., Lambert, E., and Lhermitte, S.: Unveiling spatial variability within the Dotson Melt Channel through high-resolution basal melt rates from the Reference Elevation Model of Antarctica, The Cryosphere, 17, 3785–3801, https://doi.org/10.5194/tc-17-3785-2023, 2023.

---

## Author Response (AR2)

We appreciate the constructive comments from the editor and two reviewers (Dr. Ann-Sofie Priergaard Zinck, and Dr. Mattia Poinelli). Our manuscript is much improved by their input. In the following responses, we use "**bold**" text for comments, "non-bold" text for our responses, and "italic" for changed text in the revised manuscript.

**Referee #1**

**The manuscript "Building multi-satellite DEM time series for insight into mélange inside large rifts in Antarctica" by Xia et al. presents a novel approach to monitor rifts' infill on the Filchner-Ronne ice shelf using observations from different satellites at high resolution. This is a revised version of the manuscript. I commend the authors for the time and effort spent in addressing my long list of comments. In particular, the addition of the discussion and the analysis on the second rift strongly improves the impact of this manuscript, in my opinion.**
**Overall, I believe that this paper is now suitable for publication in the Cryosphere. Below, I provide a list of minor comments and suggestions that I hope will further strengthen the manuscript.**
Response:
Thanks for the suggestions. They are fully considered and implemented.

**L13: I still thing the "significant gap" is rather vague statement. What is missing? Why is missing? Authors point at observations, but what is the reason why these observations are missing. I understand that these are trivial points, but they need to be specified. What about something along the line of: 'high resolution observations are scarce … which then leads to an understand of …, ultimately three dimensional changes are missing'. In the introduction there is a clear indication of available satellites (IceSat, Cryosat, etc) and their inability to capture small scale rift dynamics. This is the research gap: most satellites cannot capture the details of these rifts, but the authors' approach and the use of ZY-3 is the key strength of this paper. Please make this concept clear in the abstract (Line 13).**
Response:
We changed it to "*However, large-scale, high-resolution three-dimensional (3D) observations are scarce, which leads to their inability to capture small scale rift dynamics. Ultimately, the lack of knowledge in 3D rift structural changes and mélange dynamics hinders our understanding of the role of mélange in ice shelf retreat and mechanisms underlying the weakening of ice shelf stability.*"

**L25: 'can be applied for research'. This is also rather vague. What research?**
Response:
We changed it to "*…… can be applied for quantifying ice shelf instability and ……*"

**L28: The Antarctic Ice Sheet …**

Response:

We changed it accordingly.

**L44: 'There has been a lack'. Please revise this sentence by pointing at the actual gap, scarce observations? Poor sensors? Absence of sensors with necessary accuracy?**

Response:

It is changed to: "*However, large-scale high-resolution observations are scarce. Small scale mélange textures and 3D rift dynamics cannot be captured with a necessary accuracy.*"

**L50: 500 m for ICESat …**

Response:

We changed it to "*the 500 m for ICESat ...*".

**L51: Those? Maybe observations?**

Response:

It is changed to "*The DEMs ……*"

**L75: What do you mean with 'control the overall geometry'?**

Response:

We changed it to "*Since the GCPs are selected where ICESat-2 altimetric data are available, the connected DEM time series are then geometrically controlled at a centimeter elevation accuracy (Markus et al., 2017).*"

**L85: I do not think 1996 (and 1963?) can be considered 'recent'. Also please edit to 'were detected' and not 'are'.**

Response:

We deleted "*Recently,*" and changed "*are*" to "*were*".

**Figure 9: Really nice figure.**

Response:

Thanks.

**L344: What type of modifications are needed?**

Response:

We added a sentence: "*For example, tie points between adjacent sub DEMs may be selected with decreased time intervals to minimize the uncertainty caused by the compensation using ice velocity maps.*"

**Referee #2**

**Overview:**

**I appreciate that the authors have done their best to address the comments from both me and the other reviewer. I do, however, still have a few concerns with regards to the novelty and impact of the paper. A DEM-registration method, as MDAM presented here, is in itself not novel, as similar approaches already exist. Therefore, I think that the paper requires a stronger analysis and discussion of the observed mélange dynamics to be impactful enough to be published. Please see my comments below for a more elaborate suggestion on how to achieve that. All line numbers refer to the track-changes version of the manuscript.**

**Minor comments:**

**MDAM method:**

**The authors present their DEM-registration method MDAM as being the only existing method which can be used to study 3D melange dynamics, which I would disagree with. While existing methods (e.g., Shean at al. 2019 and Zinck et al., 2023) do not use their DEM-registration methods to study melange dynamics, it does not mean that they cannot be used for that. I do find the MDAM method to be a good addition to the existing methods, but it should be clearly acknowledged that other methods do exist. Furthermore, I miss a discussion of the limitations of the MDAM method. For instance, it seems that both TPs and GCPs are manually chosen, which makes the MDAM method difficult to upscale to larger regions as it would require a significant amount of manual labour. Secondly, the authors mention that the method requires adjustment to be applied to areas such as Pine Island and Thwaites. What adjustments would that be? And why?**

Response:

In responding to this major comment and the relevant specific comments bellow, we made changes to address the above concerns:

We added a statement to clearly acknowledge that other methods do exist: "*…… Furthermore, REMA DEMs are registered to a control data set, such as airborne altimetric data or satellite altimetric data of ICESat and CryoSat-2 (Howat et al., 2019; Shean et al., 2019; Zinck et al., 2023). While the existing methods can be used to register DEMs, there is a need to develop a strict mathematical model to systematically improve the registration accuracy and to study mélange dynamics.*"

We also indicate the limitations of the current MDAM method by explicitly stating that the *GCP* and *TP* selection procedures are "manual": "The above GCP selection method is implemented as a manual procedure." "The TP selection method is also implemented as a manual procedure."

And applications in Pine Island and Thwaites where ice velocity is significantly faster, additional modification is needed: "...... *To make this MDAM model working in a more dynamic open ocean environment, such as Pine Island and Thwaites ice shelves where ice velocity is significantly higher, modifications need to be carried out to address the rapid calving process with incoherent mélange changes in ice shelf front. For example, tie points between adjacent sub DEMs may be selected with decreased time intervals to minimize the uncertainty caused by the compensation using ice velocity maps.*"

**Mélange dynamics analysis and discussion:**
**You discuss a few causes behind the observed rift widenings. However, I am still left with an impression that I do not know what caused the widening of the rifts. Why did the mélanges not freeze the rifts? How do you expect the rifts to continue their development? Is there anything to learn from the evolution of rift T2, which seems to be two rifts approaching each other and potentially merging? I think that the manuscript could benefit from a more detailed discussion on the impacts of the results. What have we learned from these results which we did not already know? Having a more detailed discussion on that would increase the impact of the paper, its findings, and also on the MDAM method itself.**

Response:

We appreciate your comments and questions regarding to mélange dynamics analysis and discussion. We revised the entire second paragraph in the Discussion section to address the relationship between mélange layer thinning and widening:

"*A thick mélange layer can effectively "freeze" a rift, enabling mechanical stress transmission between its flanks and ultimately suppressing rift propagation (Rignot and MacAyeal, 1998; Larour et al., 2021). In this study, we show that the mélange thickness in T1 and T2 decreased as the rifts widened. Thus, the mélange layer may have thinned for a relatively long period, passing a possible stage of rift freeze. We further propose that the mélange production itself is related to the rift widening process. The seracs inside the rifts are formed from partial collapsing of rift flanks. We suggest that the bottom part of the rift wall is first excavated through interactions by tides (Padman et al., 2002 and 2008), melting caused by intrusion of warmer sea water into the rift (Poinelli et al., 2023a), and other factors. This, in turn, causes the collapse of the upper part of the rift flank due to the removal of the bottom support. This process, like coastal bluff erosion, repeats itself and becomes one of the mechanisms that widen the rift and increase the mélange volume. We find that the increased mélange volume in a rift promotes its widening rate.*"

As suggested, we defined two rift centerlines for T2 so that the middle part of T2 does not look odd in Fig. 11. Originally, T2 initiated at the location of its current middle part. It then stopped to propagate and, instead, developed two long cracks, extending laterally in opposite directions with an offset in the middle. Now the cracks become two sub-rifts with their own mélange

layers. We believe that these two sub rifts will completely merge soon (see the Landsat 9 image of September 30, 2025 below). We also believe that one day T1 and T2 will merge, shortly before large icebergs will be formulated (like these in 1986). We are doing a systematic study of these processes and hope that we can report the results soon. Thanks again.

[Figure]

**Results/Methods structure:**

**The restructuring of the Data and Methods section has improved the readability of the paper greatly. However, there are still Methods elements present in the first section of the Results. In my opinion most of section "3.1 Bias correction and adjusted DEM time series" belongs in the Methods part of the paper and not in the Results, as it describes how you generate the final DEM timeseries. You could keep the figures in the Results, including a description of the improved uncertainty after the bias-correction. However, the sentences describing how you calculate different components (i.e. assigning different weights to TPs in L242-249, the validation method of the bias correction and the data used therefore in L283-291, etc.) should be moved to the Methods section in my opinion. You could also choose to move the entire section to the Methods.**

Response:

We moved the content of L242-249 to the Methods section, as suggested.

We felt that the validation part of L283-291 describes the result of using ICESat-2 data to check the adjusted DEMs. Considering both the order of the data processing result sequences and the nature of the actual computational results of this part, they would better stay in the Results section, hoping that you would agree.

**While rifts and calving are natural processes in the life cycle of ice shelves, the calving of their seaward-most extensions does not necessarily lead to significant upstream glacier acceleration. I have not performed the calculations myself, but rifts T1 and T2 on the Filchner-Ronne Ice Shelf (FRIS) appear to lie beyond the 'compressive arch' (Doake, 1998), suggesting they may not directly precondition the ice shelf for collapse—similar to**

the 'passive portion' described by Fürst et al. (2016). This raises a key question: if these rifts are not an immediate destabilizing factor for FRIS, why is their study important?

Response:

T1 and T2 are actually located in south of the passive ice boundary (green line in Fig. A1). We revised the paragraph to point out the potential destabilization: "*Although the icebergs calved from the shelf front in 1986 are mostly located inside the passive shelf ice area (Fig. A1), meaning no significant buttress reduction from the ice shelf (Doake et al., 1998; Fürst et al., 2016), the combined rifts of T1 and T2 propagated rapidly recently and have already covered ~58% the ice shelf laterally from Coats Land to Berkner Island. They have the potential to cause a calving before reaching the passive shelf ice boundary (green line in Fig. A1). Furthermore, warm water observed near Berkner Island (Davis et al., 2022) may accelerate the propagation processes of T1 and T2 and destabilize the ice shelf through a mechanism like that proposed for the Larsen C Ice Shelf (Poinelli et al., 2023a and 2023b).*"

**Specific comments:**

**L40-45: I would suggest rephrasing it to something like this: "Mélanges inside rifts, which consist of shelf ice, snow, sea ice, and water, have been investigated in relation to ice shelf fracturing in Antarctica and glacier calving in Greenland (References). Specifically, reductions in mélange thickness have been observed during rift widening on the Amery, Ronne, and Larsen C ice shelves (References). Furthermore, modelling results indicate that a mélange may …"**

Response:

We have rephrased the sentences accordingly: "*Mélanges inside rifts, which consist of shelf ice, snow, sea ice, and water, have been investigated in relation to ice shelf fracturing in Antarctica and glacier calving in Greenland (Rignot and MacAyeal, 1998; Larour et al., 2004; Cassotto et al., 2021). Specifically, reductions in mélange thickness have been observed during rift widening on the Amery, Ronne, and Larsen C ice shelves (Fricker et al., 2005; Walker et al., 2021; Larour et al., 2021). Furthermore, modelling results indicate that a mélange may……*"

**L63-64: This is where I suggest you to be careful with how you present the MDAM method as this sentence still reads as if there is a general lack of methods handling heterogenous offsets between individual DEMs. I, therefore, suggest you to acknowledge that other methods already exist, but all with a different goal in mind, than the 3D mélange dynamics presented in this paper.**

Response:

Thanks for your comment. We changed it to: "*While the existing methods can be used to register DEMs, there is a need to develop a strict mathematical model to improve the registration accuracy and to study 3D mélange dynamics.*"

**L190-204: Are the GCPs manually chosen? Or are they "computationally" selected based on the mentioned criteria? Please mention what you did in a revised manuscript.**

Response:

We added a sentence at the end of the paragraph: "*The above GCP selection method is implemented as a manual procedure.*"

**L210-224: Is it correctly understood that the TPs are manually selected? If so, please state that explicitly in the manuscript.**

Response:

Similarly, we added a sentence at the end of this paragraph: "*The TP selection method is also implemented as a manual procedure.*"

**L236-237: The comment about the observed melange thinning seems misplaced here in the Methods section. I would suggest to move it to the Results or Discussion.**

Response:

It is moved to the Results section: "…… *It may also indicate that the stress propagation between flanks may be compromised (Larour et al., 2004).*"

**L241-292: This is the part which I suggest that you (partly) move to the Methods section of the paper.**

Response:

We added a new subsection in the Methods section, to which we moved the method-related text in "L241-292".

"*2.4 Bias correction*

*The proposed MDAM system estimates the bias corrections* $(dX^k,\ dY^k,\ dZ^k)$ *for all sub-DEM* $DEM^k$ *(k = 1, 2, ..., N) through the least-squares procedure where their uncertainties are provided in the covariance matrix (McGlone et al., 2004). The REMA DEMs of 2016-2017 consists of 14 sub-DEMs that cover the entire front part of Filchner Ice Shelf (Fig. 3a). We use 12 GCPs that are measured by using ICESat-2 data to tie the connected DEMs to the grounded regions of Berkner Island and Coats Land where the average ice flow speed at the GCPs is ~8 m y-1. In total 51 TPs are used to connect the sub-DEMs on the floating part of the ice shelf, with 3-4 evenly distributed TPs in each overlapping area. Horizontal displacements at the TPs caused by the ice flow are on average ~90 m and are corrected by using the velocity map (Gardner et al., 2019). We establish the observation equations in Eqs. (1) and (2) using different weights. The weights for TPs are computed as inverse distances from the TPs to the nearest GCPs on grounded regions. Hence, the weights for GCPs are set to 1. Those TPs that are farther away from the grounded regions have smaller weights.*

*We process the second set of REMA DEMs of 2020-2021 (Fig. 3b) in the same way in the least-squares process. In addition, the ZY-3 DEM of 2014 is reconstructed as a cross-ice-shelf DEM*

*and does not need to go through this bias-correction process. The DEM is co-registered to the ICESat-2 ATL06 data of 2019 through a bundle adjustment procedure (McGlone et al., 2004; Li et al., 1998 and 2017b) using GCPs that are selected from "stable" features in the same way for those used in REMA sub-DEM co-registration.*"

The remaining part in 3.1 is also revised accordingly.

**Table 1: I do not see the need for this table as the numbers are illustrated much better in Figure 6 and 7.**

Response:

Table 1 is now merged in to Table A2 in Appendix A.

**Figure 10: Please specificy in both figure and caption which is T1 and which is T2.**

Response:

Thanks for your suggestion. "*T1*" and "*T2*" are added in Figures 10a, 10b, 10e, 10f and the caption.

**L321-322: I would suggest to reduce the number of digits: 47.132 km → 47.1 km, 48.324 km → 48.3 km, 1.457 km → 1.5 km, 1.532 km → 1.5 km**

Response:

We changed them as suggested.

**Figure 11: Would it maybe make more sense to use two different centerlines for T2 as it seems from Fig. A2 that T2 is maybe two separate rifts approaching eachother rather than one long rift. This would also help to avoid confusion as to why the elevation and volume is so much higher/bigger in the middle of the central section of T2.**

Response:

Thanks for the suggestion. Now, we defined two centerlines for T2 and recalculated elevation and volume changes. They are updated in text.

[Figure]

"*The established DEM time series unveils an overall decrease of 1.8 ± 0.6 m, at a rate of -0.4 ± 0.1 m y-1, in mélange elevation inside the two rifts from 2014 to 2021 (3.1 ± 0.4 m and 0.4 ± 0.4 m inside T1 (excluding cavity) and T2, respectively) (between blue and red lines in Figs. 11a and 11c). Correspondingly, the mélange thickness decreased by 10.0 ± 3.3 m (1.6 ± 0.4 m y-1) during the period (19.7 ± 2.5 m and 2.8 ± 2.2 m for T1 and T2, respectively). This mélange thickness reduction observed during rift widening on Filchner Ice Shelf coincides with the*

*earlier findings on Amery Ice shelf, Ronne Ice Shelf, and Larsen C Ice Shelf (Fricker et al., 2005; Walker et al., 2021; Larour et al., 2021). It may also indicate that the stress propagation between flanks may be compromised (Larour et al., 2004). More importantly, our results indicate that despite the significant decrease in mélange elevation from 2014 to 2021 (Figs. 11a and 11c), the inferred periodical mélange volumes reveal a rapid expansion of the mélange by $(10.15 \pm 0.03) \times 10^9 \, km^3$ (151%) during the period, $(7.93 \pm 0.03) \times 10^9 \, km^3$ for T1 and $(2.22 \pm 0.01) \times 10^9 \, km^3$ for T2, respectively (Fig. 11), mainly attributed to newly vacated space due to partial collapse of rift flanks, associated rift widening, and other rift-mélange interaction factors.*"

**L375-377: I don't fully understand this sentence. What does "this model" refer to? Is that MDAM? And if so, why wouldn't it work for Pine Island and Thwaites? What adjustments would be needed?**

Response:

We revised the sentences to clarify the issue: "*...... To make this MDAM model working in a more dynamic open ocean environment, such as Pine Island and Thwaites ice shelves where ice velocity is significantly higher, modifications need to be carried out to address the rapid calving process with incoherent mélange changes in ice shelf front. For example, tie points between adjacent sub DEMs may be selected with decreased time intervals to minimize the uncertainty caused by the compensation using ice velocity maps.*"

**L378-386: You mention that a thick melange layer can freeze a rift, but that is clearly not what happens at T1 and T2. Why is that? This entire paragraph in general seems loosely connected. As in, how does the fact that a thick melange layer can freeze rifts relate to the next sentence seracs inside the melange?**

Response:

We revised the entire paragraph to make it consistent to explain the relationship between the mélange layer and widening:

"*A thick mélange layer can effectively "freeze" a rift, enabling mechanical stress transmission between its flanks and ultimately suppressing rift propagation (Rignot and MacAyeal, 1998; Larour et al., 2021). In this study, we show that the mélange thickness in T1 and T2 decreased as the rifts widened. Thus, the mélange layer may have thinned for a relatively long period, passing a possible stage of rift freeze. We further propose that the mélange production itself is related to the rift widening process. The seracs inside the rifts are formed from partial collapsing of rift flanks. We suggest that the bottom part of the rift wall is first excavated through interactions by tides (Padman et al., 2002 and 2008), melting caused by intrusion of warmer sea water into the rift (Poinelli et al., 2023a), and other factors. This, in turn, causes the collapse of the upper part of the rift flank due to the removal of the bottom support. This process, like coastal bluff erosion, repeats itself and becomes one of the mechanisms that widen*

*the rift and increase the mélange volume. We find that the increased mélange volume in a rift promotes its widening rate, and may further impact the ice shelf stability.*"

**L385-386: Calving does not necessarily impact ice shelf stability and as you mention in the following paragraph it seems that these rifts are in areas which do not provide much buttressing. I would, therefore, suggest that you remove the ice shelf stability part of this sentence.**

Response:

Yes, we deleted the second half of the sentence.

**L385: Change Passive Shelf Ice (PSI) to passive shelf ice. There is no need for an acronym which you only use two times.**

Response:

They are changed in all places, except one in the caption of Figure A1 because it is marked in the figure.

**L404-405: Could you clarify how the MDAM system can bu used to study ice shelf instability and sea level rise contribution?**

Response:

This sentence is deleted.

**Typos and grammar:**

**General grammar comment: The manuscript misses quite a few definitive articles ("the") throughout the text, which makes the text difficult to follow at times. I have only marked a few of them in my review, but I recommend the authors to make use of a grammar tool or similar to locate the remaing missing definitive articles.**

Response:

Thank you for your comment. We have checked the usage of the definite article "the" and made changes throughout the entire manuscript.

**L28: Antarctic Ice Sheet (AIS)… → The Antarctic Ice Sheet (AIS)…**

Response:

We changed it accordingly.

**L49: Be aware of using the definite article "the". Change "earth surface features" to "the Earth's surface features"**

Response:

We changed it accordingly.

**L52-55: Be aware of using the definite article "the". Change to:"… for example, the 500 m ICESat (Dimarzio et al., 2007) and ICESat-2 (Shen et al., 2022) DEMs, and the 1000 m CryoSat-2 DEM (Slater et al., 2018). Those from optical and SAR satellite stereo mapping data are of higher resolutions and have the potential for geometric modeling and analysis of rifts and mélange features, including the 90 m TanDEM-X DEM (Wessel et al., 2021), the 30 m ASTER DEM (Tachikawa et al., 2011), and especially, the 2 m REMA DEM (Howat et al., 2019)."**
Response:
Thanks. We changed them accordingly.

**L68-69 (Figure 1 caption): Be aware of using the definite article "the". Change to "The LIMA mosaic … the RAMP DEM …"**
Response:
They are added.

**L81: Change to "The validated MDAM is applied to the Filchner…"**
Response:
We changed it accordingly.

**L83: Change to "… is the second largest ice shelf in Antarctica."**
Response:
We changed it accordingly.

**L125: Be aware of using the definite article "the". Change to: "at the two shelf margins"**
Response:
We changed it accordingly.

**L126: Change "cover the floating ice of the ice shelf." to "cover the floating ice shelf".**
Response:
We changed it accordingly.

**L155: Change to "In a similar way,"**
Response:
We changed it accordingly.

**L246: Change to: "on average"**
Response:

We changed it accordingly.

**References:**

**Shean, D. E., Joughin, I. R., Dutrieux, P., Smith, B. E., and Berthier, E.: Ice shelf basal melt rates from a high-resolution digital elevation model (DEM) record for Pine Island Glacier, Antarctica, The Cryosphere, 13, 2633–2656, https://doi.org/10.5194/tc-13- 2633- 2019, 2019.**

**Zinck, A.-S. P., Wouters, B., Lambert, E., and Lhermitte, S.: Unveiling spatial variability within the Dotson Melt Channel through high-resolution basal melt rates from the Reference Elevation Model of Antarctica, The Cryosphere, 17, 3785–3801, https://doi.org/10.5194/tc-17-3785-2023, 2023.**
**REFERENCES:**

---

## Author Response (AR3)

We appreciate the comments from the editor in this round of review. Again, they are very helpful in further improving the manuscript. In the following responses, we use "**bold**" text for comments, "non-bold" text for our responses, and "italic" for changed text in the revised manuscript.

**Comments from editor**

**Many thanks for updating your manuscript following the suggestions by the referees. I have carefully studied your response and believe your manuscript is now ready for acceptance, pending three minor changes (see below). Please include this in your manuscript, after which I will conduct a brief, final review myself.**

**Congratulations and thank your for contributing to our journal!**
**Bert Wouters**

**Line 45:Please change to "However, high-resolution observations covering large regions are scarce and small-scale mélange textures and 3D rift dynamics cannot be captured with a necessary accuracy."**
Response:
It is changed as suggested.

**Line 357: Please change to "For example, tie points between adjacent sub DEMs may be selected with decreased time intervals to minimize the uncertainty caused by the compensation of horizontal displacement due to ice dynamics using velocity maps"**
Response:
It is changed as suggested.

**Line 62 "While the existing methods can be used to register DEMs, there is a need to develop a strict mathematical model to improve the registration accuracy and to study 3D mélange dynamics.": The statement about improved registration accuracy would require an intercomparison between different methods, which is not part of your study. Please change to:**

**"While existing methods can be used to register DEMs, a mathematical model can help formalize the registration process and provide a consistent framework for quantifying and propagating uncertainties. Such a model-based approach may facilitate systematic comparison between different regions or time periods, and can offer clearer insight into the relationship between registration parameters and mélange dynamics."The manuscript "Building multi-satellite DEM time series for insight into mélange inside large**

**rifts in Antarctica" by Xia et al. presents a novel approach to monitor rifts' infill on the Filchner-Ronne ice shelf using observations from different satellites at high resolution. This is a revised version of the manuscript. I commend the authors for the time and effort spent in addressing my long list of comments. In particular, the addition of the discussion and the analysis on the second rift strongly improves the impact of this manuscript, in my opinion.**

Response:

It is changed as suggested.